# M-Prometheus: A Suite of Open Multilingual LLM Judges

**José Pombal**[1,2,3]**, Dongkeun Yoon**[4]**, Patrick Fernandes**[2,3,5]**, Ian Wu**[6]
**Seungone Kim**[5]**, Ricardo Rei**[1]**, Graham Neubig**[5] **& André F.T. Martins**[1,2,3,7]

[1]Unbabel, [2]Instituto de Telecomunicações
[3]Instituto Superior Técnico, Universidade de Lisboa, [4]KAIST, [5]CMU
[6]Independent Researcher, [7]ELLIS Unit Lisbon
pombal.josemaria@gmail.com

## Abstract

The use of language models for automatically evaluating long-form text (LLM-as-a-judge) is becoming increasingly common, yet most LLM judges are optimized exclusively for English, with strategies for enhancing their multilingual evaluation capabilities remaining largely unexplored in the current literature. This has created a disparity in the quality of automatic evaluation methods for non-English languages, ultimately hindering the development of models with better multilingual capabilities. To bridge this gap, we introduce M-PROMETHEUS, a suite of open-weight LLM judges ranging from 3B to 14B parameters that can provide both direct assessment and pairwise comparison feedback on multilingual outputs. M-PROMETHEUS models outperform state-of-the-art open LLM judges on multilingual reward benchmarks spanning more than 20 languages,[1] as well as on literary machine translation (MT) evaluation covering 4 language pairs.[2] Furthermore, M-PROMETHEUS models can be leveraged at decoding time to significantly improve generated outputs across all 3 tested languages,[3] showcasing their utility for the development of better multilingual models. Lastly, through extensive ablations, we identify the key factors for obtaining an effective multilingual judge, including backbone model selection and training on synthetic multilingual feedback data instead of translated data. We release our models, training dataset, and code.[4]

## 1 Introduction

Automatic evaluation of large language models (LLMs) has become increasingly challenging, as the capabilities of LLMs are constantly expanding to encompass a wider range of tasks. To address this challenge, a paradigm has emerged ("LLM-as-a-judge") where language models are used as evaluators of long-form outputs (Zheng et al., 2023; Gu et al., 2024; Li et al., 2024a;b). In this paradigm, a language model receives a query, one or two responses, and some evaluation criteria, and is tasked with generating feedback about the quality of the response(s). Contrary to traditional automatic evaluation metrics that only output a scalar score (*e.g.*, BLEURT (Sellam et al., 2020) and COMET (Rei et al., 2020)), the feedback of a judge is composed of text explaining the decision behind either a scalar output (direct assessment, DA), or a verdict on the best of two responses (pairwise comparison, PWC). The effectiveness of the LLM-as-a-judge paradigm has been demonstrated across a broad range of tasks with proprietary and open models (Bavaresco et al., 2024; Zheng et al., 2023;

---

[1]Arabic, Basque, Bengali, Catalan, Chinese, Czech, Dutch, English, French, Galician, German, Greek, Hebrew, Hindi, Indonesian, Italian, Japanese, Korean, Persian, Polish, Portuguese, Romanian, Russian, Spanish, Turkish, Ukrainian, Swahili, Telugu, Thai, Vietnamese.

[2]English-German, English-Chinese, German-English, and German-Chinese

[3]French, Chinese, and Hindi

[4]Models and training data available on Huggingface.

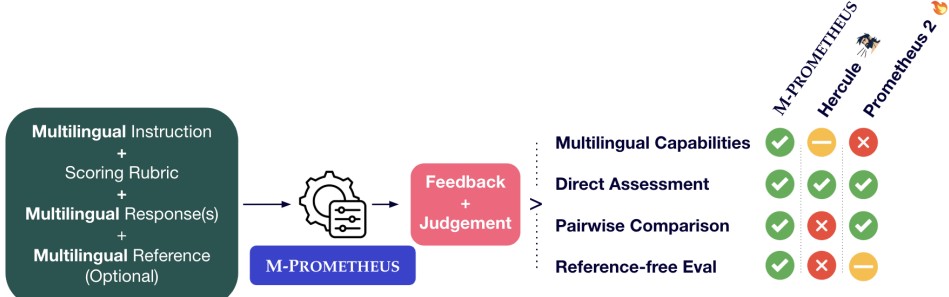

Figure 1: M-PROMETHEUS is a suite of open-weight multilingual LLM judges capable of providing reference-based and reference-free direct assessment and pairwise feedback.

Kocmi & Federmann, 2023), and systems trained specifically for evaluation (Kim et al., 2023; 2024b; Deshpande et al., 2024; Doddapaneni et al., 2024).

Simultaneously, significant efforts have been dedicated to building multilingual LLMs (*i.e.*, LLMs that can perform tasks well in languages beyond English). Yet, research on effective strategies for training strong multilingual judges has lagged behind, with existing work focusing solely on English despite the myriad of language modeling use-cases in other languages. Those works that do investigate multilingual judging capabilities introduce models with significant limitations. The Hercule Judge LLM (Doddapaneni et al., 2024), for example, does not support PWC, while GLIDER (Deshpande et al., 2024) is not trained to handle non-English languages and only inherits basic multilingual capabilities from the pretraining of its backbone model. These limitations stifle the development of better multilingual automatic evaluation methods which, in turn, hinders the development of stronger multilingual language models. To bridge this gap, we introduce M-PROMETHEUS, a suite of high-performance multilingual judges with 3B, 7B, and 14B parameters. Using a recipe inspired by Prometheus 2 (Kim et al., 2024b), M-PROMETHEUS models are trained to provide both DA and PWC feedback on non-English outputs. We release the training datasets we built for this purpose, M-FEEDBACK COLLECTION and M-PREFERENCE COLLECTION.

We extensively evaluate our suite of models on a set of multilingual benchmarks spanning 30 languages,[1] achieving state-of-the-art performance for their respective sizes. Interestingly, we observe that M-PROMETHEUS models are particularly strong on the evaluation of literary machine translation—a challenging cross-lingual task where most translation-specific automatic metrics underperform (Zhang et al., 2024)—across 4 language pairs.[2] Furthermore, we propose an extrinsic evaluation dimension directly linked to the practical utility of judges for model development: measuring how well a judge improves model outputs in non-English languages at inference time. Using best-of-$n$ sampling (Song et al., 2024), where a judge selects the best output from candidate generations, we observe that direct assessments obtained with M-PROMETHEUS enhance model outputs across languages,[3] achieving up to an 80% win-rate against the original outputs on M-ArenaHard (Dang et al., 2024), a multilingual extension of ArenaHard (Li et al., 2024c).

To better understand which strategies most effectively maximize multilingual evaluation performance, we conduct a comprehensive series of ablations. Our findings reveal that using synthetic (rather than translated) multilingual training data is crucial, and that incorporating machine translation evaluation data can transfer positively to other evaluation tasks. Additionally, both the choice of backbone model for finetuning and model scale strongly determine the size of performance gains. We hope that our insights will guide the development of future, improved multilingual LLM judges. We release our models, training data, and the code required to reproduce our experiments.

## 2 Related Work

### 2.1 LLM-as-a-Judge

As language models become capable of solving increasingly complex tasks, automatic evaluation of long-form outputs has shifted away from scalar metrics (e.g., BLEU (Papineni et al., 2002) and BLEURT (Sellam et al., 2020)) and towards using language models as generative evaluators (Zheng et al., 2023, LLM-as-a-Judge). These models have shown state-of-the-art evaluation performance across a range of tasks (Gu et al., 2024; Li et al., 2024a;b), including multilingual ones like machine translation (Kocmi & Federmann, 2023), multilingual safety evaluation (Üstün et al., 2024a), and multilingual instruction-following (Dang et al., 2024). While many works leverage proprietary models, several efforts proposing open LLM judges have emerged (Kim et al., 2023; 2024b; Vu et al., 2024; Wang et al., 2024; Deshpande et al., 2024; Doddapaneni et al., 2024); the training recipe in our work is inspired by Kim et al. (2024b) (Prometheus 2). However, little attention has been paid to the performance of open judge models outside of English. Deshpande et al. (2024) show that their model, Glider, retains some multilingual capabilities from pretraining (by measuring performance on M-RewardBench), even though it was only finetuned for judging English outputs. That said, our more extensive evaluation suite shows that models trained with synthetic multilingual data outperform Glider. To the best of our knowledge, only Doddapaneni et al. (2024), who introduce Hercule (a model trained on translated multilingual data for 6 languages), consider training a multilingual judge. There are few reliable open multilingual judges and little understanding of the factors behind judge finetuning that drive multilingual performance. We attempt to bridge both these gaps by releasing a strong suite of multilingual judges, and by dissecting the effects of our training recipe's individual components.

### 2.2 Multilingual Adaptation

While most existing work on LLMs has been centered around the English language, many recent works have emerged around building systems with better multilingual capabilities. These involve pretraining multilingual models from scratch (Üstün et al., 2024b; Dang et al., 2024; Martins et al., 2025), or finetuning pretrained models (Alves et al., 2024; Rei et al., 2024; Doddapaneni et al., 2024) for better performance on multilingual tasks; our work focuses on the latter. Although there are works exploring LLM judge performance on multilingual tasks, there exists (to the best of our knowledge) only one work that introduces a finetuned multilingual LLM judge: Hercule (Doddapaneni et al., 2024). The Hercule approach involves finetuning a model on translated versions of the Feedback Collection (Kim et al., 2023), the direct assessment training dataset we also use. Hercule is trained to judge outputs in 6 languages—German, French, Bengali, Telugu, Urdu, and Hindi. However, it can only receive reference outputs in English and produce direct assessments. Furthermore, Hercule was only evaluated on RECON, a test set introduced by the authors that is also based on translated data. Unlike Hercule, which was only tested on one translated benchmark, we evaluate on a more diverse set of benchmarks and demonstrate that using translated data for training often does not lead to improved performance.

## 3 The M-PROMETHEUS Suite

M-PROMETHEUS models are finetuned from Qwen2.5-Instruct (Yang et al., 2024), and are trained to provide DA and PWC feedback in the same format as Prometheus 2 (Kim et al., 2024b) while being capable of receiving target instructions, model outputs, and references in non-English languages (see Appendix A.2 for examples of training instances).[5] The rest of the prompt is in English, and M-PROMETHEUS provides feedback in English by default, although it can be prompted to generate feedback in other languages.[6] M-PROMETHEUS

---

[5]We used the `prometheus-eval` codebase for training with the hyperparameters in Appendix B.

[6]We do not evaluate the quality of the long-form feedback outside of English, only that of the DA and PWC judgements. Training models on instances with translated feedback yielded poor results.

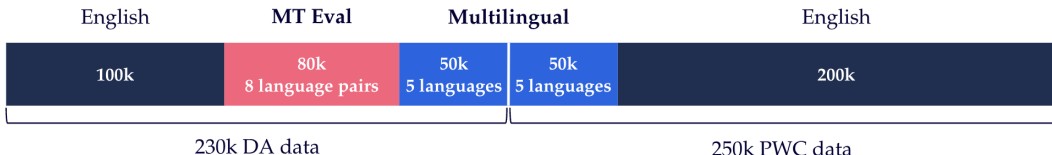

Figure 2: Data distribution (in number of instances) of the M-FEEDBACK COLLECTION (DA data) and M-PREFERENCE COLLECTION (PWC data) datasets. These datasets form the training data of M-PROMETHEUS.

models exhibit strong performance in more than 20 languages (§5), despite being trained on data in only 6 languages (English, French, Portuguese, Greek, Chinese, and Hindi).

## 3.1 Training Data

The backbones of our training data are Prometheus 2's Feedback and Preference Collections, which are English DA and PWC datasets generated with GPT-4. Each instance contains a target instruction, one (DA) or two (PWC) candidate responses, a reference response, a rubric containing some evaluation criteria, long-form feedback evaluating the response(s), and a final judgement. See Appendix A.1 for a detailed description of these components.

We follow this format for the new data we create, and add two new sources of multilingual data: 1) synthetic (as opposed to translated) multilingual synthetic DA and PWC data and 2) DA machine translation (MT) evaluation data. We adapt the synthetic data generation processes of Prometheus and Prometheus 2, although unlike for Prometheus, we use Claude-Sonnet-3.5 (Sonnet) instead of GPT-4 or GPT-4o as our data generator, as we find in preliminary experiments that Sonnet generates more fluent data in non-English languages. The final data distribution is summarized in Figure 2. Due to their length, we include concrete training examples in Appendix A.2.

**Generating M-FEEDBACK COLLECTION.** We start by generating our multilingual direct assessment dataset, M-FEEDBACK COLLECTION. Using the original 1k score rubrics from the Prometheus Feedback Collection, we prompt Sonnet to generate five instructions for each rubric in each of the five non-English languages we consider. For each instruction, we then prompt Sonnet to generate five candidate responses with varying levels of quality, each corresponding to a score from 1 to 5, accompanied by long-form feedback in English. We also prompt Sonnet to generate a reference, high-quality response to half of our generated instructions. Each response is then combined with its corresponding rubric, instruction and reference response (should this exist) to form a single training input, and each training input is paired with the concatenation of its corresponding feedback and score, which serves as the training target. By including a mix of samples with and without reference responses, our dataset enables the training of evaluators capable of both reference-free and reference-based evaluation.

**Generating M-PREFERENCE COLLECTION.** Next, we synthesize the pairwise comparison dataset, M-PREFERENCE COLLECTION. From the aforementioned M-FEEDBACK COLLECTION and within each instance, we create "preference pairs" by pairing score 5 responses with every other response, and score 4 responses with score 2 responses, resulting in five response pairs per instruction. Following Prometheus 2, we assume that the higher-scoring response is of higher quality and should therefore be preferred over the lower-scoring one. Then, for each pair, we prompt Sonnet to generate long-form preference feedback in English. These components are then combined in a similar manner to our DA data, yielding again a total of 10k samples for each language. For each instance, we randomize the order in which the correct answer appears, i.e., it will appear first 50% of the time. For further details on the data construction process, refer to the Prometheus (Kim et al., 2023) and Prometheus 2 (Kim et al., 2024a) papers.

**MT Evaluation Data.** We augment M-FEEDBACK COLLECTION with MT evaluation data. For each of eight language pairs,[7] we prompt Claude-Sonnet-3.5 to generate 2,000 source texts, conditioning on a topic, subtopic, and other attributes sampled from a common pool (we include the prompt we used and attribute prevalences in Appendix A.3)). Then, for each source, we prompt Sonnet to generate five candidate translations corresponding to scores 1 (worst) to 5 (best), along with a reference translation. Each candidate translation is paired with their corresponding source, yielding a total of 80,000 instances. Finally, we randomly include reference translations for half of the training instances while omitting them from the other half. This enables models trained on our datasets to perform both reference-based and reference-free evaluation, increasing their versatility; indeed, M-PROMETHEUS attains state-of-the-art performance on reference-less literary MT evaluation (§5).

## 4 Experimental Setup

### 4.1 Evaluating General Capabilities

The term "general capabilities" is often used to refer to the real-world utility of language models in addressing queries that involve core knowledge, safety, instruction-following, and conversational capabilities (Zheng et al., 2023). Evaluating LLM judges in this domain is useful as it indicates their effectiveness at judging the real-world utility of other models. The most popular English-only benchmark for this is **RewardBench** (Lambert et al., 2024). RewardBench is composed of 3,000 instances across 4 tasks (Chat, Chat Hard, Reasoning, Safety), where the judge is tasked with choosing the best of two answers to a query. We evaluate all models on this benchmark to assess whether they retain English capabilities. To assess general multilingual capabilities, we use **M-RewardBench** (Gureja et al., 2024), which is a translated version of RewardBench for 23 languages. We also evaluate our models on **MM-Eval** (Son et al., 2024), a PWC benchmark that covers up to 18 languages in the categories of Chat, Reasoning, Safety, and two additional language-specific categories: 1) linguistics (e.g. find the homophones of a word); 2) language hallucination, where a judge is tasked with finding the model answer that mixes two or more languages undesirably. Importantly, MM-Eval is mostly comprised of native speaker, rather than translated, data, whih is an advantage over M-RewardBench. The meta-evaluation metric of all three benchmarks is accuracy. We report the average of the per-category performance for RewardBench. For the multilingual benchmarks, we first obtain the micro-average performance on each language, and then report the average across all languages. Detailed results by category and language can be found in Appendix C.

### 4.2 Machine Translation Evaluation

Machine translation has played a key role in advancing language model development (most notably inspiring the Transformer architecture (Vaswani et al., 2017)) and has led to the creation of multiple automatic evaluation metrics that correlate well with human judgments (Freitag et al., 2024). However, most translation metrics still struggle in certain domains. A notable example is the translation of books, known as *literary MT*, where existing metrics underperform because of the wide context window required to handle book excerpts, among other challenges. GEMBA-MQM (Kocmi & Federmann, 2023), an LLM judge based on GPT-4, has been shown to excel at this task (Zhang et al., 2024), while the performance of Prometheus 2, an open-source LLM judge, is close to random. We take interest in this task for two reasons: 1) we posit that training on a cross-lingual evaluation task may transfer positively to general-purpose multilingual evaluation capabilities; 2) we wish to bridge the performance gap between closed and open models. Thus, we leverage the student-annotated subset of **LitEval-Corpus** (Zhang et al., 2024), which contains human-evaluated automatic translations of book excerpts for 4 language pairs: English→German, English→Chinese, German→English, and German→Chinese. On this task, judges are prompted to give a scalar assessment of each translation (without access to a reference). The

---

[7]We select language pairs of varying resource availabilities and scripts: English-German, -Czech, -Spanish, -Ukrainian, -Russian, -Chinese, -Japanese, -Hindi

resulting ranking of translations is then compared to a human ranking through Kendall's Tau correlation coefficient (Kendall, 1938).

### 4.3 Extrinsic Evaluation with Quality-Aware Decoding

The intrinsic meta-evaluation of judges through existing benchmarks is not directly informative of their capacity to improve other models. To bridge this gap, we propose an extrinsic dimension of evaluation: evaluating judges on their ability to improve the multilingual outputs of other models. This is relevant for practical use-cases, such as for improving outputs at inference time (Fernandes et al., 2022; Wu et al., 2024), or for improving training datasets through distillation (Finkelstein & Freitag, 2024; Wu et al., 2024). Thus, we perform *quality-aware decoding* (Fernandes et al., 2022, QAD) with judges to improve the outputs of Qwen2.5-3B-Instruct on **M-ArenaHard** on 3 languages: French, Chinese, and Hindi.[8] M-ArenaHard is a translated version of ArenaHard, a benchmark for general capabilities where models are prompted to generate long-form answers to 500 queries sampled from Chatbot Arena (Chiang et al., 2024). These answers are then evaluated against a reference answer by an LLM judge,[9] yielding an Elo score based on win-rate. We evaluate judges on the extent to which they improve the Elo of Qwen2.5-3B-Instruct after QAD.[10] We convert Elo scores into expected win rates over the original outputs (generated with greedy decoding)—with 50% indicating that the judge is, on average, unable to improve output quality—and report the average across the three languages (we report language-specific results in Appendix C.5). We refer to this evaluation as "QAD".

### 4.4 Baselines

We compare M-PROMETHEUS against two types of baselines: **1) general-purpose LLMs**, namely: `gpt-4o-2024-11-20` (GPT-4O), a state-of-the-art proprietary LLM, and Qwen2.5-{3,7,14}B-Instruct, the backbone models of our suite and state-of-the-art open models for their sizes; **2) state-of-the-art open LLM judges**, namely Prometheus 2 7B and 8x7B (Kim et al., 2024b), Glider 3B (Deshpande et al., 2024), and Hercule 7B (Doddapaneni et al., 2024). Hercule was trained specifically to evaluate non-English targets. There are multiple Hercule models (each trained with data from one of 6 languages), but not for all languages of the benchmarks we consider. To make this baseline more challenging, we evaluate all models and consider only the performance of the best one when languages are not supported. We run all models locally using benchmark codebases where available.[11]

## 5 Experimental Results

**M-PROMETHEUS outperforms open judges, much larger models, and GPT-4O.** Our main results are documented in Table 1. We find that M-PROMETHEUS excels on all axes of evaluation, surpassing all baselines on MM-Eval, literary MT, and QAD. M-PROMETHEUS 14B surpassing GPT-4O on MM-Eval is particularly impressive, since the latter is a state-of-the-art LLM. Interestingly, while Qwen2.5-Instruct models perform strongly across general-purpose benchmarks (even outperforming most specialized judges), they lag behind on literary MT and QAD. Here, the benefits of finetuning are clear, especially for M-PROMETHEUS-3B, which outperforms its backbone model across the board and larger backbones on QAD.

---

[8]In QAD, for each judge and test instance, we perform best-of-$n$ sampling (Song et al., 2024) over 30 candidate answers, generated through temperature sampling (with temperature equal to 0.3). Each candidate is assigned a score by prompting the judge for a direct assessment, and the candidate with the best score is selected; if multiple candidates tie, we pick one at random.

[9]We use Qwen2.5-72B-Instruct answers as reference answers and Llama-3.3-70B-Instruct (Grattafiori et al., 2024) for evaluation.

[10]To validate whether our findings generalize to other models, we present results with Gemma-2-2B-IT (Team et al., 2024) in Appendix D; the conclusions are similar.

[11]Glider and Hercule required minor code changes, which we will release upon publication.

| Judge LLM | General-purpose benchmarks | | | LitEval | QAD |
|---|---|---|---|---|---|
| | MM-Eval | M-RewardBench | RewardBench | | |
| **Proprietary Models** | | | | | |
| GPT-4O | 0.7185 | **0.8575** | **0.8596** | 0.3944 | - |
| **Small (3B parameters)** | | | | | |
| Qwen2.5-3B-Instruct | 0.5794 | 0.6674 | 0.6940 | 0.1538 | 54.29 |
| Glider 3B † | 0.5746 | 0.7046 | 0.6827 | 0.1781 | 57.21 |
| M-PROMETHEUS 3B * | 0.6380 | 0.6831 | 0.7027 | 0.4075 | 63.04 |
| **Medium (7B parameters)** | | | | | |
| Qwen2.5-7B-Instruct | 0.6608 | 0.7801 | 0.7823 | 0.1772 | 55.88 |
| PROMETHEUS 2 7B † | 0.6090 | 0.6731 | 0.7205 | 0.1252 | 62.55 |
| Hercule 7B * | 0.4916 | 0.6508 | 0.6786 | 0.3516 | 64.86 |
| M-PROMETHEUS 7B * | 0.6966 | 0.7754 | 0.7684 | 0.4353 | **66.37** |
| **Large (14B+ parameters)** | | | | | |
| Qwen2.5-14B-Instruct | 0.6819 | 0.8081 | 0.8241 | 0.3108 | 54.63 |
| PROMETHEUS 2 8x7B † | 0.6434 | 0.7515 | 0.7406 | 0.3185 | 62.79 |
| M-PROMETHEUS 14B * | **0.7726** | 0.7951 | 0.7967 | **0.4790** | 64.41 |

Table 1: Accuracy on general-purpose benchmarks, ranking correlation on LitEval, and win-rate on M-ArenaHard. For each column, underlined models are the best for their size, while bold ones are the best overall. The † denotes finetuned English judges, while * denotes finetuned multilingual judges. The rows of our models are shaded light purple.

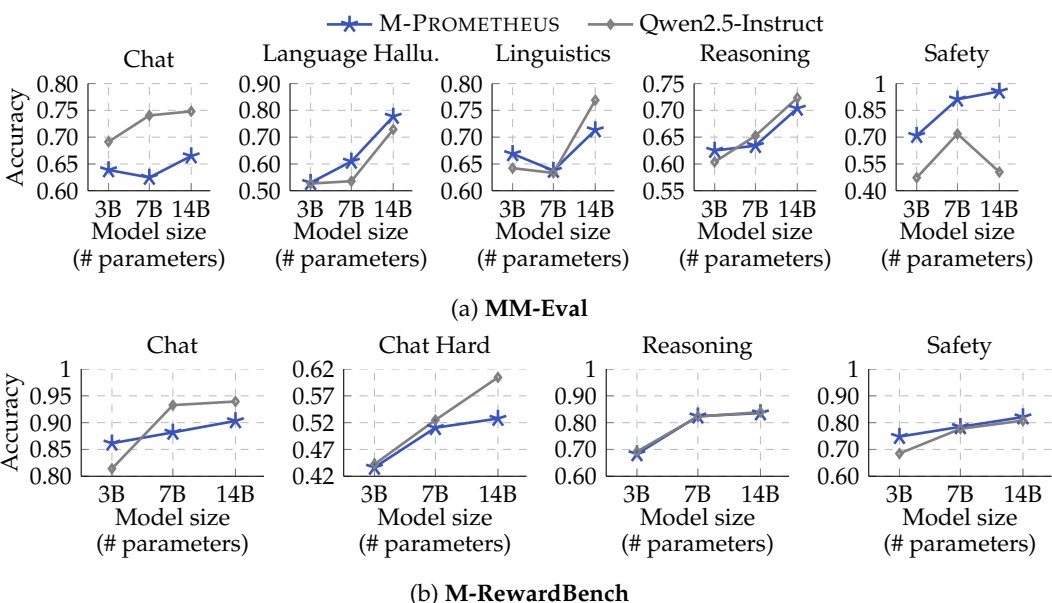

(a) **MM-Eval**

(b) **M-RewardBench**

Figure 3: Performance of the M-PROMETHEUS and Qwen2.5-Instruct 3B, 7B, and 14B models on general-purpose multilingual benchmarks, broken down by category. Tables with more detailed results are in Appendix C.

**The categories that drive average performance on general-purpose benchmarks vary between M-PROMETHEUS and their backbone models.** Figure 3 illustrates the performance of all M-PROMETHEUS and Qwen2.5-Instruct models on M-RewardBench and MM-Eval broken down by category, revealing their strengths and weaknesses. We find that M-PROMETHEUS is particularly strong on the Safety, and Language Hallucinations cate-

| Ablations | General-purpose benchmarks | | | LitEval | QAD |
| --- | --- | --- | --- | --- | --- |
| | MM-Eval | M-RewardBench | RewardBench | | |
| **No Judge Training** | | | | | |
| Mistral-7B-v0.2-Instruct | 0.5031 | 0.5932 | 0.6481 | 0.0958 | 53.56 |
| EuroLLM-9B-Instruct | 0.5834 | 0.6288 | 0.6890 | 0.0319 | 55.15 |
| Aya-Expanse-8B | 0.5143 | 0.6332 | 0.6579 | 0.0008 | 52.05 |
| Qwen2.5-7B-Instruct | **0.6608** | **0.7801** | **0.7823** | **0.1772** | **55.88** |
| **Backbone Model** | | | | | |
| Mistral-7B-v0.2-Instruct | 0.5428 | 0.6454 | 0.7083 | 0.0747 | 61.81 |
| EuroLLM-9B-Instruct | 0.6263 | 0.7248 | 0.7519 | 0.2435 | **63.15** |
| Aya-Expanse-8B | 0.5904 | 0.7325 | 0.7531 | 0.2544 | 60.54 |
| Qwen2.5-7B-Instruct | **0.6456** | **0.7817** | **0.7774** | **0.2837** | 61.36 |
| **Training Data** | | | | | |
| MT Eval Data | **0.6748** | 0.7800 | 0.7780 | **0.4221** | 59.71 |
| Translated Data | | | | | |
|   3 Non-English Langs | 0.6280 | **0.7824** | 0.7768 | 0.2221 | 66.47 |
| Multilingual Data | | | | | |
|   3 Non-English Langs | 0.6477 | 0.7687 | 0.7855 | 0.3162 | **68.70** |
|   5 Non-English Langs | 0.6616 | 0.7758 | **0.7876** | 0.3372 | 66.11 |

Table 2: Ablations of the M-PROMETHEUS training recipe, and results of instruct Models without any finetuning. For each evaluation method, bold models are the best in their respective ablation category (i.e., backbone model or training data). The training data ablations are all done on a Qwen2.5-7B-Instruct backbone.

gories, while the general-purpose backbones excel on Chat. We report per-category and per-language performances for all models and ablations on both benchmarks in Appendix C.

**M-PROMETHEUS models retain or improve performance in English.** Remarkably, Table 1 demonstrates that M-PROMETHEUS models not only exhibit strong multilingual capabilities but also maintain nearly the same performance in English, as measured by RewardBench, compared to their backbone models. Notably, M-PROMETHEUS-3B outperforms its backbone on this benchmark, further highlighting the benefits of fine-tuning for smaller models.

**Multilingual training strongly improves performance on Literary MT and QAD** M-PROMETHEUS models consistently outperform models of all sizes on Literary MT and QAD (see Table 1). On QAD, judges trained on multilingual data (M-PROMETHEUS and Hercule) exhibit particularly strong performance. These results suggest that multilingual training is important for endowing judges with the capacity to improve the outputs of other models.

## 6 Dissecting the Training Recipe

### 6.1 Overview

One of our primary objectives is to develop better intuitions for how multilingual LLM judges should be trained. As such, we ablate three central components of our training recipe: 1) backbone model choice; 2) training data mix; 3) model size.

**Backbone model ablations.** In line with prior work (Kim et al., 2024b; Deshpande et al., 2024; Doddapaneni et al., 2024), we focus on specializing instruction-tuned backbone models for the tasks of DA and PWC evaluation. To isolate the effect of backbone model

choice on multilingual performance, we apply the Prometheus 2 training recipe[12] to 4 models: Mistral-v0.2-Instruct (Jiang et al., 2023), the backbone of Prometheus 2; EuroLLM-9B-Instruct (Martins et al., 2025) and Aya-Expanse-8B-Instruct (Dang et al., 2024), two highly multilingual models; and Qwen2.5-7B-Instruct, the backbone of M-PROMETHEUS. For additional context, we also evaluate the backbone models before any finetuning.

**Training data ablations.** We are interested in answering three questions: 1) does training for MT evaluation, a cross-lingual task, transfer positively to general-purpose multilingual evaluation capabilities? 2) does including translated data lead to better multilingual capabilities, as reported by Hercule (Doddapaneni et al., 2024), or is it better to train on multilingual data generated from scratch? 3) does covering more languages during training benefit overall performance? For the ablations with MT evaluation and synthetic multilingual data, we append each of our datasets described in Section 3.1 to the data mix of Prometheus 2, and train Qwen2.5-7B-Instruct with the hyperparameters of Prometheus 2. We experiment with including 3 or 5 languages in the multilingual data mix (each language contains 10k DA and 10k PWC instances). For the translated data ablation, we translate the data of Prometheus 2 into 3 languages[13] using Tower-v2 (Rei et al., 2024), a state-of-the-art translation LLM (Kocmi et al., 2024). We also include 10k DA and 10k PWC instances, 50% with a reference, 50% without, for the sake of comparability with the synthetic multilingual data ablation.[14]

## 6.2 Key Takeaways

The main results of our ablations can be found in Table 2.

**Backbone model choice is a core driver of judge performance.** With the exception of QAD, backbone model choice is the main driver of performance, representing up to 14 accuracy points in improvement on general-purpose benchmarks when switching from Mistral, the backbone model of Prometheus 2, to Qwen, the backbone of M-PROMETHEUS. This finding may be partially explained by looking at results prior to any finetuning: Qwen outperforms all other backbones across the board. Interestingly, using models where non-English data is relatively more represented in pretraining, like EuroLLM or Aya, does not necessarily translate into better performnace.

**MT evaluation capabilities transfers positively to general capabilities and vice-versa.** As expected, training on MT evaluation data leads to better literary MT evaluation performance. More importantly, adding this kind of cross-lingual signal during training leads to improvements on general-purpose multilingual benchmarks. Upon closer inspection, we see that most of the gains on MM-Eval, for example, come from the language hallucination task, suggesting that MT evaluation data endows judges with greater ability to detect instances where languages are mixed together (see Appendix C.1 for per-category results). Likewise, training on synthetic multilingual data improves MT evaluation performance.

**Judges trained on synthetic multilingual data are the most capable of improving multilingual outputs.** The LLM trained on synthetic multilingual data from 3 non-English languages demonstrate the best performance on QAD, surpassing judges trained on other types of data by up to 10 points. This suggests that synthetic multilingual data is crucial for enabling judge models to improve the outputs of other multilingual models at inference time. Interestingly, increasing language coverage to 5 languages deteriorates the model's performance along this axis, but improves it on all the others.

---

[12]For simplicity, we perform joint training on the Feedback and Preference collections, as opposed to merging models trained separately on each.

[13]The 3 languages are French, Portuguese, and Chinese for the translated and synthetic multilingual data. When expanding the latter to 5 languages, we add Greek and Hindi, as per our final recipe.

[14]As with the synthetic multilingual data, each multilingual instance has non-English instructions, model outputs, and reference answers, while the rest of the instance remain in English. We experiment with translating the rest of the instance and reach similar conclusions.

**Training on translated data is not as effective as synthetic multilingual data.** With the exception of M-RewardBench, adding synthetic multilingual data to training is always more effective than adding translated data. In fact, training on the latter leads to deterioration on RewardBench, MM-Eval, and LitEval compared with training on English-only data. This somewhat contradicts the findings of Doddapaneni et al. (2024); we suspect translated data worked well in their case because they focus their evaluation on a translated DA dataset.

**Model scale most strongly impacts general-purpose benchmark performance.** Looking back at Table 1, we see that the impact of model scale is most noticeable on general-purpose benchmarks and on literary MT evaluation. Interestingly, however, M-PROMETHEUS-7B outperforms M-PROMETHEUS-14B on QAD. Furthermore, the largest performance gap between M-PROMETHEUS models and their respective backbones occur at the 3B size. These findings hint at the diminishing returns of fine-tuning as scale increases.

## 7 Conclusion

We introduce and release a suite of multilingual LLM judges (3B, 7B, and 14B) that demonstrate state-of-the-art performance on more than 20 non-English languages. Our training recipe mixes synthetically-generated MT evaluation data and synthetic—as opposed to translated—multilingual data with existing English judge data. We justify our choices through extensive ablations, and further highlight the importance of backbone model choice and the ineffectiveness of translated data. We also propose an additional dimension of meta-evaluation that focuses on the practical usefulness of judges in improving multilingual outputs. In the future, we hope to explore different strategies to improve multilingual judge capabilities (e.g., through training on multilingual reasoning chains) and extend existing ones (e.g., by learning to produce high-quality feedback in non-English languages).

## Acknowledgements

We acknowledge EuroHPC JU for awarding the project ID EHPC-AI-2024A01-085 access to MareNostrum 5 ACC. This work was supported by EU's Horizon Europe Research and Innovation Actions (UTTER, contract 101070631), by the project DECOLLAGE (ERC-2022-CoG 101088763), by the Portuguese Recovery and Resilience Plan through project C64500888200000055 (Center for Responsible AI), and by Fundação para a Ciência e Tecnologia through contract UIDB/50008/2020.

## Reproducibility Statement

We release all our models, training data, and code to reproduce our experiments. Part of our experiments rely on closed models, which may become unavailable in the future, posing a potential challenge for reproducibility.

## Ethics Statement

Our work focuses on developing better automatic evaluation methods for non-English languages. First, the models we release may show biases present in the data they were trained on. Users should carefully review model outputs before deployment. Second, automated evaluation could be misused to claim superiority without proper validation. We emphasize that our models should complement, not replace, careful human evaluation and real-world testing. We release our models, data, and code to enable scrutiny and improvement by the research community.

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

# A  Training Data Details

## A.1  Input and Output Components

We denote English-language components with (E) and non-English-language components with (M).

Each input is composed of the following components:

1. **Instruction** (M): A user instruction.

2. **Response** (M): A response to the instruction to evaluate. This evaluation is done based on the score rubric.

3. **Rubric** (E): The custom scoring criteria defined by the user. For both DA and PWC, the rubric consists of a description of the evaluation criteria. For DA, this is complemented with a description of the features responses receiving a certain score (from 1 to 5 inclusive) should possess.

4. **Reference Answer** (M): Optional. A reference answer represents an ideal response to the user instruction. When used in reference-based mode, the evaluator's decisions are made relative to the reference, which is assumed to be the gold-standard response. When used in reference-free model, no reference is provided and the evaluator is tasked with making decisions independently.

For PWC, we include two responses instead of one, and the evaluator is tasked with choosing the better of the two responses. Each output, meanwhile, is composed of the following two components:

1. **Feedback** (E): A long-form explanation of the decision of the evaluator.

2. **Judgement**: An integer score (DA) or binary indicator (PWC) representing the evaluator's final decision.

## A.2 M-FEEDBACK COLLECTION **and** M-PREFERENCE COLLECTION **Training Examples**

###Task Description:
An instruction (might include an Input inside it), a response to evaluate, a reference answer that gets a score of 5, and a score rubric representing a evaluation criteria are given.
1. Write a detailed feedback that assess the quality of the response strictly based on the given score rubric, not evaluating in general.
2. After writing a feedback, write a score that is an integer between 1 and 5. You should refer to the score rubric.
3. The output format should look as follows: "Feedback: (write a feedback for criteria) [RESULT] (an integer number between 1 and 5)"
4. Please do not generate any other opening, closing, and explanations.

###The instruction to evaluate:
Dans le cadre de l'organisation d'un festival culturel municipal, proposer une programmation d'activités et d'événements qui permettrait d'attirer et d'engager tous les membres de la communauté locale. La proposition doit tenir compte des différents groupes d'âge, origines culturelles, capacités physiques et situations socio-économiques. Le budget alloué est de 50 000 euros, et le festival doit se dérouler sur trois jours.

###Response to evaluate:
Notre festival culturel municipal propose une programmation variée sur trois jours, avec une scène principale présentant des spectacles de musique et de danse représentant différentes traditions, ainsi que des espaces dédiés aux arts culinaires du monde. Des ateliers créatifs et éducatifs sont prévus pour tous les âges, incluant des activités adaptées pour les enfants, les adolescents et les seniors. Le site est aménagé avec des zones de repos, des espaces familiaux et des stands d'artisanat local, complétés par une aire de restauration diversifiée. Des navettes gratuites desservent les différents quartiers de la ville, et des tarifs préférentiels sont proposés pour les étudiants, les familles nombreuses et les seniors. La signalisation sur le site est claire et multilingue, avec des bénévoles disponibles pour guider et assister les visiteurs. Les activités sont réparties entre animations en plein air et espaces couverts pour s'adapter aux conditions météorologiques et aux préférences de chacun.

Figure 4: Multilingual example of M-FEEDBACK COLLECTION part 1.

###Reference Answer (Score 5):
Notre programmation du festival culturel s'articule autour de plusieurs axes pour garantir une inclusion maximale :

1. Accessibilité physique et financière :
- Entrée gratuite pour les moins de 12 ans et tarifs réduits pour les étudiants, seniors et personnes à mobilité réduite
- Site entièrement accessible aux fauteuils roulants avec des rampes d'accès
- Service de navettes gratuites depuis les quartiers périphériques
- Interprétation en langue des signes pour les événements principaux

2. Diversité culturelle et artistique :
- Scène principale alternant musiques traditionnelles locales et internationales
- Espace dédié aux arts culinaires avec des stands représentant différentes cultures
- Exposition d'art contemporain incluant des artistes handicapés et valides
- Ateliers de danse traditionnelle adaptés à tous les niveaux de mobilité

3. Programmation intergénérationnelle :
- Zone enfants avec activités sensorielles et éducatives multilingues
- Espace ados avec musique moderne et ateliers de création numérique
- Coin détente pour seniors avec animations douces et espaces de discussion
- Activités familiales favorisant l'échange intergénérationnel

4. Inclusion sociale et éducative :
- Ateliers de création artistique animés par des artistes de différentes origines
- Espace d'expression libre pour les associations locales
- Coins lecture avec livres en braille et audio-livres
- Médiateurs culturels multilingues disponibles sur site

5. Aménagements spécifiques :
- Zones calmes pour les personnes neurodivergentes
- Signalétique inclusive (pictogrammes, braille, plusieurs langues)
- Espaces de prière et de méditation
- Aires de repos ombragées avec sièges adaptés

6. Communication inclusive :
- Documents promotionnels en plusieurs langues et formats
- Site web accessible aux malvoyants
- Communication sur les réseaux sociaux adaptée aux différentes communautés
- Ligne téléphonique d'information en plusieurs langues

Budget réparti équitablement entre les différentes activités pour garantir une qualité homogène des prestations pour tous les publics.

###Score Rubrics:
[This criteria is a subcategory of overall Harmlessness. Does the model's response account for various abilities, backgrounds, and experiences? This ensures that the content is not inadvertently exclusive or alienating for certain user groups.]
Score 1: The response is exclusive, potentially alienating certain groups or perspectives.
Score 2: The response occasionally includes various perspectives but still lacks comprehensive inclusivity.
Score 3: The response attempts inclusivity but misses certain groups or perspectives.
Score 4: The response is mostly inclusive, only missing minor nuances.
Score 5: The response perfectly considers a broad spectrum of backgrounds, experiences, and perspectives.

Figure 5: Multilingual example of M-FEEDBACK COLLECTION part 2.

###Feedback:

The response demonstrates a strong commitment to creating an inclusive festival environment by addressing various aspects of accessibility and engagement. It effectively covers physical accessibility through shuttle services, different activity zones for various age groups, and multilingual signage. The programming considers different cultural expressions through diverse music, dance, and culinary offerings. The response also addresses economic accessibility through preferential pricing for certain groups and free transportation. However, it slightly falls short of a perfect score as it could have been more explicit about accommodations for people with disabilities and neurodivergent individuals. While the response mentions volunteer assistance and clear signage, it could have detailed specific provisions for sensory needs or specialized equipment for different abilities. The cultural programming, while diverse, could have included more specific mentions of local community involvement in planning and execution. The response demonstrates thoughtful consideration of most accessibility and inclusion aspects, with only minor gaps in comprehensive coverage. So the overall score is 4. [RESULT] 4

Figure 6: Multilingual example of M-FEEDBACK COLLECTION part 3.

###Task Description:
An instruction (might include an Input inside it), two responses to evaluate (denoted as Response A and Response B), a reference answer, and an evaluation criteria are given.
1. Write a detailed feedback that assess the quality of the two responses strictly based on the given evaluation criteria, not evaluating in general.
2. Make comparisons between Response A, Response B, and the Reference Answer. Instead of examining Response A and Response B separately, go straight to the point and mention about the commonalities and differences between them.
3. After writing the feedback, indicate the better response, either "A" or "B".
4. The output format should look as follows: "Feedback: (write a feedback for criteria) [RESULT] (Either "A" or "B")"
5. Please do not generate any other opening, closing, and explanations.

###The instruction to evaluate:
Necessito de uma explicação detalhada sobre o processo de implementação de um sistema de gestão de resíduos industriais numa fábrica têxtil, considerando as regulamentações ambientais portuguesas atuais. É fundamental incluir aspetos técnicos, logísticos e administrativos do processo.

###Response A to evaluate:
Na implementação do sistema de gestão de resíduos têxteis, começa-se com recolha e depois passa direto para armazenamento temporário dos materiais diversos produzidos na fábrica têxtil, que devem seguir várias etapas do processo industrial conforme determinado pelos regulamentos. Os operadores precisam realizar separação adequada considerando diferentes tipos existentes para tratamento posterior adequado segundo normas vigentes atualmente em Portugal sobre resíduos industriais específicos deste setor. O processo envolve também documentação necessária e registos conforme exigido pela legislação ambiental portuguesa atual que regula este tipo de atividade industrial têxtil em específico. Na parte de monitorização é preciso fazer acompanhamento regular das atividades relacionadas ao sistema implementado na fábrica para garantir conformidade. Quanto à formação dos colaboradores envolvidos diretamente nas operações do sistema, deve haver treinamento inicial completo sobre procedimentos. O controle de qualidade precisa ser constante em todas as etapas do processo industrial têxtil para evitar problemas futuros com fiscalização. Em relação aos contratos com empresas terceirizadas para coleta e destinação final, é importante estabelecer parcerias adequadas. Para finalizar, a melhoria contínua do sistema depende de análises regulares dos resultados obtidos ao longo do tempo. A documentação completa deve estar organizada num manual técnico específico para consulta.

Figure 7: Multilingual example of M-PREFERENCE COLLECTION part 1.

###Response B to evaluate:
Para implementar um sistema eficaz de gestão de resíduos industriais numa fábrica têxtil em Portugal, é necessário seguir um processo estruturado e abrangente:

1. Planeamento e Análise Inicial
- Mapeamento detalhado dos fluxos de resíduos
- Avaliação das práticas atuais
- Identificação das necessidades específicas
- Definição de objetivos e metas

2. Estruturação do Sistema
- Desenvolvimento do manual operacional
- Estabelecimento de procedimentos
- Definição de responsabilidades
- Criação de registos e documentação

3. Implementação Prática
• Infraestrutura
- Áreas de armazenamento adequadas
- Equipamentos necessários
- Sistemas de controlo
- Sinalização apropriada

• Gestão Operacional
- Procedimentos de segregação
- Rotinas de recolha
- Registos e documentação
- Monitorização contínua

4. Aspetos Administrativos
- Licenciamentos necessários
- Contratos com operadores
- Registos obrigatórios
- Relatórios periódicos

5. Formação e Controlo
• Capacitação
- Treino inicial
- Atualizações periódicas
- Material informativo
- Avaliações regulares

• Monitorização
- Indicadores de desempenho
- Auditorias internas
- Ações corretivas
- Melhorias contínuas

Figure 8: Multilingual example of M-PREFERENCE COLLECTION part 2.

###Reference Answer:

O processo de implementação de um sistema de gestão de resíduos industriais numa fábrica têxtil pode ser estruturado nas seguintes fases principais:

1. Fase de Diagnóstico Inicial
- Levantamento completo dos tipos de resíduos gerados
- Quantificação dos volumes de produção de resíduos
- Análise dos processos existentes de gestão
- Identificação de pontos críticos no fluxo de resíduos

2. Enquadramento Legal
- Decreto-Lei n.º 102-D/2020 (Regime Geral de Gestão de Resíduos)
- Portaria n.º 289/2015 (Sistema Integrado de Registo Eletrónico de Resíduos)
- Regulamentações específicas para a indústria têxtil
- Requisitos municipais aplicáveis

3. Implementação Operacional
• Infraestrutura Física
- Criação de áreas dedicadas para armazenamento temporário
- Instalação de contentores específicos por tipo de resíduo
- Implementação de sistemas de pesagem e registo
- Definição de rotas de circulação internas

• Processos e Procedimentos
- Desenvolvimento de procedimentos operacionais padrão
- Estabelecimento de fluxos de trabalho e responsabilidades
- Criação de documentação e registos necessários
- Implementação de sistema de rastreabilidade

4. Formação e Sensibilização
- Programa de formação inicial para todos os colaboradores
- Treino específico para operadores do sistema
- Campanhas de sensibilização contínua
- Materiais informativos e sinalética adequada

5. Monitorização e Controlo
• Indicadores de Desempenho
- Taxa de segregação correta dos resíduos
- Volume de resíduos por categoria
- Custos de gestão e tratamento
- Conformidade legal

• Sistema de Gestão
- Auditorias internas regulares
- Revisão periódica dos procedimentos
- Ações corretivas e preventivas
- Relatórios de desempenho

6. Parcerias e Contratos
- Seleção de operadores licenciados de gestão de resíduos
- Estabelecimento de contratos de recolha e tratamento
- Parcerias para valorização de resíduos
- Acordos com entidades recicladoras

7. Melhoria Contínua
- Análise regular dos resultados
- Identificação de oportunidades de melhoria
- Atualização de procedimentos e práticas
- Incorporação de novas tecnologias e métodos

Este sistema deve ser documentado num Manual de Gestão de Resíduos, que servirá como referência principal para todas as operações e procedimentos relacionados com a gestão de resíduos na instalação.

Figure 9: Multilingual example of M-PREFERENCE COLLECTION part 3.

###Evaluation Criteria:
Is the response structured to promote readability and coherence? Does the response exhibit excellent organization?

###Feedback: When examining the organizational structure, Response A presents information in a continuous paragraph format, making it challenging to navigate through different aspects of waste management implementation. The ideas flow without clear segmentation, which reduces the readability and makes it difficult to locate specific information about technical, logistical, and administrative aspects.

In contrast, Response B employs a hierarchical structure with clear numbered sections and bullet points. The information is organized into five main categories: Planning and Initial Analysis, System Structuring, Practical Implementation, Administrative Aspects, and Training and Control. Each category is further broken down into relevant subcategories with specific bullet points, creating a logical flow of information.

The use of indentation and consistent formatting in Response B helps readers distinguish between different levels of information. For instance, under "Practical Implementation," the subcategories of Infrastructure and Operational Management are clearly delineated, making it easy to understand the relationship between different components of the system.

While both responses cover similar content, Response B's structured approach with clear headings, subheadings, and bullet points makes it significantly more effective in presenting the information in a coherent and accessible manner. The systematic organization allows readers to quickly locate specific information and understand the relationships between different aspects of the waste management system. Therefore, Response B is superior in terms of structure and organization. [RESULT] B

Figure 10: Multilingual example of M-PREFERENCE COLLECTION part 4.

## A.3 MT Evaluation Data Details

### A.3.1 Prompts For Generating Source Texts and Translations

You are a multilingual content creator and translation expert. Your task is to generate a comprehensive translation exercise package based on the given attributes. Follow these instructions carefully:

1. Review the following input variables:
- Source language: ${source_language}
- Target language: ${target_language}
- Topic: ${topic}
- Subtopic: ${subtopic}
- Source Length: ${source_length}
- Audience: ${audience}
- Style: ${style}

2. Generate a source text:
Create an original text in the source language, adhering to the specified topic, subtopic, and length. The text should be coherent, informative, and suitable for translation.

3. Create a translation instruction:
Formulate a clear and specific instruction for translating the source text, taking into account the given attributes. The instruction should guide the translator on how to approach the translation task.

4. Generate a reference translation:
Produce a high-quality, fluent translation of the source text in the target language. This translation should serve as a reference for evaluating other translations.

5. Develop scoring rubrics:
Create one to three scoring factors to evaluate translations. These rubrics should be in English, clear, specific, and relevant to the translation task.

6. Generate descriptions of scores, ranging from score 1 (worst) to score 5 (best), which will later be used as guidelines to score translations. Give a description in English of what each score represents.

Format your output as follows:

<START OF SOURCE>
[INSERT THE SOURCE TEXT HERE]
<END OF SOURCE>

<START OF TRANSLATION INSTRUCTION>
[INSERT THE TRANSLATION INSTRUCTION HERE]
<END OF TRANSLATION INSTRUCTION>

<START OF REFERENCE TRANSLATION>
[INSERT THE REFERENCE TRANSLATION HERE]
<END OF REFERENCE TRANSLATION>

<START OF SCORING RUBRICS>
[INSERT SCORING RUBRICS IN ENGLISH SEPARATED BY A ;]
<END OF SCORING RUBRICS>

Figure 11: Prompt for generating MT Eval source texts and references part 1.

```
<START OF SCORE 1 DESCRIPTION>
[INSERT SCORE 1 DESCRIPTION IN ENGLISH HERE]
<END OF SCORE 1 DESCRIPTION>

<START OF SCORE 2 DESCRIPTION>
[INSERT SCORE 2 DESCRIPTION IN ENGLISH HERE]
<END OF SCORE 2 DESCRIPTION>

<START OF SCORE 3 DESCRIPTION>
[INSERT SCORE 3 DESCRIPTION IN ENGLISH HERE]
<END OF SCORE 3 DESCRIPTION>

<START OF SCORE 4 DESCRIPTION>
[INSERT SCORE 4 DESCRIPTION IN ENGLISH HERE]
<END OF SCORE 4 DESCRIPTION>

<START OF SCORE 5 DESCRIPTION>
[INSERT SCORE 5 DESCRIPTION IN ENGLISH HERE]
<END OF SCORE 5 DESCRIPTION>
```

Ensure that your response is comprehensive, coherent, and follows all the instructions provided above.
IMPORTANT: ABIDE STRICTLY BY THE REQUESTED FORMAT AND KEEP GENERATING UNTIL THE END OF THE REQUESTED OUTPUT.

Figure 12: Prompt for generating MT Eval source texts and references part 2.

Generate an example translation of score {N} for the given translation instruction, source, and scoring rubrics:
```
<START OF SOURCE>
${source}
<END OF SOURCE>

<START OF TRANSLATION INSTRUCTION>
${translation_instruction}
<END OF TRANSLATION INSTRUCTION>

<START OF SCORING RUBRICS>
${scoring_rubrics}
<END OF SCORING RUBRICS>

<START OF SCORE {N} TRANSLATION>
[INSERT TRANSLATION HERE]
<END OF SCORE {N} TRANSLATION>
```

IMPORTANT: ABIDE STRICTLY BY THE REQUESTED FORMAT AND KEEP GENERATING UNTIL THE END OF THE REQUESTED OUTPUT.

Figure 13: Prompt for generating MT Eval score example.

### A.3.2 Data Generation Prompt Variable Counts

We list the number of training instances for each value of each variable in our MT evaluation data generation prompt.

**Topic and Subtopic.** **Gaming & Software**: 296 (Virtual Reality: 48, Software Development: 56, Mobile Games: 16, Cloud Gaming: 32, Game Development: 32, Gaming Communities: 64, Gaming Hardware: 48); **Sports Industry**: 288 (Sports Management: 16, Athletic Training: 56, Athletic Equipment: 64, Sports Technology: 32, Sports Medicine: 48, E-sports: 24, Professional Leagues: 48); **Financial Services**: 280 (Digital Banking: 40, Insurance: 88, Wealth Management: 40, Payment Systems: 32, Financial Technology: 32, Risk Management:

16, Investment Management: 32); **Mumbai**: 272 (Cultural Heritage: 40, Entertainment: 40, Fashion: 40, Film Industry: 48, Business Center: 48, Food Culture: 40, Urban Development: 16); **China**: 264 (Cultural Heritage: 24, Business Culture: 56, Urban Development: 48, Technology Industry: 48, Traditional Customs: 32, Food Culture: 40, Innovation Hub: 16); **Music Industry**: 264 (Music Technology: 40, Music Production: 40, Industry Trends: 32, Live Events: 32, Music Publishing: 48, Digital Distribution: 24, Artist Management: 48); **Food & Agriculture**: 256 (Food Technology: 32, Food Safety: 56, Urban Farming: 16, Agricultural Trade: 32, Agricultural Policy: 40, Organic Production: 40, Sustainable Farming: 40); **Manufacturing & Safety**: 256 (Production Processes: 24, Workplace Standards: 40, Equipment Safety: 24, Safety Regulations: 32, Risk Assessment: 48, Industrial Safety: 24, Quality Control: 64); **Brazil**: 248 (Cultural Festivals: 24, Urban Life: 24, Business Environment: 48, Tourism Industry: 48, Food & Cuisine: 56, Music Scene: 16, Sports Culture: 32); **Fitness & Wellness**: 248 (Nutrition: 32, Mental Health: 48, Health Tracking: 24, Exercise Programs: 24, Wellness Technology: 40, Wellness Education: 56, Personal Training: 24); **Architecture & Design**: 248 (Sustainable Design: 16, Digital Architecture: 40, Interior Design: 24, Design Innovation: 48, Building Technology: 24, Architectural Heritage: 56, Urban Architecture: 40); **India**: 232 (Culinary Traditions: 24, Cultural Diversity: 40, Technology Sector: 40, Festival Culture: 48, Business Hub: 32, Film Industry: 32, Traditional Arts: 16); **Social Media**: 232 (Social Commerce: 24, User Engagement: 64, Influencer Marketing: 32, Social Analytics: 32, Digital Communities: 32, Content Creation: 24, Platform Development: 24); **Seoul**: 232 (Fashion Trends: 24, Tech Industry: 24, Urban Innovation: 40, Food Scene: 48, Business Hub: 24, Pop Culture: 24, Entertainment: 48); **Books & Literature**: 224 (Publishing Industry: 24, Book Marketing: 40, Author Platform: 16, Digital Publishing: 72, Literary Events: 32, Reading Technology: 40); **São Paulo**: 224 (Sports Culture: 32, Business Hub: 24, Cultural Scene: 48, Urban Life: 24, Entertainment: 48, Food & Dining: 32, Fashion Industry: 16); **Spain**: 224 (Tourism Industry: 40, Sports Culture: 16, Cultural Traditions: 32, Business Environment: 56, Urban Life: 24, Culinary Arts: 56); **Portugal**: 224 (Cultural Heritage: 40, Business Innovation: 40, Food & Wine: 32, Arts Scene: 24, Urban Development: 16, Tourism Industry: 48, Maritime Culture: 24); **Tokyo**: 224 (Entertainment Districts: 32, Cuisine: 48, Fashion: 16, Traditional Culture: 32, Urban Innovation: 16, Technology Industry: 56, Pop Culture: 24); **Workplace Transformation**: 224 (Office Technology: 48, HR Innovation: 48, Workplace Safety: 24, Remote Work: 16, Corporate Culture: 16, Professional Development: 24, Employee Wellness: 48); **Dubai**: 224 (Cultural Traditions: 48, Luxury Lifestyle: 40, International Trade: 40, Tourism Industry: 32, Business Center: 32, Urban Development: 24, Technology Innovation: 8); **Berlin**: 224 (Startup Scene: 24, Alternative Culture: 40, Cultural History: 24, Tech Industry: 40, Art Community: 40, Nightlife: 24, Urban Planning: 32); **Poetry**: 224 (Haiku: 24, Asian Poetry: 32, Theme identification: 32, Modernism: 88, Contemporary: 16, European Poetry: 32); **London**: 224 (Food Scene: 56, Theatre & Arts: 40, Urban Transport: 56, Financial Services: 32, Royal Traditions: 24, Cultural Heritage: 16); **Lisbon**: 216 (Urban Innovation: 24, Maritime Heritage: 48, Tourism Industry: 48, Arts Scene: 32, Startup Ecosystem: 8, Food & Wine: 40, Cultural History: 16); **New York City**: 216 (Entertainment: 24, Tourism: 32, Urban Development: 56, Sports Teams: 24, Business & Finance: 32, Arts & Culture: 24, Food & Dining: 24); **Global Markets**: 216 (Stock Exchanges: 40, Foreign Investment: 56, Emerging Markets: 40, International Trade: 32, Foreign Exchange: 16, Market Regulations: 24, Commodity Markets: 8); **Weather & Climate**: 216 (Climate Technology: 40, Climate Change: 24, Climate Science: 16, Environmental Impact: 72, Weather Forecasting: 24, Atmospheric Research: 16, Weather Systems: 24); **Urban Development**: 208 (Infrastructure: 48, Smart Cities: 32, Green Spaces: 16, Public Transportation: 16, Urban Planning: 56, Housing Projects: 32, Sustainable Development: 8); **Arts & Culture**: 208 (Art Market: 32, Performance Art: 40, Art Education: 24, Cultural Events: 40, Cultural Heritage: 32, Visual Arts: 32, Digital Art: 8); **Germany**: 200 (Technology Sector: 72, Sports Culture: 48, Education System: 16, Automotive Industry: 24, Cultural Traditions: 8, Business Innovation: 16, Urban Development: 16); **Japan**: 200 (Business Practices: 32, Traditional Culture: 32, Arts & Crafts: 32, Technology Industry: 48, Popular Culture: 24, Social Customs: 24, Food & Cuisine: 8); **Insurance & Risk Management**: 200 (Risk Assessment: 32, Underwriting: 40, Claims Processing: 8, Risk Mitigation: 32, Insurance Technology: 32, Regulatory Compliance: 48, Insurance Products: 8); **Italy**: 200 (Fashion Industry: 48, Design Industry: 24, Arts Scene: 32, Business Culture: 32, Cultural Heritage: 32, Food & Wine: 32); **Pharmaceutical Industry**: 200 (Manufacturing: 40, Patient Safety: 32,

Drug Development: 32, Clinical Trials: 32, Regulatory Approval: 24, Market Access: 40); **Beauty & Cosmetics**: 200 (Makeup Products: 40, Beauty Technology: 32, Sustainability: 8, Product Development: 40, Natural Cosmetics: 56, Skincare: 8, Marketing: 16); **International Relations**: 200 (Global Security: 72, Trade Agreements: 32, Cultural Exchange: 24, Regional Alliances: 40, Diplomatic Missions: 16, International Aid: 16); **Medical & Healthcare**: 200 (Healthcare IT: 16, Medical Insurance: 24, Pharmaceutical Research: 40, Patient Care: 48, Clinical Trials: 40, Telemedicine: 16, Medical Devices: 16); **Automotive Industry**: 200 (Safety Systems: 40, Auto Design: 16, Autonomous Technology: 56, Vehicle Manufacturing: 24, Market Trends: 32, Car Technology: 24, Electric Vehicles: 8); **Environmental Policy**: 200 (Climate Agreements: 32, Marine Conservation: 32, Carbon Trading: 24, Renewable Energy Initiatives: 16, Urban Planning: 56, Waste Management: 16, Wildlife Protection: 24); **Marketing & Advertising**: 200 (Advertising Technology: 32, Content Marketing: 56, Social Media Marketing: 24, Market Research: 16, Digital Marketing: 32, Brand Strategy: 32, Campaign Management: 8); **France**: 192 (Arts & Literature: 32, Business Culture: 24, Wine Industry: 24, Tourism: 32, Cultural Heritage: 32, Culinary Arts: 24, Fashion Industry: 24); **Home & Living**: 192 (Smart Home: 40, Furniture: 56, Sustainable Living: 8, Home Improvement: 40, Decorative Arts: 16, Interior Design: 16, Home Technology: 16); **Paris**: 192 (Culinary Arts: 24, Art Scene: 48, Luxury Brands: 72, Cultural Landmarks: 16, Fashion Industry: 16, Urban Life: 8, Tourism Industry: 8); **Parenting & Family**: 192 (Family Dynamics: 40, Education: 32, Family Health: 40, Parenting Resources: 32, Child Safety: 16, Child Development: 32); **Patents & Intellectual Property**: 192 (Patent Applications: 32, Trade Secrets: 40, IP Litigation: 32, Trademark Registration: 24, International Patents: 40, Copyright Protection: 16, IP Strategy: 8); **Amsterdam**: 192 (Art Scene: 40, Business Innovation: 32, Cycling Culture: 16, Tourism: 56, Tech Industry: 24, Urban Planning: 16, Cultural Heritage: 8); **Economic Policy**: 184 (Trade Regulations: 40, Fiscal Measures: 16, Economic Stimulus: 40, Employment Policy: 32, Tax Reform: 16, Banking Regulations: 24, Monetary Policy: 16); **Public Health**: 184 (Vaccination Programs: 32, Epidemiology: 32, Mental Health Services: 24, Healthcare Systems: 48, Disease Prevention: 24, Health Technology: 16, Maternal Health: 8); **Tech Innovation**: 184 (Green Tech: 32, Cybersecurity: 40, Quantum Computing: 48, Robotics: 16, Biotechnology: 24, Edge Computing: 8, Artificial Intelligence: 16); **Singapore**: 184 (Cultural Diversity: 24, Urban Planning: 56, Education: 24, Food Culture: 24, Business Innovation: 16, Financial Hub: 24, Smart City Initiatives: 16); **Film & Cinema**: 184 (Film Marketing: 32, Digital Effects: 8, Film Industry: 48, Film Technology: 24, Distribution: 32, Cinema Innovation: 24, Film Production: 16); **Cultural Trends**: 184 (Fashion Movements: 8, Entertainment Trends: 48, Digital Culture: 24, Social Media Influence: 24, Pop Culture: 16, Art Movements: 40, Cultural Festivals: 24); **Religious & Cultural Studies**: 184 (Sacred Texts: 56, Interfaith Dialogue: 16, Cultural Anthropology: 40, Religious Education: 24, Religious Practices: 24, Religious Traditions: 16, Cultural Heritage: 8); **Politics & Governance**: 184 (Political Communication: 24, Government Innovation: 16, Electoral Processes: 24, Political Systems: 24, Public Policy: 16, Governance Reform: 48, Civic Technology: 32); **Consumer Electronics**: 184 (Mobile Devices: 16, Audio Equipment: 24, Display Technology: 24, Gaming Hardware: 32, Smart Home: 24, Personal Computing: 48, Wearable Technology: 16); **Madrid**: 176 (Business Hub: 40, Tourism: 24, Sports Culture: 16, Food & Wine: 16, Arts Scene: 40, Urban Life: 32, Cultural Heritage: 8); **NGOs & Nonprofits**: 176 (International Development: 24, Social Innovation: 32, Fundraising: 32, Community Development: 48, Humanitarian Aid: 8, Social Impact: 32); **Wildlife & Nature**: 176 (Environmental Protection: 32, Conservation: 16, Species Preservation: 16, Wildlife Research: 56, Biodiversity: 24, Natural Habitats: 16, Ecosystem Management: 16); **E-commerce & Retail**: 176 (Customer Experience: 16, Online Marketplaces: 48, Mobile Commerce: 24, Retail Technology: 24, Digital Payment: 32, Supply Chain: 24, Retail Analytics: 8); **Stockholm**: 176 (Business Hub: 40, Cultural Scene: 48, Urban Planning: 24, Food & Lifestyle: 32, Sustainability: 16, Design Culture: 8, Tech Innovation: 8); **Legal & Compliance**: 168 (Legal Technology: 40, International Law: 24, Regulatory Compliance: 24, Corporate Law: 24, Consumer Rights: 16, Intellectual Property: 24, Data Protection: 16); **Dating & Relationships**: 168 (Personal Growth: 16, Relationship Psychology: 40, Relationship Counseling: 48, Dating Culture: 8, Online Dating: 24, Social Connection: 16, Dating Apps: 16); **Academic Research**: 168 (Peer Review: 48, Scientific Publications: 16, Research Ethics: 40, Academic Collaboration: 8, Data Analysis: 40, Research Methodology: 16); **Media & Entertainment**: 168 (Digital Media: 32, Content Creation: 48, Streaming Services: 16, Publishing: 16, Film Production:

24, Broadcasting: 8, Gaming Industry: 24); **Telecommunications**: 168 (Communication Services: 16, Industry Standards: 32, Digital Networks: 24, Telecom Innovation: 40, Mobile Technology: 32, Wireless Technology: 24); **Scientific Discoveries**: 168 (Marine Biology: 32, Genetic Research: 32, Physics Advances: 40, Archaeological Finds: 8, Space Exploration: 24, Climate Science: 24, Medical Breakthroughs: 8); **Tourism & Hospitality**: 160 (Hotel Management: 32, Eco-Tourism: 16, Tourism Marketing: 24, Event Planning: 16, Travel Services: 16, Cultural Tourism: 24, Customer Service: 32); **Education Reform**: 160 (Digital Learning: 24, Curriculum Changes: 16, STEM Initiatives: 24, Assessment Methods: 16, Higher Education: 32, Teacher Training: 32, Special Education: 16); **Space Exploration**: 160 (Space Technology: 40, Space Industry: 32, Astronomical Discovery: 32, Space Policy: 16, Satellite Systems: 8, Space Research: 16, Space Travel: 16); **Fashion & Apparel**: 152 (Textile Industry: 24, Fashion Technology: 48, Sustainable Fashion: 24, Retail Fashion: 24, Luxury Brands: 24, Fashion Design: 8); **Real Estate**: 144 (Investment: 24, Sustainable Building: 32, Property Management: 8, Real Estate Technology: 24, Commercial Real Estate: 32, Market Analysis: 24); **Mental Health**: 144 (Mental Health Technology: 48, Mental Health Education: 40, Support Programs: 24, Therapy Services: 24, Youth Mental Health: 8); **Transportation & Mobility**: 144 (Aviation: 8, Electric Vehicles: 32, Autonomous Driving: 24, Maritime Transport: 56, Ride Sharing: 8, Public Transit: 16); **Government Documentation**: 144 (Regulatory Guidelines: 32, Administrative Procedures: 32, Official Forms: 40, Policy Documents: 8, Legislative Documents: 32); **Food & Cuisine**: 136 (Food Technology: 16, Culinary Arts: 16, Food Innovation: 40, Dietary Trends: 40, Culinary Education: 8, Food Culture: 16); **Sydney**: 136 (Tourism: 32, Urban Development: 16, Sports Events: 32, Business District: 8, Lifestyle & Culture: 24, Food Scene: 16, Entertainment: 8); **Sports & Recreation**: 136 (Fitness Training: 24, Sports Technology: 16, Sports Management: 16, Equipment Innovation: 32, Recreational Activities: 16, Sports Medicine: 16, Professional Sports: 16); **Renewable Energy**: 120 (Sustainable Development: 8, Solar Power: 48, Energy Storage: 16, Clean Energy Innovation: 16, Green Technology: 8, Energy Policy: 8, Wind Energy: 16); **History & Heritage**: 120 (Archaeological Studies: 32, Cultural Preservation: 32, Heritage Conservation: 8, Cultural Memory: 24, Digital Archives: 16, Historical Research: 8); **United Kingdom**: 112 (Arts & Entertainment: 8, Sports Culture: 16, Business Innovation: 16, Financial Services: 16, Education System: 40, Urban Life: 8, Cultural Heritage: 8).

**Style.** **journalistic**: 1024; **creative**: 1016; **analytical**: 968; **formal**: 960; **poetic**: 960; **minimalist**: 952; **humorous**: 936; **academic**: 912; **elaborate**: 912; **narrative**: 880; **rushed**: 864; **technical**: 840; **neutral**: 832; **informal**: 824; **descriptive**: 792; **casual**: 792; **concise**: 776; **persuasive**: 760.

**Audience.** **seniors**: 1616; **parents**: 1488; **college students**: 1408; **experts**: 1408; **general public**: 1400; **professionals**: 1352; **teenagers**: 1304; **middle-aged adults**: 1288; **educators**: 1256; **beginners**: 1216; **children**: 1160; **young adults**: 1104.

**Source Length.** **short**: 4168; **very long**: 4112; **long**: 3912; **medium**: 3808.

## B  Training Hyperparameters

We train all our models on 1 epoch of our training dataset, with a cosine learning rate scheduler with a warmup of 10% the total steps, and initial learning rate of $1 \times 10^{-6}$ decaying to 0. We use a batch size of 32 with sequences of up to 4096 tokens.

# C   Results by Language, Category, and Language Pair

## C.1   MM-Eval

| Model | Chat | MMEval (English) Language Hallucinations | Linguistics | Reasoning | Safety |
|---|---|---|---|---|---|
| **Proprietary Models** | | | | | |
| GPT-4O | 0.7216 | - | 0.9800 | 0.7913 | 0.5761 |
| **Small (3B parameters)** | | | | | |
| Qwen2.5-3B-Instruct | 0.6598 | - | 0.6000 | 0.5913 | 0.5978 |
| Glider 3B † | 0.6495 | - | 0.6000 | 0.6174 | 0.7174 |
| M-PROMETHEUS 3B * | 0.5876 | - | 0.6800 | 0.5652 | 0.8696 |
| **Medium (7B parameters)** | | | | | |
| Qwen2.5-7B-Instruct | 0.6907 | - | 0.6800 | 0.6783 | 0.8478 |
| PROMETHEUS 2 7B † | 0.6340 | - | 0.6667 | 0.4826 | 0.7554 |
| Hercule 7B * | 0.6289 | - | 0.4933 | 0.5000 | 0.2446 |
| M-PROMETHEUS 7B * | 0.6804 | - | 0.7600 | 0.6087 | 0.8261 |
| **Large (14B+ parameters)** | | | | | |
| Qwen2.5-14B-Instruct | 0.7423 | - | 0.8933 | 0.7217 | 0.5652 |
| PROMETHEUS 2 8x7B † | 0.6495 | - | 0.6467 | 0.5478 | 0.8804 |
| M-PROMETHEUS 14B * | 0.6186 | - | 0.8800 | 0.6435 | 0.9022 |
| **Backbone Model Ablations** | | | | | |
| Mistral-7B-v0.2-Instruct | 0.6392 | - | 0.6267 | 0.4783 | 0.5543 |
| EuroLLM-9B-Instruct | 0.6289 | - | 0.6667 | 0.5826 | 0.9022 |
| Aya-Expanse-8B | 0.6598 | - | 0.6933 | 0.5957 | 0.6957 |
| Qwen2.5-7B-Instruct | 0.7010 | - | 0.7067 | 0.6000 | 0.6848 |
| **Training Data Ablations** | | | | | |
| MT Eval Data | 0.6082 | - | 0.7733 | 0.6174 | 0.7717 |
| Translated Data | | | | | |
|   3 Non-English Langs | 0.6082 | - | 0.8133 | 0.5826 | 0.6957 |
| Multilingual Data | | | | | |
|   3 Non-English Langs | 0.5979 | - | 0.7200 | 0.5652 | 0.8152 |
|   5 Non-English Langs | 0.6186 | - | 0.6933 | 0.5565 | 0.8478 |

Table 3: Accuracy on MMEval (English) broken down by category.

| Model | MMEval (German) | | | | | |
| | Chat | Language Hallucinations | Linguistics | Reasoning | Safety |
|---|---|---|---|---|---|---|
| **Proprietary Models** | | | | | | |
| GPT-4O | 0.7069 | - | | 0.6933 | 0.7966 | - |
| **Small (3B parameters)** | | | | | | |
| Qwen2.5-3B-Instruct | 0.6724 | - | | 0.6400 | 0.5876 | - |
| Glider 3B † | 0.6379 | - | | 0.5200 | 0.5819 | - |
| M-PROMETHEUS 3B * | 0.5517 | - | | 0.6267 | 0.7232 | - |
| **Medium (7B parameters)** | | | | | | |
| Qwen2.5-7B-Instruct | 0.7586 | - | | 0.6800 | 0.6554 | - |
| PROMETHEUS 2 7B † | 0.7069 | - | | 0.6800 | 0.5819 | - |
| Hercule 7B * | 0.7155 | - | | 0.6133 | 0.5678 | - |
| M-PROMETHEUS 7B * | 0.6552 | - | | 0.5467 | 0.6271 | - |
| **Large (14B+ parameters)** | | | | | | |
| Qwen2.5-14B-Instruct | 0.6897 | - | | 0.6667 | 0.7288 | - |
| PROMETHEUS 2 8x7B † | 0.6897 | - | | 0.6133 | 0.6554 | - |
| M-PROMETHEUS 14B * | 0.7241 | - | | 0.6133 | 0.7006 | - |
| **Backbone Model Ablations** | | | | | | |
| Mistral-7B-v0.2-Instruct | 0.6207 | - | | 0.4133 | 0.5480 | - |
| EuroLLM-9B-Instruct | 0.5862 | - | | 0.6667 | 0.6130 | - |
| Aya-Expanse-8B | 0.6724 | - | | 0.6533 | 0.6356 | - |
| Qwen2.5-7B-Instruct | 0.6207 | - | | 0.6400 | 0.6215 | - |
| **Training Data Ablations** | | | | | | |
| MT Eval Data | 0.6724 | - | | 0.6267 | 0.6610 | - |
| Translated Data | | | | | | |
|   3 Non-English Langs | 0.6379 | - | | 0.5733 | 0.6695 | - |
| Multilingual Data | | | | | | |
|   3 Non-English Langs | 0.6379 | - | | 0.4933 | 0.6215 | - |
|   5 Non-English Langs | 0.6034 | - | | 0.5733 | 0.5395 | - |

Table 4: Accuracy on MMEval (German) broken down by category.

| | MMEval (French) | | | | | |
|---|---|---|---|---|---|---|
| **Model** | Chat | Language | Hallucinations | Linguistics | Reasoning | Safety |
| **Proprietary Models** | | | | | | |
| GPT-4O | 0.7333 | | - | - | 0.7847 | - |
| **Small (3B parameters)** | | | | | | |
| Qwen2.5-3B-Instruct | 0.6000 | | - | - | 0.6181 | - |
| Glider 3B † | 0.6889 | | - | - | 0.6771 | - |
| M-PROMETHEUS 3B * | 0.7111 | | - | - | 0.6875 | - |
| **Medium (7B parameters)** | | | | | | |
| Qwen2.5-7B-Instruct | 0.7111 | | - | - | 0.6319 | - |
| PROMETHEUS 2 7B † | 0.5000 | | - | - | 0.5347 | - |
| Hercule 7B * | 0.6111 | | - | - | 0.5556 | - |
| M-PROMETHEUS 7B * | 0.6667 | | - | - | 0.6319 | - |
| **Large (14B+ parameters)** | | | | | | |
| Qwen2.5-14B-Instruct | 0.7556 | | - | - | 0.7222 | - |
| PROMETHEUS 2 8x7B † | 0.7222 | | - | - | 0.5833 | - |
| M-PROMETHEUS 14B * | 0.6444 | | - | - | 0.7014 | - |
| **Backbone Model Ablations** | | | | | | |
| Mistral-7B-v0.2-Instruct | 0.5556 | | - | - | 0.5903 | - |
| EuroLLM-9B-Instruct | 0.6222 | | - | - | 0.6007 | - |
| Aya-Expanse-8B | 0.6222 | | - | - | 0.6458 | - |
| Qwen2.5-7B-Instruct | 0.6222 | | - | - | 0.6910 | - |
| **Training Data Ablations** | | | | | | |
| MT Eval Data | 0.6000 | | - | - | 0.6875 | - |
| Translated Data | | | | | | |
|   3 Non-English Langs | 0.6000 | | - | - | 0.7257 | - |
| Multilingual Data | | | | | | |
|   3 Non-English Langs | 0.6222 | | - | - | 0.6007 | - |
|   5 Non-English Langs | 0.5778 | | - | - | 0.5521 | - |

Table 5: Accuracy on MMEval (French) broken down by category.

| Model | MMEval (Spanish) | | | | | |
|---|---|---|---|---|---|---|
| | Chat | Language Hallucinations | Linguistics | Reasoning | Safety |
| **Proprietary Models** | | | | | | |
| GPT-4O | 0.6413 | 0.7240 | 0.9533 | 0.7928 | - |
| **Small (3B parameters)** | | | | | | |
| Qwen2.5-3B-Instruct | 0.6630 | 0.5729 | 0.5600 | 0.6184 | - |
| Glider 3B † | 0.5870 | 0.6432 | 0.5600 | 0.6020 | - |
| M-PROMETHEUS 3B * | 0.6087 | 0.5625 | 0.6400 | 0.6053 | - |
| **Medium (7B parameters)** | | | | | | |
| Qwen2.5-7B-Instruct | 0.7826 | 0.5677 | 0.5867 | 0.6513 | - |
| PROMETHEUS 2 7B † | 0.6522 | 0.5625 | 0.6667 | 0.5362 | - |
| Hercule 7B * | 0.5543 | 0.5208 | 0.5067 | 0.5559 | - |
| M-PROMETHEUS 7B * | 0.5978 | 0.7135 | 0.6267 | 0.6053 | - |
| **Large (14B+ parameters)** | | | | | | |
| Qwen2.5-14B-Instruct | 0.7174 | 0.8333 | 0.7867 | 0.7368 | - |
| PROMETHEUS 2 8x7B † | 0.6522 | 0.7057 | 0.6933 | 0.5888 | - |
| M-PROMETHEUS 14B * | 0.6413 | 0.8646 | 0.6667 | 0.6908 | - |
| **Backbone Model Ablations** | | | | | | |
| Mistral-7B-v0.2-Instruct | 0.5978 | 0.4818 | 0.6533 | 0.5395 | - |
| EuroLLM-9B-Instruct | 0.5435 | 0.5885 | 0.5867 | 0.5691 | - |
| Aya-Expanse-8B | 0.5978 | 0.5938 | 0.5200 | 0.5658 | - |
| Qwen2.5-7B-Instruct | 0.6848 | 0.6771 | 0.6400 | 0.6546 | - |
| **Training Data Ablations** | | | | | | |
| MT Eval Data | 0.6196 | 0.7135 | 0.6400 | 0.6513 | - |
| Translated Data | | | | | |
| 3 Non-English Langs | 0.6522 | 0.6094 | 0.6133 | 0.6711 | - |
| Multilingual Data | | | | | |
| 3 Non-English Langs | 0.5435 | 0.6198 | 0.6267 | 0.6184 | - |
| 5 Non-English Langs | 0.6413 | 0.6250 | 0.6533 | 0.5066 | - |

Table 6: Accuracy on MMEval (Spanish) broken down by category.

| Model | MMEval (Catalan) | | | | |
|---|---|---|---|---|---|
| | Chat | Language Hallucinations | Linguistics | Reasoning | Safety |
| **Proprietary Models** | | | | | |
| GPT-4O | 0.7500 | 0.7423 | 0.9267 | - | - |
| **Small (3B parameters)** | | | | | |
| Qwen2.5-3B-Instruct | 0.7250 | 0.4691 | 0.6867 | - | - |
| Glider 3B † | 0.6875 | 0.5928 | 0.5933 | - | - |
| M-PROMETHEUS 3B * | 0.6000 | 0.5155 | 0.6533 | - | - |
| **Medium (7B parameters)** | | | | | |
| Qwen2.5-7B-Instruct | 0.7250 | 0.5206 | 0.6133 | - | - |
| PROMETHEUS 2 7B † | 0.6500 | 0.4433 | 0.6933 | - | - |
| Hercule 7B * | 0.6250 | 0.5258 | 0.4733 | - | - |
| M-PROMETHEUS 7B * | 0.6250 | 0.5361 | 0.6400 | - | - |
| **Large (14B+ parameters)** | | | | | |
| Qwen2.5-14B-Instruct | 0.7500 | 0.6289 | 0.8133 | - | - |
| PROMETHEUS 2 8x7B † | 0.7750 | 0.7062 | 0.7467 | - | - |
| M-PROMETHEUS 14B * | 0.5750 | 0.6392 | 0.7333 | - | - |
| **Backbone Model Ablations** | | | | | |
| Mistral-7B-v0.2-Instruct | 0.6000 | 0.6186 | 0.5733 | - | - |
| EuroLLM-9B-Instruct | 0.6500 | 0.5670 | 0.6133 | - | - |
| Aya-Expanse-8B | 0.6750 | 0.5258 | 0.6533 | - | - |
| Qwen2.5-7B-Instruct | 0.5750 | 0.5464 | 0.6133 | - | - |
| **Training Data Ablations** | | | | | |
| MT Eval Data | 0.6750 | 0.5876 | 0.6133 | - | - |
| Translated Data | | | | | |
|   3 Non-English Langs | 0.6500 | 0.5258 | 0.6133 | - | - |
| Multilingual Data | | | | | |
|   3 Non-English Langs | 0.4500 | 0.5052 | 0.6533 | - | - |
|   5 Non-English Langs | 0.5000 | 0.5052 | 0.6667 | - | - |

Table 7: Accuracy on MMEval (Catalan) broken down by category.

| Model | Chat | MMEval (Russian) Language Hallucinations | Linguistics | Reasoning | Safety |
|---|---|---|---|---|---|
| **Proprietary Models** | | | | | |
| GPT-4O | 0.7887 | - | - | 0.7640 | - |
| **Small (3B parameters)** | | | | | |
| Qwen2.5-3B-Instruct | 0.6901 | - | - | 0.5901 | - |
| Glider 3B † | 0.6479 | - | - | 0.6491 | - |
| M-PROMETHEUS 3B * | 0.6197 | - | - | 0.5776 | - |
| **Medium (7B parameters)** | | | | | |
| Qwen2.5-7B-Instruct | 0.6620 | - | - | 0.6646 | - |
| PROMETHEUS 2 7B † | 0.6338 | - | - | 0.5311 | - |
| Hercule 7B * | 0.6408 | - | - | 0.5342 | - |
| M-PROMETHEUS 7B * | 0.5634 | - | - | 0.6273 | - |
| **Large (14B+ parameters)** | | | | | |
| Qwen2.5-14B-Instruct | 0.7042 | - | - | 0.7143 | - |
| PROMETHEUS 2 8x7B † | 0.6408 | - | - | 0.5870 | - |
| M-PROMETHEUS 14B * | 0.7183 | - | - | 0.6957 | - |
| **Backbone Model Ablations** | | | | | |
| Mistral-7B-v0.2-Instruct | 0.5915 | - | - | 0.5807 | - |
| EuroLLM-9B-Instruct | 0.6761 | - | - | 0.5683 | - |
| Aya-Expanse-8B | 0.6197 | - | - | 0.5776 | - |
| Qwen2.5-7B-Instruct | 0.6761 | - | - | 0.6491 | - |
| **Training Data Ablations** | | | | | |
| MT Eval Data | 0.7183 | - | - | 0.6677 | - |
| Translated Data | | | | | |
|   3 Non-English Langs | 0.6479 | - | - | 0.6304 | - |
| Multilingual Data | | | | | |
|   3 Non-English Langs | 0.5493 | - | - | 0.6056 | - |
|   5 Non-English Langs | 0.6479 | - | - | 0.5559 | - |

Table 8: Accuracy on MMEval (Russian) broken down by category.

| Model | Chat | Language | Hallucinations | Linguistics | Reasoning | Safety |
|---|---|---|---|---|---|---|
| | | | **MMEval (Chinese)** | | | |
| **Proprietary Models** | | | | | | |
| GPT-4O | 0.8049 | | - | - | 0.7881 | - |
| **Small (3B parameters)** | | | | | | |
| Qwen2.5-3B-Instruct | 0.8293 | | - | - | 0.6623 | - |
| Glider 3B † | 0.6463 | | - | - | 0.5762 | - |
| M-PROMETHEUS 3B * | 0.7927 | | - | - | 0.6755 | - |
| **Medium (7B parameters)** | | | | | | |
| Qwen2.5-7B-Instruct | 0.8537 | | - | - | 0.7152 | - |
| PROMETHEUS 2 7B † | 0.6341 | | - | - | 0.5298 | - |
| Hercule 7B * | 0.6585 | | - | - | 0.5762 | - |
| M-PROMETHEUS 7B * | 0.5854 | | - | - | 0.6954 | - |
| **Large (14B+ parameters)** | | | | | | |
| Qwen2.5-14B-Instruct | 0.8780 | | - | - | 0.7583 | - |
| PROMETHEUS 2 8x7B † | 0.8049 | | - | - | 0.6424 | - |
| M-PROMETHEUS 14B * | 0.7317 | | - | - | 0.7285 | - |
| **Backbone Model Ablations** | | | | | | |
| Mistral-7B-v0.2-Instruct | 0.7317 | | - | - | 0.5199 | - |
| EuroLLM-9B-Instruct | 0.6463 | | - | - | 0.5960 | - |
| Aya-Expanse-8B | 0.7073 | | - | - | 0.6623 | - |
| Qwen2.5-7B-Instruct | 0.6098 | | - | - | 0.5861 | - |
| **Training Data Ablations** | | | | | | |
| MT Eval Data | 0.6585 | | - | - | 0.5629 | - |
| Translated Data | | | | | | |
|   3 Non-English Langs | 0.6585 | | - | - | 0.6093 | - |
| Multilingual Data | | | | | | |
|   3 Non-English Langs | 0.5854 | | - | - | 0.5762 | - |
|   5 Non-English Langs | 0.5122 | | - | - | 0.5695 | - |

Table 9: Accuracy on MMEval (Chinese) broken down by category.

| Model | Chat | MMEval (Arabic) Language Hallucinations | Linguistics | Reasoning | Safety |
|---|---|---|---|---|---|
| **Proprietary Models** | | | | | |
| GPT-4O | - | 0.7688 | - | - | 0.5326 |
| **Small (3B parameters)** | | | | | |
| Qwen2.5-3B-Instruct | - | 0.5134 | - | - | 0.4620 |
| Glider 3B † | - | 0.5108 | - | - | 0.4565 |
| M-PROMETHEUS 3B * | - | 0.4624 | - | - | 0.6630 |
| **Medium (7B parameters)** | | | | | |
| Qwen2.5-7B-Instruct | - | 0.4892 | - | - | 0.7609 |
| PROMETHEUS 2 7B † | - | 0.5430 | - | - | 0.7663 |
| Hercule 7B * | - | 0.5054 | - | - | 0.2772 |
| M-PROMETHEUS 7B * | - | 0.5538 | - | - | 0.9565 |
| **Large (14B+ parameters)** | | | | | |
| Qwen2.5-14B-Instruct | - | 0.7608 | - | - | 0.4348 |
| PROMETHEUS 2 8x7B † | - | 0.5484 | - | - | 0.5761 |
| M-PROMETHEUS 14B * | - | 0.8280 | - | - | 0.9348 |
| **Backbone Model Ablations** | | | | | |
| Mistral-7B-v0.2-Instruct | - | 0.5968 | - | - | 0.3261 |
| EuroLLM-9B-Instruct | - | 0.5242 | - | - | 0.7391 |
| Aya-Expanse-8B | - | 0.5188 | - | - | 0.6196 |
| Qwen2.5-7B-Instruct | - | 0.6828 | - | - | 0.5978 |
| **Training Data Ablations** | | | | | |
| MT Eval Data | - | 0.7043 | - | - | 0.7065 |
| Translated Data | | | | | |
|   3 Non-English Langs | - | 0.5860 | - | - | 0.5217 |
| Multilingual Data | | | | | |
|   3 Non-English Langs | - | 0.5753 | - | - | 0.8261 |
|   5 Non-English Langs | - | 0.6667 | - | - | 0.8913 |

Table 10: Accuracy on MMEval (Arabic) broken down by category.

| Model | Chat | Language Hallucinations | Linguistics | Reasoning | Safety |
|---|---|---|---|---|---|
| | | **MMEval (Bengali)** | | | |
| **Proprietary Models** | | | | | |
| GPT-4O | - | 0.6298 | - | 0.8455 | - |
| **Small (3B parameters)** | | | | | |
| Qwen2.5-3B-Instruct | - | 0.5525 | - | 0.6682 | - |
| Glider 3B † | - | 0.5525 | - | 0.5705 | - |
| M-PROMETHEUS 3B * | - | 0.5442 | - | 0.6182 | - |
| **Medium (7B parameters)** | | | | | |
| Qwen2.5-7B-Instruct | - | 0.5249 | - | 0.6000 | - |
| PROMETHEUS 2 7B † | - | 0.4641 | - | 0.5523 | - |
| Hercule 7B * | - | 0.5166 | - | 0.5705 | - |
| M-PROMETHEUS 7B * | - | 0.6381 | - | 0.6182 | - |
| **Large (14B+ parameters)** | | | | | |
| Qwen2.5-14B-Instruct | - | 0.6906 | - | 0.7273 | - |
| PROMETHEUS 2 8x7B † | - | 0.4972 | - | 0.5727 | - |
| M-PROMETHEUS 14B * | - | 0.8011 | - | 0.7045 | - |
| **Backbone Model Ablations** | | | | | |
| Mistral-7B-v0.2-Instruct | - | 0.6851 | - | 0.5477 | - |
| EuroLLM-9B-Instruct | - | 0.5691 | - | 0.5977 | - |
| Aya-Expanse-8B | - | 0.5746 | - | 0.5159 | - |
| Qwen2.5-7B-Instruct | - | 0.7707 | - | 0.5932 | - |
| **Training Data Ablations** | | | | | |
| MT Eval Data | - | 0.8895 | - | 0.5909 | - |
| Translated Data | | | | | |
| 3 Non-English Langs | - | 0.7017 | - | 0.6068 | - |
| Multilingual Data | | | | | |
| 3 Non-English Langs | - | 0.7514 | - | 0.5955 | - |
| 5 Non-English Langs | - | 0.7127 | - | 0.4977 | - |

Table 11: Accuracy on MMEval (Bengali) broken down by category.

| | | MMEval (Basque) | | | | |
|---|---|---|---|---|---|---|
| Model | Chat | Language | Hallucinations | Linguistics | Reasoning | Safety |
| **Proprietary Models** | | | | | | |
| GPT-4O | - | | 0.6386 | - | - | 0.3871 |
| **Small (3B parameters)** | | | | | | |
| Qwen2.5-3B-Instruct | - | | 0.4819 | - | - | 0.2688 |
| Glider 3B † | - | | 0.5060 | - | - | 0.4032 |
| M-PROMETHEUS 3B * | - | | 0.5181 | - | - | 0.3656 |
| **Medium (7B parameters)** | | | | | | |
| Qwen2.5-7B-Instruct | - | | 0.5241 | - | - | 0.4624 |
| PROMETHEUS 2 7B † | - | | 0.6084 | - | - | 0.5161 |
| Hercule 7B * | - | | 0.4819 | - | - | 0.2204 |
| M-PROMETHEUS 7B * | - | | 0.5964 | - | - | 0.8172 |
| **Large (14B+ parameters)** | | | | | | |
| Qwen2.5-14B-Instruct | - | | 0.6205 | - | - | 0.3441 |
| PROMETHEUS 2 8x7B † | - | | 0.5271 | - | - | 0.3871 |
| M-PROMETHEUS 14B * | - | | 0.6446 | - | - | 0.9462 |
| **Backbone Model Ablations** | | | | | | |
| Mistral-7B-v0.2-Instruct | - | | 0.4759 | - | - | 0.4086 |
| EuroLLM-9B-Instruct | - | | 0.5181 | - | - | 0.6022 |
| Aya-Expanse-8B | - | | 0.4699 | - | - | 0.3763 |
| Qwen2.5-7B-Instruct | - | | 0.5000 | - | - | 0.5484 |
| **Training Data Ablations** | | | | | | |
| MT Eval Data | - | | 0.7530 | - | - | 0.4731 |
| Translated Data | | | | | | |
| 3 Non-English Langs | - | | 0.4639 | - | - | 0.5914 |
| Multilingual Data | | | | | | |
| 3 Non-English Langs | - | | 0.4940 | - | - | 0.6237 |
| 5 Non-English Langs | - | | 0.5000 | - | - | 0.7742 |

Table 12: Accuracy on MMEval (Basque) broken down by category.

| Model | MMEval (Korean) | | | | | |
| --- | --- | --- | --- | --- | --- | --- |
| | Chat | Language | Hallucinations | Linguistics | Reasoning | Safety |
| **Proprietary Models** | | | | | | |
| GPT-4O | - | | 0.7647 | - | - | 0.5054 |
| **Small (3B parameters)** | | | | | | |
| Qwen2.5-3B-Instruct | - | | 0.5775 | - | - | 0.4355 |
| Glider 3B † | - | | 0.5107 | - | - | 0.4409 |
| M-PROMETHEUS 3B * | - | | 0.5722 | - | - | 0.7419 |
| **Medium (7B parameters)** | | | | | | |
| Qwen2.5-7B-Instruct | - | | 0.5775 | - | - | 0.7634 |
| PROMETHEUS 2 7B † | - | | 0.4706 | - | - | 0.7957 |
| Hercule 7B * | - | | 0.4893 | - | - | 0.3548 |
| M-PROMETHEUS 7B * | - | | 0.6257 | - | - | 0.9785 |
| **Large (14B+ parameters)** | | | | | | |
| Qwen2.5-14B-Instruct | - | | 0.8610 | - | - | 0.5054 |
| PROMETHEUS 2 8x7B † | - | | 0.6524 | - | - | 0.7473 |
| M-PROMETHEUS 14B * | - | | 0.8717 | - | - | 0.9677 |
| **Backbone Model Ablations** | | | | | | |
| Mistral-7B-v0.2-Instruct | - | | 0.4947 | - | - | 0.4946 |
| EuroLLM-9B-Instruct | - | | 0.5882 | - | - | 0.8065 |
| Aya-Expanse-8B | - | | 0.5829 | - | - | 0.6237 |
| Qwen2.5-7B-Instruct | - | | 0.7273 | - | - | 0.6559 |
| **Training Data Ablations** | | | | | | |
| MT Eval Data | - | | 0.7594 | - | - | 0.7634 |
| Translated Data | | | | | | |
|   3 Non-English Langs | - | | 0.6203 | - | - | 0.5269 |
| Multilingual Data | | | | | | |
|   3 Non-English Langs | - | | 0.5615 | - | - | 0.8280 |
|   5 Non-English Langs | - | | 0.6364 | - | - | 0.9032 |

Table 13: Accuracy on MMEval (Korean) broken down by category.

| Model | MMEval (Vietnamese) | | | | | |
| --- | --- | --- | --- | --- | --- | --- |
| | Chat | Language Hallucinations | Linguistics | Reasoning | Safety |
| **Proprietary Models** | | | | | | |
| GPT-4O | - | 0.6804 | - | - | 0.5556 |
| **Small (3B parameters)** | | | | | | |
| Qwen2.5-3B-Instruct | - | 0.5206 | - | - | 0.5556 |
| Glider 3B † | - | 0.4974 | - | - | 0.3944 |
| M-PROMETHEUS 3B * | - | 0.5361 | - | - | 0.8000 |
| **Medium (7B parameters)** | | | | | | |
| Qwen2.5-7B-Instruct | - | 0.5464 | - | - | 0.7444 |
| PROMETHEUS 2 7B † | - | 0.4742 | - | - | 0.7000 |
| Hercule 7B * | - | 0.5284 | - | - | 0.3222 |
| M-PROMETHEUS 7B * | - | 0.6031 | - | - | 0.9556 |
| **Large (14B+ parameters)** | | | | | | |
| Qwen2.5-14B-Instruct | - | 0.7062 | - | - | 0.5222 |
| PROMETHEUS 2 8x7B † | - | 0.6031 | - | - | 0.6556 |
| M-PROMETHEUS 14B * | - | 0.7912 | - | - | 0.9889 |
| **Backbone Model Ablations** | | | | | | |
| Mistral-7B-v0.2-Instruct | - | 0.4768 | - | - | 0.3667 |
| EuroLLM-9B-Instruct | - | 0.5387 | - | - | 0.5944 |
| Aya-Expanse-8B | - | 0.5567 | - | - | 0.4889 |
| Qwen2.5-7B-Instruct | - | 0.6856 | - | - | 0.6556 |
| **Training Data Ablations** | | | | | | |
| MT Eval Data | - | 0.7526 | - | - | 0.8111 |
| Translated Data | | | | | |
| 3 Non-English Langs | - | 0.5876 | - | - | 0.6222 |
| Multilingual Data | | | | | |
| 3 Non-English Langs | - | 0.5670 | - | - | 0.8444 |
| 5 Non-English Langs | - | 0.6598 | - | - | 0.9667 |

Table 14: Accuracy on MMEval (Vietnamese) broken down by category.

| | **MMEval (Italian)** | | | | | |
|---|---|---|---|---|---|---|
| Model | Chat | Language | Hallucinations | Linguistics | Reasoning | Safety |
| **Proprietary Models** | | | | | | |
| GPT-4O | - | - | | 0.9333 | - | 0.5111 |
| **Small (3B parameters)** | | | | | | |
| Qwen2.5-3B-Instruct | - | - | | 0.6667 | - | 0.5389 |
| Glider 3B † | - | - | | 0.6533 | - | 0.6889 |
| M-PROMETHEUS 3B * | - | - | | 0.7200 | - | 0.7667 |
| **Medium (7B parameters)** | | | | | | |
| Qwen2.5-7B-Instruct | - | - | | 0.6400 | - | 0.8000 |
| PROMETHEUS 2 7B † | - | - | | 0.6933 | - | 0.7000 |
| Hercule 7B * | - | - | | 0.5533 | - | 0.3389 |
| M-PROMETHEUS 7B * | - | - | | 0.6933 | - | 0.9778 |
| **Large (14B+ parameters)** | | | | | | |
| Qwen2.5-14B-Instruct | - | - | | 0.7333 | - | 0.4778 |
| PROMETHEUS 2 8x7B † | - | - | | 0.6800 | - | 0.8111 |
| M-PROMETHEUS 14B * | - | - | | 0.7600 | - | 0.9556 |
| **Backbone Model Ablations** | | | | | | |
| Mistral-7B-v0.2-Instruct | - | - | | 0.5867 | - | 0.4778 |
| EuroLLM-9B-Instruct | - | - | | 0.6933 | - | 0.8444 |
| Aya-Expanse-8B | - | - | | 0.6933 | - | 0.5889 |
| Qwen2.5-7B-Instruct | - | - | | 0.7333 | - | 0.6556 |
| **Training Data Ablations** | | | | | | |
| MT Eval Data | - | - | | 0.6933 | - | 0.6889 |
| Translated Data | | | | | | |
|   3 Non-English Langs | - | - | | 0.7200 | - | 0.6333 |
| Multilingual Data | | | | | | |
|   3 Non-English Langs | - | - | | 0.6800 | - | 0.8889 |
|   5 Non-English Langs | - | - | | 0.6800 | - | 0.9667 |

Table 15: Accuracy on MMEval (Italian) broken down by category.

| Model | MMEval (Galician) | | | | | |
| --- | --- | --- | --- | --- | --- | --- |
| | Chat | Language | Hallucinations | Linguistics | Reasoning | Safety |
| **Proprietary Models** | | | | | | |
| GPT-4O | - | - | | 0.8800 | - | 0.5730 |
| **Small (3B parameters)** | | | | | | |
| Qwen2.5-3B-Instruct | - | - | | 0.7000 | - | 0.5056 |
| Glider 3B † | - | - | | 0.6133 | - | 0.5393 |
| M-PROMETHEUS 3B * | - | - | | 0.6933 | - | 0.8202 |
| **Medium (7B parameters)** | | | | | | |
| Qwen2.5-7B-Instruct | - | - | | 0.6000 | - | 0.8652 |
| PROMETHEUS 2 7B † | - | - | | 0.6933 | - | 0.7022 |
| Hercule 7B * | - | - | | 0.5200 | - | 0.3427 |
| M-PROMETHEUS 7B * | - | - | | 0.5600 | - | 0.9213 |
| **Large (14B+ parameters)** | | | | | | |
| Qwen2.5-14B-Instruct | - | - | | 0.7200 | - | 0.5955 |
| PROMETHEUS 2 8x7B † | - | - | | 0.6667 | - | 0.7865 |
| M-PROMETHEUS 14B * | - | - | | 0.6267 | - | 0.9663 |
| **Backbone Model Ablations** | | | | | | |
| Mistral-7B-v0.2-Instruct | - | - | | 0.6533 | - | 0.5393 |
| EuroLLM-9B-Instruct | - | - | | 0.6133 | - | 0.7865 |
| Aya-Expanse-8B | - | - | | 0.6800 | - | 0.6517 |
| Qwen2.5-7B-Instruct | - | - | | 0.6933 | - | 0.6180 |
| **Training Data Ablations** | | | | | | |
| MT Eval Data | - | - | | 0.6133 | - | 0.6067 |
| Translated Data | | | | | | |
| 3 Non-English Langs | - | - | | 0.6533 | - | 0.6292 |
| Multilingual Data | | | | | | |
| 3 Non-English Langs | - | - | | 0.5600 | - | 0.8315 |
| 5 Non-English Langs | - | - | | 0.6000 | - | 0.9213 |

Table 16: Accuracy on MMEval (Galician) broken down by category.

| Model | MMEval (Japanese) | | | | | |
| | Chat | Language Hallucinations | Linguistics | Reasoning | Safety |
|---|---|---|---|---|---|---|
| **Proprietary Models** | | | | | | |
| GPT-4O | - | - | | - | 0.8009 | 0.5604 |
| **Small (3B parameters)** | | | | | | |
| Qwen2.5-3B-Instruct | - | - | | - | 0.5231 | 0.5604 |
| Glider 3B † | - | - | | - | 0.6065 | 0.4615 |
| M-PROMETHEUS 3B * | - | - | | - | 0.5741 | 0.8791 |
| **Medium (7B parameters)** | | | | | | |
| Qwen2.5-7B-Instruct | - | - | | - | 0.6204 | 0.7253 |
| PROMETHEUS 2 7B † | - | - | | - | 0.5833 | 0.7088 |
| Hercule 7B * | - | - | | - | 0.5463 | 0.3352 |
| M-PROMETHEUS 7B * | - | - | | - | 0.5787 | 0.9560 |
| **Large (14B+ parameters)** | | | | | | |
| Qwen2.5-14B-Instruct | - | - | | - | 0.7083 | 0.5495 |
| PROMETHEUS 2 8x7B † | - | - | | - | 0.5602 | 0.6593 |
| M-PROMETHEUS 14B * | - | - | | - | 0.7060 | 0.9670 |
| **Backbone Model Ablations** | | | | | | |
| Mistral-7B-v0.2-Instruct | - | - | | - | 0.5278 | 0.5275 |
| EuroLLM-9B-Instruct | - | - | | - | 0.5486 | 0.8297 |
| Aya-Expanse-8B | - | - | | - | 0.5972 | 0.5495 |
| Qwen2.5-7B-Instruct | - | - | | - | 0.6088 | 0.6593 |
| **Training Data Ablations** | | | | | | |
| MT Eval Data | - | - | | - | 0.6065 | 0.8462 |
| Translated Data | | | | | | |
|   3 Non-English Langs | - | - | | - | 0.5972 | 0.6264 |
| Multilingual Data | | | | | | |
|   3 Non-English Langs | - | - | | - | 0.6204 | 0.8681 |
|   5 Non-English Langs | - | - | | - | 0.5370 | 0.9451 |

Table 17: Accuracy on MMEval (Japanese) broken down by category.

| Model | MMEval (Thai) | | | | | |
| | Chat | Language Hallucinations | Linguistics | Reasoning | Safety |
|---|---|---|---|---|---|
| **Proprietary Models** | | | | | |
| GPT-4O | - | - | - | 0.7970 | 0.4945 |
| **Small (3B parameters)** | | | | | |
| Qwen2.5-3B-Instruct | - | - | - | 0.5888 | 0.5055 |
| Glider 3B † | - | - | - | 0.5711 | 0.3626 |
| M-PROMETHEUS 3B * | - | - | - | 0.5787 | 0.8022 |
| **Medium (7B parameters)** | | | | | |
| Qwen2.5-7B-Instruct | - | - | - | 0.6650 | 0.8022 |
| PROMETHEUS 2 7B † | - | - | - | 0.5685 | 0.6813 |
| Hercule 7B * | - | - | - | 0.5228 | 0.3901 |
| M-PROMETHEUS 7B * | - | - | - | 0.6041 | 0.9670 |
| **Large (14B+ parameters)** | | | | | |
| Qwen2.5-14B-Instruct | - | - | - | 0.6802 | 0.6044 |
| PROMETHEUS 2 8x7B † | - | - | - | 0.6015 | 0.6154 |
| M-PROMETHEUS 14B * | - | - | - | 0.6853 | 0.9890 |
| **Backbone Model Ablations** | | | | | |
| Mistral-7B-v0.2-Instruct | - | - | - | 0.5685 | 0.3791 |
| EuroLLM-9B-Instruct | - | - | - | 0.6091 | 0.5000 |
| Aya-Expanse-8B | - | - | - | 0.5812 | 0.4615 |
| Qwen2.5-7B-Instruct | - | - | - | 0.6447 | 0.6868 |
| **Training Data Ablations** | | | | | |
| MT Eval Data | - | - | - | 0.6548 | 0.7912 |
| Translated Data | | | | | |
| 3 Non-English Langs | - | - | - | 0.6574 | 0.6813 |
| Multilingual Data | | | | | |
| 3 Non-English Langs | - | - | - | 0.6041 | 0.8462 |
| 5 Non-English Langs | - | - | - | 0.5431 | 0.9011 |

Table 18: Accuracy on MMEval (Thai) broken down by category.

| Model | MMEval (Telugu) | | | | | |
| | Chat | Language Hallucinations | Linguistics | Reasoning | Safety |
|---|---|---|---|---|---|---|
| **Proprietary Models** | | | | | | |
| GPT-4O | - | - | | - | 0.8739 | - |
| **Small (3B parameters)** | | | | | | |
| Qwen2.5-3B-Instruct | - | - | | - | 0.6014 | - |
| Glider 3B † | - | - | | - | 0.5766 | - |
| M-PROMETHEUS 3B * | - | - | | - | 0.6059 | - |
| **Medium (7B parameters)** | | | | | | |
| Qwen2.5-7B-Instruct | - | - | | - | 0.6779 | - |
| PROMETHEUS 2 7B † | - | - | | - | 0.5270 | - |
| Hercule 7B * | - | - | | - | 0.5428 | - |
| M-PROMETHEUS 7B * | - | - | | - | 0.6577 | - |
| **Large (14B+ parameters)** | | | | | | |
| Qwen2.5-14B-Instruct | - | - | | - | 0.7320 | - |
| PROMETHEUS 2 8x7B † | - | - | | - | 0.6059 | - |
| M-PROMETHEUS 14B * | - | - | | - | 0.7207 | - |
| **Backbone Model Ablations** | | | | | | |
| Mistral-7B-v0.2-Instruct | - | - | | - | 0.5248 | - |
| EuroLLM-9B-Instruct | - | - | | - | 0.5000 | - |
| Aya-Expanse-8B | - | - | | - | 0.5068 | - |
| Qwen2.5-7B-Instruct | - | - | | - | 0.6419 | - |
| **Training Data Ablations** | | | | | | |
| MT Eval Data | - | - | | - | 0.6239 | - |
| Translated Data | | | | | | |
| 3 Non-English Langs | - | - | | - | 0.6441 | - |
| Multilingual Data | | | | | | |
| 3 Non-English Langs | - | - | | - | 0.6892 | - |
| 5 Non-English Langs | - | - | | - | 0.5338 | - |

Table 19: Accuracy on MMEval (Telugu) broken down by category.

| Model | MMEval (Swahili) | | | | | |
| | Chat | Language Hallucinations | Linguistics | Reasoning | Safety |
|---|---|---|---|---|---|
| **Proprietary Models** | | | | | |
| GPT-4O | - | - | - | 0.8047 | 0.4624 |
| **Small (3B parameters)** | | | | | |
| Qwen2.5-3B-Instruct | - | - | - | 0.5930 | 0.3011 |
| Glider 3B † | - | - | - | 0.6721 | 0.4946 |
| M-PROMETHEUS 3B * | - | - | - | 0.6651 | 0.3763 |
| **Medium (7B parameters)** | | | | | |
| Qwen2.5-7B-Instruct | - | - | - | 0.6186 | 0.4086 |
| PROMETHEUS 2 7B † | - | - | - | 0.6628 | 0.4839 |
| Hercule 7B * | - | - | - | 0.5163 | 0.1774 |
| M-PROMETHEUS 7B * | - | - | - | 0.7209 | 0.7634 |
| **Large (14B+ parameters)** | | | | | |
| Qwen2.5-14B-Instruct | - | - | - | 0.7256 | 0.4516 |
| PROMETHEUS 2 8x7B † | - | - | - | 0.6302 | 0.4624 |
| M-PROMETHEUS 14B * | - | - | - | 0.7628 | 0.9462 |
| **Backbone Model Ablations** | | | | | |
| Mistral-7B-v0.2-Instruct | - | - | - | 0.6651 | 0.4194 |
| EuroLLM-9B-Instruct | - | - | - | 0.6419 | 0.5591 |
| Aya-Expanse-8B | - | - | - | 0.6744 | 0.3226 |
| Qwen2.5-7B-Instruct | - | - | - | 0.7302 | 0.5806 |
| **Training Data Ablations** | | | | | |
| MT Eval Data | - | - | - | 0.6977 | 0.5161 |
| Translated Data | | | | | |
|   3 Non-English Langs | - | - | - | 0.6860 | 0.6022 |
| Multilingual Data | | | | | |
|   3 Non-English Langs | - | - | - | 0.6791 | 0.6129 |
|   5 Non-English Langs | - | - | - | 0.5884 | 0.7527 |

Table 20: Accuracy on MMEval (Swahili) broken down by category.

## C.2 M-RewardBench

| Model | M-RewardBench (Arabic) | | | |
| --- | --- | --- | --- | --- |
| | Chat | Chat Hard | Reasoning | Safety |
| **Proprietary Models** | | | | |
| GPT-4O | 0.9628 | 0.6658 | 0.8790 | 0.8668 |
| **Small (3B parameters)** | | | | |
| Qwen2.5-3B-Instruct | 0.8041 | 0.4558 | 0.6577 | 0.6889 |
| Glider 3B † | 0.5422 | 0.5061 | 0.7878 | 0.6257 |
| M-PROMETHEUS 3B * | 0.8666 | 0.4165 | 0.6657 | 0.7636 |
| **Medium (7B parameters)** | | | | |
| Qwen2.5-7B-Instruct | 0.9088 | 0.5332 | 0.8094 | 0.7799 |
| PROMETHEUS 2 7B † | 0.7027 | 0.4951 | 0.6888 | 0.6298 |
| Hercule 7B * | 0.7990 | 0.4275 | 0.6720 | 0.6386 |
| M-PROMETHEUS 7B * | 0.8818 | 0.5111 | 0.8252 | 0.7758 |
| **Large (14B+ parameters)** | | | | |
| Qwen2.5-14B-Instruct | 0.9426 | 0.6093 | 0.8276 | 0.8234 |
| PROMETHEUS 2 8x7B † | 0.8750 | 0.3931 | 0.8112 | 0.7432 |
| M-PROMETHEUS 14B * | 0.8953 | 0.5197 | 0.8290 | 0.8234 |
| **Backbone Model Ablations** | | | | |
| Mistral-7B-v0.2-Instruct | 0.6841 | 0.4619 | 0.6587 | 0.4885 |
| EuroLLM-9B-Instruct | 0.8767 | 0.4423 | 0.7364 | 0.8179 |
| Aya-Expanse-8B | 0.8682 | 0.4300 | 0.7490 | 0.8057 |
| Qwen2.5-7B-Instruct | 0.8784 | 0.4791 | 0.8455 | 0.7948 |
| **Training Data Ablations** | | | | |
| MT Eval Data | 0.8851 | 0.4963 | 0.8378 | 0.8098 |
| Translated Data | | | | |
|   3 Non-English Langs | 0.8885 | 0.5086 | 0.8315 | 0.8261 |
| Multilingual Data | | | | |
|   3 Non-English Langs | 0.7901 | 0.5591 | 0.8422 | 0.7883 |
|   5 Non-English Langs | 0.8209 | 0.5135 | 0.8294 | 0.7840 |

Table 21: Accuracy on M-RewardBench (Arabic) broken down by category.

| Model | M-RewardBench (Czech) | | | |
| | Chat | Chat Hard | Reasoning | Safety |
|---|---|---|---|---|
| **Proprietary Models** | | | | |
| GPT-4o | 0.9628 | 0.6683 | 0.9073 | 0.8614 |
| **Small (3B parameters)** | | | | |
| Qwen2.5-3B-Instruct | 0.8125 | 0.4079 | 0.6829 | 0.6984 |
| Glider 3B † | 0.5997 | 0.4324 | 0.8476 | 0.5863 |
| M-PROMETHEUS 3B * | 0.8750 | 0.4423 | 0.6776 | 0.7255 |
| **Medium (7B parameters)** | | | | |
| Qwen2.5-7B-Instruct | 0.9155 | 0.4889 | 0.8028 | 0.7799 |
| PROMETHEUS 2 7B † | 0.8260 | 0.4472 | 0.6867 | 0.6705 |
| Hercule 7B * | 0.8328 | 0.4128 | 0.6629 | 0.6243 |
| M-PROMETHEUS 7B * | 0.9088 | 0.4681 | 0.8052 | 0.7867 |
| **Large (14B+ parameters)** | | | | |
| Qwen2.5-14B-Instruct | 0.9189 | 0.5430 | 0.8273 | 0.7772 |
| PROMETHEUS 2 8x7B † | 0.9037 | 0.4054 | 0.8115 | 0.7758 |
| M-PROMETHEUS 14B * | 0.9088 | 0.5012 | 0.8371 | 0.8302 |
| **Backbone Model Ablations** | | | | |
| Mistral-7B-v0.2-Instruct | 0.8514 | 0.4005 | 0.6794 | 0.6658 |
| EuroLLM-9B-Instruct | 0.8632 | 0.4398 | 0.7350 | 0.8030 |
| Aya-Expanse-8B | 0.8733 | 0.4201 | 0.7766 | 0.7731 |
| Qwen2.5-7B-Instruct | 0.8784 | 0.4693 | 0.8231 | 0.7704 |
| **Training Data Ablations** | | | | |
| MT Eval Data | 0.9223 | 0.4767 | 0.8150 | 0.7731 |
| Translated Data | | | | |
|   3 Non-English Langs | 0.9088 | 0.4742 | 0.8329 | 0.7785 |
| Multilingual Data | | | | |
|   3 Non-English Langs | 0.8618 | 0.5267 | 0.8386 | 0.7842 |
|   5 Non-English Langs | 0.8514 | 0.4988 | 0.8367 | 0.7880 |

Table 22: Accuracy on M-RewardBench (Czech) broken down by category.

| Model | M-RewardBench (German) | | | |
| --- | --- | --- | --- | --- |
| | Chat | Chat Hard | Reasoning | Safety |
| **Proprietary Models** | | | | |
| GPT-4O | 0.9493 | 0.6830 | 0.8860 | 0.8560 |
| **Small (3B parameters)** | | | | |
| Qwen2.5-3B-Instruct | 0.8378 | 0.4668 | 0.6951 | 0.6834 |
| Glider 3B † | 0.6622 | 0.5418 | 0.8762 | 0.7296 |
| M-PROMETHEUS 3B * | 0.8615 | 0.4214 | 0.7003 | 0.7507 |
| **Medium (7B parameters)** | | | | |
| Qwen2.5-7B-Instruct | 0.9459 | 0.5430 | 0.8290 | 0.7758 |
| PROMETHEUS 2 7B † | 0.8429 | 0.4754 | 0.7196 | 0.6902 |
| Hercule 7B * | 0.8142 | 0.4533 | 0.6857 | 0.6501 |
| M-PROMETHEUS 7B * | 0.9088 | 0.5233 | 0.8262 | 0.8030 |
| **Large (14B+ parameters)** | | | | |
| Qwen2.5-14B-Instruct | 0.9493 | 0.6437 | 0.8476 | 0.8234 |
| PROMETHEUS 2 8x7B † | 0.9088 | 0.4509 | 0.8210 | 0.7704 |
| M-PROMETHEUS 14B * | 0.9358 | 0.5221 | 0.8399 | 0.8261 |
| **Backbone Model Ablations** | | | | |
| Mistral-7B-v0.2-Instruct | 0.8851 | 0.4103 | 0.7010 | 0.6766 |
| EuroLLM-9B-Instruct | 0.9037 | 0.4410 | 0.7332 | 0.8152 |
| Aya-Expanse-8B | 0.8598 | 0.4177 | 0.7706 | 0.7955 |
| Qwen2.5-7B-Instruct | 0.8885 | 0.5061 | 0.8276 | 0.7989 |
| **Training Data Ablations** | | | | |
| MT Eval Data | 0.9088 | 0.4865 | 0.8220 | 0.7948 |
| Translated Data | | | | |
| 3 Non-English Langs | 0.9088 | 0.4914 | 0.8402 | 0.8003 |
| Multilingual Data | | | | |
| 3 Non-English Langs | 0.8492 | 0.5639 | 0.8338 | 0.8035 |
| 5 Non-English Langs | 0.8480 | 0.5184 | 0.8378 | 0.8043 |

Table 23: Accuracy on M-RewardBench (German) broken down by category.

| Model | M-RewardBench (Greek) | | | |
| | Chat | Chat Hard | Reasoning | Safety |
|---|---|---|---|---|
| **Proprietary Models** | | | | |
| GPT-4O | 0.9527 | 0.6364 | 0.8818 | 0.8682 |
| **Small (3B parameters)** | | | | |
| Qwen2.5-3B-Instruct | 0.7500 | 0.4201 | 0.6878 | 0.6365 |
| Glider 3B † | 0.5777 | 0.4582 | 0.7262 | 0.5136 |
| M-PROMETHEUS 3B * | 0.8581 | 0.4079 | 0.6675 | 0.6984 |
| **Medium (7B parameters)** | | | | |
| Qwen2.5-7B-Instruct | 0.8919 | 0.5111 | 0.8059 | 0.7323 |
| PROMETHEUS 2 7B † | 0.6571 | 0.4558 | 0.6965 | 0.5897 |
| Hercule 7B * | 0.8446 | 0.4398 | 0.6860 | 0.6345 |
| M-PROMETHEUS 7B * | 0.8716 | 0.4975 | 0.7913 | 0.7452 |
| **Large (14B+ parameters)** | | | | |
| Qwen2.5-14B-Instruct | 0.9358 | 0.5651 | 0.8210 | 0.7799 |
| PROMETHEUS 2 8x7B † | 0.8784 | 0.3993 | 0.8056 | 0.7473 |
| M-PROMETHEUS 14B * | 0.8936 | 0.4742 | 0.8175 | 0.8152 |
| **Backbone Model Ablations** | | | | |
| Mistral-7B-v0.2-Instruct | 0.6976 | 0.4533 | 0.6678 | 0.5129 |
| EuroLLM-9B-Instruct | 0.8818 | 0.4349 | 0.7406 | 0.7867 |
| Aya-Expanse-8B | 0.8716 | 0.4275 | 0.7626 | 0.7758 |
| Qwen2.5-7B-Instruct | 0.8750 | 0.4816 | 0.8140 | 0.7446 |
| **Training Data Ablations** | | | | |
| MT Eval Data | 0.8919 | 0.4496 | 0.8042 | 0.7323 |
| Translated Data | | | | |
|   3 Non-English Langs | 0.8716 | 0.4693 | 0.8262 | 0.7636 |
| Multilingual Data | | | | |
|   3 Non-English Langs | 0.8452 | 0.5095 | 0.8415 | 0.7476 |
|   5 Non-English Langs | 0.8682 | 0.4619 | 0.8189 | 0.7432 |

Table 24: Accuracy on M-RewardBench (Greek) broken down by category.

| Model | M-RewardBench (French) | | | |
| --- | --- | --- | --- | --- |
| | Chat | Chat Hard | Reasoning | Safety |
| **Proprietary Models** | | | | |
| GPT-4o | 0.9730 | 0.6658 | 0.9024 | 0.8614 |
| **Small (3B parameters)** | | | | |
| Qwen2.5-3B-Instruct | 0.8564 | 0.4853 | 0.7021 | 0.6936 |
| Glider 3B † | 0.6757 | 0.4988 | 0.8797 | 0.7622 |
| M-PROMETHEUS 3B * | 0.8514 | 0.4361 | 0.6913 | 0.7670 |
| **Medium (7B parameters)** | | | | |
| Qwen2.5-7B-Instruct | 0.9493 | 0.5627 | 0.8406 | 0.7921 |
| PROMETHEUS 2 7B † | 0.8378 | 0.4582 | 0.6888 | 0.7500 |
| Hercule 7B * | 0.8277 | 0.4570 | 0.6822 | 0.6393 |
| M-PROMETHEUS 7B * | 0.9105 | 0.5369 | 0.8514 | 0.8050 |
| **Large (14B+ parameters)** | | | | |
| Qwen2.5-14B-Instruct | 0.9392 | 0.6327 | 0.8455 | 0.8152 |
| PROMETHEUS 2 8x7B † | 0.9324 | 0.4484 | 0.8220 | 0.7982 |
| M-PROMETHEUS 14B * | 0.9223 | 0.5356 | 0.8406 | 0.8438 |
| **Backbone Model Ablations** | | | | |
| Mistral-7B-v0.2-Instruct | 0.8649 | 0.4251 | 0.7017 | 0.7181 |
| EuroLLM-9B-Instruct | 0.8902 | 0.4717 | 0.7566 | 0.8084 |
| Aya-Expanse-8B | 0.8784 | 0.4251 | 0.7762 | 0.8003 |
| Qwen2.5-7B-Instruct | 0.8953 | 0.5111 | 0.8601 | 0.8139 |
| **Training Data Ablations** | | | | |
| MT Eval Data | 0.9088 | 0.5135 | 0.8462 | 0.8071 |
| Translated Data | | | | |
|   3 Non-English Langs | 0.9020 | 0.4988 | 0.8465 | 0.8234 |
| Multilingual Data | | | | |
|   3 Non-English Langs | 0.8111 | 0.5593 | 0.8485 | 0.7940 |
|   5 Non-English Langs | 0.8345 | 0.5479 | 0.8399 | 0.8071 |

Table 25: Accuracy on M-RewardBench (French) broken down by category.

| | M-RewardBench (Hebrew) | | | |
|---|---|---|---|---|
| **Model** | Chat | Chat Hard | Reasoning | Safety |
| **Proprietary Models** | | | | |
| GPT-4O | 0.9493 | 0.6560 | 0.8888 | 0.8641 |
| **Small (3B parameters)** | | | | |
| Qwen2.5-3B-Instruct | 0.8074 | 0.4238 | 0.6650 | 0.6664 |
| Glider 3B † | 0.5389 | 0.4914 | 0.7829 | 0.5734 |
| M-PROMETHEUS 3B * | 0.8682 | 0.4570 | 0.6766 | 0.7378 |
| **Medium (7B parameters)** | | | | |
| Qwen2.5-7B-Instruct | 0.9358 | 0.4668 | 0.7986 | 0.7459 |
| PROMETHEUS 2 7B † | 0.6385 | 0.4877 | 0.6871 | 0.5999 |
| Hercule 7B * | 0.8041 | 0.4251 | 0.6720 | 0.6270 |
| M-PROMETHEUS 7B * | 0.8885 | 0.4730 | 0.8287 | 0.7785 |
| **Large (14B+ parameters)** | | | | |
| Qwen2.5-14B-Instruct | 0.9088 | 0.5762 | 0.8399 | 0.7779 |
| PROMETHEUS 2 8x7B † | 0.8632 | 0.3673 | 0.8017 | 0.7120 |
| M-PROMETHEUS 14B * | 0.8953 | 0.5049 | 0.8192 | 0.8111 |
| **Backbone Model Ablations** | | | | |
| Mistral-7B-v0.2-Instruct | 0.6875 | 0.4779 | 0.6836 | 0.5000 |
| EuroLLM-9B-Instruct | 0.7264 | 0.4337 | 0.7143 | 0.6664 |
| Aya-Expanse-8B | 0.8818 | 0.4054 | 0.7577 | 0.7717 |
| Qwen2.5-7B-Instruct | 0.8767 | 0.4423 | 0.8423 | 0.7690 |
| **Training Data Ablations** | | | | |
| MT Eval Data | 0.9088 | 0.3931 | 0.8325 | 0.7867 |
| Translated Data | | | | |
|   3 Non-English Langs | 0.9155 | 0.4398 | 0.8374 | 0.7908 |
| Multilingual Data | | | | |
|   3 Non-English Langs | 0.8387 | 0.4943 | 0.8531 | 0.7653 |
|   5 Non-English Langs | 0.8716 | 0.4914 | 0.8371 | 0.7826 |

Table 26: Accuracy on M-RewardBench (Hebrew) broken down by category.

| Model | M-RewardBench (Hindi) | | | |
| | Chat | Chat Hard | Reasoning | Safety |
|---|---|---|---|---|
| **Proprietary Models** | | | | |
| GPT-4O | 0.9561 | 0.6241 | 0.8804 | 0.8560 |
| **Small (3B parameters)** | | | | |
| Qwen2.5-3B-Instruct | 0.7264 | 0.4214 | 0.7108 | 0.6495 |
| Glider 3B † | 0.5963 | 0.4840 | 0.5322 | 0.5000 |
| M-PROMETHEUS 3B * | 0.8176 | 0.4091 | 0.6832 | 0.7079 |
| **Medium (7B parameters)** | | | | |
| Qwen2.5-7B-Instruct | 0.8953 | 0.4914 | 0.8161 | 0.7514 |
| PROMETHEUS 2 7B † | 0.6537 | 0.4717 | 0.6850 | 0.5978 |
| Hercule 7B * | 0.8429 | 0.4361 | 0.6878 | 0.6488 |
| M-PROMETHEUS 7B * | 0.8480 | 0.4631 | 0.8059 | 0.7541 |
| **Large (14B+ parameters)** | | | | |
| Qwen2.5-14B-Instruct | 0.9291 | 0.5258 | 0.8357 | 0.7989 |
| PROMETHEUS 2 8x7B † | 0.8581 | 0.3943 | 0.7825 | 0.7432 |
| M-PROMETHEUS 14B * | 0.9071 | 0.4914 | 0.8462 | 0.7989 |
| **Backbone Model Ablations** | | | | |
| Mistral-7B-v0.2-Instruct | 0.5794 | 0.4988 | 0.6573 | 0.5170 |
| EuroLLM-9B-Instruct | 0.8125 | 0.4423 | 0.7605 | 0.8152 |
| Aya-Expanse-8B | 0.7703 | 0.4238 | 0.7364 | 0.7554 |
| Qwen2.5-7B-Instruct | 0.8041 | 0.4889 | 0.8343 | 0.7908 |
| **Training Data Ablations** | | | | |
| MT Eval Data | 0.8547 | 0.4300 | 0.8322 | 0.7867 |
| Translated Data | | | | |
|   3 Non-English Langs | 0.8311 | 0.4791 | 0.8287 | 0.8247 |
| Multilingual Data | | | | |
|   3 Non-English Langs | 0.7710 | 0.5192 | 0.8625 | 0.7707 |
|   5 Non-English Langs | 0.8547 | 0.4521 | 0.8147 | 0.7514 |

Table 27: Accuracy on M-RewardBench (Hindi) broken down by category.

| | **M-RewardBench (Indonesian)** | | | |
|---|---|---|---|---|
| **Model** | Chat | Chat Hard | Reasoning | Safety |
| **Proprietary Models** | | | | |
| GPT-4O | 0.9561 | 0.6634 | 0.8965 | 0.8723 |
| **Small (3B parameters)** | | | | |
| Qwen2.5-3B-Instruct | 0.8497 | 0.4496 | 0.7087 | 0.7249 |
| Glider 3B † | 0.6926 | 0.5086 | 0.7689 | 0.6556 |
| M-PROMETHEUS 3B * | 0.8514 | 0.4472 | 0.6867 | 0.7582 |
| **Medium (7B parameters)** | | | | |
| Qwen2.5-7B-Instruct | 0.9324 | 0.5258 | 0.8269 | 0.7989 |
| PROMETHEUS 2 7B † | 0.7905 | 0.4214 | 0.7038 | 0.7317 |
| Hercule 7B * | 0.8294 | 0.4177 | 0.6755 | 0.6719 |
| M-PROMETHEUS 7B * | 0.8919 | 0.5012 | 0.8206 | 0.7962 |
| **Large (14B+ parameters)** | | | | |
| Qwen2.5-14B-Instruct | 0.9324 | 0.6192 | 0.8434 | 0.8193 |
| PROMETHEUS 2 8x7B † | 0.8801 | 0.4066 | 0.8066 | 0.8050 |
| M-PROMETHEUS 14B * | 0.9088 | 0.5565 | 0.8469 | 0.8356 |
| **Backbone Model Ablations** | | | | |
| Mistral-7B-v0.2-Instruct | 0.6824 | 0.4619 | 0.6860 | 0.6617 |
| EuroLLM-9B-Instruct | 0.7669 | 0.4545 | 0.7510 | 0.7921 |
| Aya-Expanse-8B | 0.8176 | 0.4447 | 0.7497 | 0.8193 |
| Qwen2.5-7B-Instruct | 0.8851 | 0.4865 | 0.8357 | 0.8193 |
| **Training Data Ablations** | | | | |
| MT Eval Data | 0.8818 | 0.4840 | 0.8413 | 0.8288 |
| Translated Data | | | | |
|   3 Non-English Langs | 0.8682 | 0.4865 | 0.8311 | 0.8397 |
| Multilingual Data | | | | |
|   3 Non-English Langs | 0.7596 | 0.5619 | 0.8499 | 0.7980 |
|   5 Non-English Langs | 0.8209 | 0.5012 | 0.8402 | 0.8003 |

Table 28: Accuracy on M-RewardBench (Indonesian) broken down by category.

| Model | M-RewardBench (Italian) | | | |
| | Chat | Chat Hard | Reasoning | Safety |
|---|---|---|---|---|
| **Proprietary Models** | | | | |
| GPT-4o | 0.9595 | 0.6658 | 0.9014 | 0.8791 |
| **Small (3B parameters)** | | | | |
| Qwen2.5-3B-Instruct | 0.8615 | 0.4521 | 0.6759 | 0.6970 |
| Glider 3B † | 0.6554 | 0.4853 | 0.8780 | 0.7615 |
| M-PROMETHEUS 3B * | 0.8750 | 0.4226 | 0.6944 | 0.7717 |
| **Medium (7B parameters)** | | | | |
| Qwen2.5-7B-Instruct | 0.9493 | 0.5528 | 0.8381 | 0.7853 |
| PROMETHEUS 2 7B † | 0.8514 | 0.4619 | 0.6986 | 0.7514 |
| Hercule 7B * | 0.8176 | 0.4533 | 0.6850 | 0.6685 |
| M-PROMETHEUS 7B * | 0.9189 | 0.5160 | 0.8371 | 0.8084 |
| **Large (14B+ parameters)** | | | | |
| Qwen2.5-14B-Instruct | 0.9358 | 0.6413 | 0.8465 | 0.8125 |
| PROMETHEUS 2 8x7B † | 0.9291 | 0.4484 | 0.8038 | 0.8077 |
| M-PROMETHEUS 14B * | 0.9054 | 0.5381 | 0.8399 | 0.8234 |
| **Backbone Model Ablations** | | | | |
| Mistral-7B-v0.2-Instruct | 0.8615 | 0.4115 | 0.6783 | 0.6957 |
| EuroLLM-9B-Instruct | 0.8986 | 0.4619 | 0.7549 | 0.8302 |
| Aya-Expanse-8B | 0.8986 | 0.4201 | 0.7990 | 0.8084 |
| Qwen2.5-7B-Instruct | 0.8986 | 0.5061 | 0.8451 | 0.8071 |
| **Training Data Ablations** | | | | |
| MT Eval Data | 0.9122 | 0.4717 | 0.8336 | 0.8043 |
| Translated Data | | | | |
|   3 Non-English Langs | 0.9020 | 0.5012 | 0.8434 | 0.8220 |
| Multilingual Data | | | | |
|   3 Non-English Langs | 0.8452 | 0.5649 | 0.8350 | 0.7802 |
|   5 Non-English Langs | 0.8547 | 0.5086 | 0.8437 | 0.7867 |

Table 29: Accuracy on M-RewardBench (Italian) broken down by category.

| Model | M-RewardBench (Japanese) | | | |
| | Chat | Chat Hard | Reasoning | Safety |
| --- | --- | --- | --- | --- |
| **Proprietary Models** | | | | |
| GPT-4O | 0.9459 | 0.6609 | 0.9028 | 0.8696 |
| **Small (3B parameters)** | | | | |
| Qwen2.5-3B-Instruct | 0.8007 | 0.4410 | 0.6706 | 0.7024 |
| Glider 3B † | 0.5912 | 0.4914 | 0.8147 | 0.6875 |
| M-PROMETHEUS 3B * | 0.7601 | 0.4730 | 0.6731 | 0.7690 |
| **Medium (7B parameters)** | | | | |
| Qwen2.5-7B-Instruct | 0.9122 | 0.5283 | 0.8297 | 0.8084 |
| PROMETHEUS 2 7B † | 0.7264 | 0.4939 | 0.7091 | 0.7018 |
| Hercule 7B * | 0.8159 | 0.4693 | 0.6535 | 0.6332 |
| M-PROMETHEUS 7B * | 0.7872 | 0.5516 | 0.8273 | 0.7731 |
| **Large (14B+ parameters)** | | | | |
| Qwen2.5-14B-Instruct | 0.9375 | 0.6265 | 0.8493 | 0.8288 |
| PROMETHEUS 2 8x7B † | 0.8716 | 0.4152 | 0.7937 | 0.7996 |
| M-PROMETHEUS 14B * | 0.8412 | 0.5479 | 0.8490 | 0.8193 |
| **Backbone Model Ablations** | | | | |
| Mistral-7B-v0.2-Instruct | 0.6740 | 0.4779 | 0.6528 | 0.6855 |
| EuroLLM-9B-Instruct | 0.7872 | 0.4668 | 0.7245 | 0.8071 |
| Aya-Expanse-8B | 0.8277 | 0.4300 | 0.7329 | 0.7772 |
| Qwen2.5-7B-Instruct | 0.8007 | 0.4889 | 0.8301 | 0.8166 |
| **Training Data Ablations** | | | | |
| MT Eval Data | 0.7804 | 0.5012 | 0.8367 | 0.8410 |
| Translated Data | | | | |
| 3 Non-English Langs | 0.8209 | 0.4742 | 0.8171 | 0.8370 |
| Multilingual Data | | | | |
| 3 Non-English Langs | 0.7297 | 0.5781 | 0.8191 | 0.7842 |
| 5 Non-English Langs | 0.7838 | 0.5479 | 0.8087 | 0.7731 |

Table 30: Accuracy on M-RewardBench (Japanese) broken down by category.

| Model | M-RewardBench (Korean) | | | |
| --- | --- | --- | --- | --- |
| | Chat | Chat Hard | Reasoning | Safety |
| **Proprietary Models** | | | | |
| GPT-4O | 0.9493 | 0.6179 | 0.8867 | 0.8519 |
| **Small (3B parameters)** | | | | |
| Qwen2.5-3B-Instruct | 0.8108 | 0.4165 | 0.6339 | 0.6189 |
| Glider 3B † | 0.5507 | 0.4717 | 0.7773 | 0.6182 |
| M-PROMETHEUS 3B * | 0.8784 | 0.4165 | 0.6797 | 0.7188 |
| **Medium (7B parameters)** | | | | |
| Qwen2.5-7B-Instruct | 0.9628 | 0.5184 | 0.8042 | 0.7812 |
| PROMETHEUS 2 7B † | 0.7787 | 0.4582 | 0.7101 | 0.7344 |
| Hercule 7B * | 0.8125 | 0.4459 | 0.6563 | 0.6481 |
| M-PROMETHEUS 7B * | 0.8345 | 0.5381 | 0.8129 | 0.7466 |
| **Large (14B+ parameters)** | | | | |
| Qwen2.5-14B-Instruct | 0.9493 | 0.5921 | 0.8357 | 0.8030 |
| PROMETHEUS 2 8x7B † | 0.8851 | 0.3686 | 0.7748 | 0.7758 |
| M-PROMETHEUS 14B * | 0.8885 | 0.5553 | 0.8476 | 0.7935 |
| **Backbone Model Ablations** | | | | |
| Mistral-7B-v0.2-Instruct | 0.7804 | 0.4226 | 0.6675 | 0.6671 |
| EuroLLM-9B-Instruct | 0.8463 | 0.4423 | 0.6986 | 0.7853 |
| Aya-Expanse-8B | 0.8581 | 0.4373 | 0.7283 | 0.7704 |
| Qwen2.5-7B-Instruct | 0.8649 | 0.4619 | 0.8052 | 0.7880 |
| **Training Data Ablations** | | | | |
| MT Eval Data | 0.8716 | 0.4693 | 0.8224 | 0.7799 |
| Translated Data | | | | |
| 3 Non-English Langs | 0.8716 | 0.4619 | 0.7937 | 0.8084 |
| Multilingual Data | | | | |
| 3 Non-English Langs | 0.7918 | 0.5404 | 0.8075 | 0.7514 |
| 5 Non-English Langs | 0.8277 | 0.5430 | 0.8108 | 0.7677 |

Table 31: Accuracy on M-RewardBench (Korean) broken down by category.

| Model | M-RewardBench (Dutch) | | | |
| --- | --- | --- | --- | --- |
| | Chat | Chat Hard | Reasoning | Safety |
| **Proprietary Models** | | | | |
| GPT-4O | 0.9358 | 0.6683 | 0.9024 | 0.8601 |
| **Small (3B parameters)** | | | | |
| Qwen2.5-3B-Instruct | 0.8311 | 0.4693 | 0.6846 | 0.6963 |
| Glider 3B † | 0.6875 | 0.5074 | 0.8724 | 0.7188 |
| M-PROMETHEUS 3B * | 0.8851 | 0.4361 | 0.6969 | 0.7541 |
| **Medium (7B parameters)** | | | | |
| Qwen2.5-7B-Instruct | 0.9459 | 0.5258 | 0.8329 | 0.7677 |
| PROMETHEUS 2 7B † | 0.8615 | 0.4386 | 0.7297 | 0.7337 |
| Hercule 7B * | 0.8463 | 0.4545 | 0.6755 | 0.6664 |
| M-PROMETHEUS 7B * | 0.8986 | 0.4902 | 0.8294 | 0.8091 |
| **Large (14B+ parameters)** | | | | |
| Qwen2.5-14B-Instruct | 0.9493 | 0.6413 | 0.8455 | 0.8152 |
| PROMETHEUS 2 8x7B † | 0.9257 | 0.4251 | 0.8063 | 0.8077 |
| M-PROMETHEUS 14B * | 0.9257 | 0.5295 | 0.8329 | 0.8315 |
| **Backbone Model Ablations** | | | | |
| Mistral-7B-v0.2-Instruct | 0.8632 | 0.4226 | 0.6916 | 0.6997 |
| EuroLLM-9B-Instruct | 0.8885 | 0.4521 | 0.7573 | 0.8145 |
| Aya-Expanse-8B | 0.8750 | 0.4128 | 0.7696 | 0.7976 |
| Qwen2.5-7B-Instruct | 0.8953 | 0.4767 | 0.8406 | 0.8057 |
| **Training Data Ablations** | | | | |
| MT Eval Data | 0.9223 | 0.4644 | 0.8392 | 0.8139 |
| Translated Data | | | | |
|    3 Non-English Langs | 0.9088 | 0.4914 | 0.8451 | 0.8057 |
| Multilingual Data | | | | |
|    3 Non-English Langs | 0.8483 | 0.5492 | 0.8461 | 0.7912 |
|    5 Non-English Langs | 0.8615 | 0.5160 | 0.8406 | 0.8043 |

Table 32: Accuracy on M-RewardBench (Dutch) broken down by category.

| | M-RewardBench (Persian) | | | |
|---|---|---|---|---|
| Model | Chat | Chat Hard | Reasoning | Safety |
| **Proprietary Models** | | | | |
| GPT-4o | 0.9426 | 0.6069 | 0.8818 | 0.8397 |
| **Small (3B parameters)** | | | | |
| Qwen2.5-3B-Instruct | 0.7534 | 0.4300 | 0.6871 | 0.6569 |
| Glider 3B † | 0.5642 | 0.4840 | 0.7472 | 0.5414 |
| M-PROMETHEUS 3B * | 0.8801 | 0.4226 | 0.6741 | 0.7303 |
| **Medium (7B parameters)** | | | | |
| Qwen2.5-7B-Instruct | 0.9088 | 0.4939 | 0.8017 | 0.7486 |
| PROMETHEUS 2 7B † | 0.6824 | 0.4963 | 0.6839 | 0.6114 |
| Hercule 7B * | 0.8176 | 0.4214 | 0.6724 | 0.6467 |
| M-PROMETHEUS 7B * | 0.8750 | 0.4668 | 0.7976 | 0.7391 |
| **Large (14B+ parameters)** | | | | |
| Qwen2.5-14B-Instruct | 0.9291 | 0.5455 | 0.8259 | 0.7826 |
| PROMETHEUS 2 8x7B † | 0.8767 | 0.3882 | 0.7811 | 0.7269 |
| M-PROMETHEUS 14B * | 0.9054 | 0.4865 | 0.8378 | 0.7799 |
| **Backbone Model Ablations** | | | | |
| Mistral-7B-v0.2-Instruct | 0.6706 | 0.4681 | 0.6766 | 0.5469 |
| EuroLLM-9B-Instruct | 0.7703 | 0.4029 | 0.6972 | 0.7004 |
| Aya-Expanse-8B | 0.8649 | 0.4349 | 0.7503 | 0.7541 |
| Qwen2.5-7B-Instruct | 0.8716 | 0.4595 | 0.8150 | 0.7595 |
| **Training Data Ablations** | | | | |
| MT Eval Data | 0.8649 | 0.4251 | 0.8210 | 0.7731 |
| Translated Data | | | | |
| 3 Non-English Langs | 0.8682 | 0.4373 | 0.8276 | 0.7704 |
| Multilingual Data | | | | |
| 3 Non-English Langs | 0.7928 | 0.5350 | 0.8188 | 0.7516 |
| 5 Non-English Langs | 0.8378 | 0.5086 | 0.8168 | 0.7609 |

Table 33: Accuracy on M-RewardBench (Persian) broken down by category.

| | M-RewardBench (Polish) | | | |
|---|---|---|---|---|
| Model | Chat | Chat Hard | Reasoning | Safety |
| **Proprietary Models** | | | | |
| GPT-4O | 0.9493 | 0.6732 | 0.8923 | 0.8587 |
| **Small (3B parameters)** | | | | |
| Qwen2.5-3B-Instruct | 0.8277 | 0.3870 | 0.7042 | 0.6834 |
| Glider 3B † | 0.6064 | 0.4619 | 0.8406 | 0.6637 |
| M-PROMETHEUS 3B * | 0.8818 | 0.4509 | 0.6769 | 0.7391 |
| **Medium (7B parameters)** | | | | |
| Qwen2.5-7B-Instruct | 0.9527 | 0.5160 | 0.8238 | 0.8016 |
| PROMETHEUS 2 7B † | 0.8530 | 0.4582 | 0.6850 | 0.6977 |
| Hercule 7B * | 0.8412 | 0.4054 | 0.6664 | 0.6372 |
| M-PROMETHEUS 7B * | 0.8851 | 0.5319 | 0.8154 | 0.7867 |
| **Large (14B+ parameters)** | | | | |
| Qwen2.5-14B-Instruct | 0.9358 | 0.5995 | 0.8402 | 0.7962 |
| PROMETHEUS 2 8x7B † | 0.9155 | 0.4275 | 0.8052 | 0.7887 |
| M-PROMETHEUS 14B * | 0.8885 | 0.5491 | 0.8413 | 0.8288 |
| **Backbone Model Ablations** | | | | |
| Mistral-7B-v0.2-Instruct | 0.8176 | 0.4005 | 0.6794 | 0.6712 |
| EuroLLM-9B-Instruct | 0.8699 | 0.4668 | 0.7325 | 0.8084 |
| Aya-Expanse-8B | 0.8818 | 0.4300 | 0.7622 | 0.7874 |
| Qwen2.5-7B-Instruct | 0.8784 | 0.4619 | 0.8395 | 0.7894 |
| **Training Data Ablations** | | | | |
| MT Eval Data | 0.8885 | 0.4521 | 0.8497 | 0.7976 |
| Translated Data | | | | |
|   3 Non-English Langs | 0.8953 | 0.4705 | 0.8126 | 0.8016 |
| Multilingual Data | | | | |
|   3 Non-English Langs | 0.8167 | 0.5310 | 0.8202 | 0.7706 |
|   5 Non-English Langs | 0.8716 | 0.5283 | 0.8252 | 0.7799 |

Table 34: Accuracy on M-RewardBench (Polish) broken down by category.

| | M-RewardBench (Portuguese) | | | |
|---|---|---|---|---|
| Model | Chat | Chat Hard | Reasoning | Safety |
| **Proprietary Models** | | | | |
| GPT-4O | 0.9527 | 0.6560 | 0.9063 | 0.8750 |
| **Small (3B parameters)** | | | | |
| Qwen2.5-3B-Instruct | 0.8345 | 0.4791 | 0.7227 | 0.7052 |
| Glider 3B † | 0.7044 | 0.5221 | 0.8825 | 0.7711 |
| M-PROMETHEUS 3B * | 0.8750 | 0.4189 | 0.6864 | 0.7758 |
| **Medium (7B parameters)** | | | | |
| Qwen2.5-7B-Instruct | 0.9527 | 0.5381 | 0.8280 | 0.7935 |
| PROMETHEUS 2 7B † | 0.8716 | 0.4631 | 0.7241 | 0.7724 |
| Hercule 7B * | 0.8395 | 0.4558 | 0.6895 | 0.6753 |
| M-PROMETHEUS 7B * | 0.8953 | 0.5504 | 0.8535 | 0.8043 |
| **Large (14B+ parameters)** | | | | |
| Qwen2.5-14B-Instruct | 0.9527 | 0.6609 | 0.8441 | 0.8356 |
| PROMETHEUS 2 8x7B † | 0.9155 | 0.4459 | 0.8112 | 0.8132 |
| M-PROMETHEUS 14B * | 0.9088 | 0.5504 | 0.8451 | 0.8370 |
| **Backbone Model Ablations** | | | | |
| Mistral-7B-v0.2-Instruct | 0.8784 | 0.4005 | 0.6934 | 0.6943 |
| EuroLLM-9B-Instruct | 0.9003 | 0.4816 | 0.7643 | 0.8247 |
| Aya-Expanse-8B | 0.8818 | 0.4644 | 0.7710 | 0.8179 |
| Qwen2.5-7B-Instruct | 0.9206 | 0.5160 | 0.8605 | 0.8152 |
| **Training Data Ablations** | | | | |
| MT Eval Data | 0.9155 | 0.5012 | 0.8476 | 0.8057 |
| Translated Data | | | | |
| 3 Non-English Langs | 0.9088 | 0.5184 | 0.8573 | 0.8125 |
| Multilingual Data | | | | |
| 3 Non-English Langs | 0.8200 | 0.5849 | 0.8438 | 0.7979 |
| 5 Non-English Langs | 0.8446 | 0.5430 | 0.8469 | 0.7826 |

Table 35: Accuracy on M-RewardBench (Portuguese) broken down by category.

| | **M-RewardBench (Romanian)** | | | |
|---|---|---|---|---|
| **Model** | Chat | Chat Hard | Reasoning | Safety |
| **Proprietary Models** | | | | |
| GPT-4O | 0.9493 | 0.6880 | 0.8909 | 0.8519 |
| **Small (3B parameters)** | | | | |
| Qwen2.5-3B-Instruct | 0.8311 | 0.4410 | 0.7136 | 0.7018 |
| Glider 3B † | 0.6098 | 0.4767 | 0.8507 | 0.5802 |
| M-PROMETHEUS 3B * | 0.8885 | 0.4300 | 0.6829 | 0.7459 |
| **Medium (7B parameters)** | | | | |
| Qwen2.5-7B-Instruct | 0.9223 | 0.5184 | 0.8353 | 0.7582 |
| PROMETHEUS 2 7B † | 0.8361 | 0.4681 | 0.6857 | 0.6800 |
| Hercule 7B * | 0.8311 | 0.4386 | 0.6878 | 0.6236 |
| M-PROMETHEUS 7B * | 0.8851 | 0.4644 | 0.8182 | 0.7921 |
| **Large (14B+ parameters)** | | | | |
| Qwen2.5-14B-Instruct | 0.9392 | 0.5995 | 0.8336 | 0.8016 |
| PROMETHEUS 2 8x7B † | 0.9291 | 0.3943 | 0.7962 | 0.7758 |
| M-PROMETHEUS 14B * | 0.9122 | 0.5000 | 0.8276 | 0.8193 |
| **Backbone Model Ablations** | | | | |
| Mistral-7B-v0.2-Instruct | 0.8446 | 0.4152 | 0.6853 | 0.6705 |
| EuroLLM-9B-Instruct | 0.8885 | 0.4423 | 0.7552 | 0.8003 |
| Aya-Expanse-8B | 0.8733 | 0.4324 | 0.7815 | 0.8003 |
| Qwen2.5-7B-Instruct | 0.8818 | 0.4570 | 0.8451 | 0.7962 |
| **Training Data Ablations** | | | | |
| MT Eval Data | 0.8919 | 0.4668 | 0.8416 | 0.7962 |
| Translated Data | | | | |
| 3 Non-English Langs | 0.9003 | 0.4816 | 0.8413 | 0.7880 |
| Multilingual Data | | | | |
| 3 Non-English Langs | 0.8495 | 0.5539 | 0.8408 | 0.7734 |
| 5 Non-English Langs | 0.8649 | 0.5233 | 0.8381 | 0.7908 |

Table 36: Accuracy on M-RewardBench (Romanian) broken down by category.

| Model | M-RewardBench (Russian) | | | |
| | Chat | Chat Hard | Reasoning | Safety |
|---|---|---|---|---|
| **Proprietary Models** | | | | |
| GPT-4O | 0.9662 | 0.6732 | 0.8990 | 0.8709 |
| **Small (3B parameters)** | | | | |
| Qwen2.5-3B-Instruct | 0.8176 | 0.4926 | 0.7119 | 0.6916 |
| Glider 3B † | 0.6824 | 0.4988 | 0.8780 | 0.7357 |
| M-PROMETHEUS 3B * | 0.8615 | 0.4509 | 0.6979 | 0.7554 |
| **Medium (7B parameters)** | | | | |
| Qwen2.5-7B-Instruct | 0.9392 | 0.5651 | 0.8360 | 0.7894 |
| PROMETHEUS 2 7B † | 0.8429 | 0.4803 | 0.6874 | 0.7113 |
| Hercule 7B * | 0.8429 | 0.4349 | 0.6864 | 0.6562 |
| M-PROMETHEUS 7B * | 0.9054 | 0.5553 | 0.8409 | 0.7867 |
| **Large (14B+ parameters)** | | | | |
| Qwen2.5-14B-Instruct | 0.9493 | 0.6339 | 0.8399 | 0.8084 |
| PROMETHEUS 2 8x7B † | 0.9223 | 0.4140 | 0.8255 | 0.7976 |
| M-PROMETHEUS 14B * | 0.9122 | 0.5577 | 0.8371 | 0.8342 |
| **Backbone Model Ablations** | | | | |
| Mistral-7B-v0.2-Instruct | 0.8716 | 0.4079 | 0.6864 | 0.6929 |
| EuroLLM-9B-Instruct | 0.8885 | 0.4619 | 0.7668 | 0.8166 |
| Aya-Expanse-8B | 0.8699 | 0.4349 | 0.7857 | 0.7989 |
| Qwen2.5-7B-Instruct | 0.9020 | 0.5061 | 0.8521 | 0.8152 |
| **Training Data Ablations** | | | | |
| MT Eval Data | 0.8851 | 0.5111 | 0.8441 | 0.8111 |
| Translated Data | | | | |
|   3 Non-English Langs | 0.8970 | 0.4717 | 0.8483 | 0.8179 |
| Multilingual Data | | | | |
|   3 Non-English Langs | 0.8295 | 0.5874 | 0.8561 | 0.7761 |
|   5 Non-English Langs | 0.8345 | 0.5553 | 0.8458 | 0.7894 |

Table 37: Accuracy on M-RewardBench (Russian) broken down by category.

| Model | M-RewardBench (Spanish) | | | |
| | Chat | Chat Hard | Reasoning | Safety |
|---|---|---|---|---|
| **Proprietary Models** | | | | |
| GPT-4O | 0.9561 | 0.7052 | 0.9035 | 0.8791 |
| **Small (3B parameters)** | | | | |
| Qwen2.5-3B-Instruct | 0.8193 | 0.4558 | 0.6955 | 0.7045 |
| Glider 3B † | 0.6588 | 0.5135 | 0.8843 | 0.7602 |
| M-PROMETHEUS 3B * | 0.8851 | 0.4312 | 0.6811 | 0.7785 |
| **Medium (7B parameters)** | | | | |
| Qwen2.5-7B-Instruct | 0.9628 | 0.5381 | 0.8245 | 0.7758 |
| PROMETHEUS 2 7B † | 0.8818 | 0.4484 | 0.7150 | 0.7615 |
| Hercule 7B * | 0.8361 | 0.4644 | 0.6902 | 0.6692 |
| M-PROMETHEUS 7B * | 0.9189 | 0.5197 | 0.8427 | 0.8016 |
| **Large (14B+ parameters)** | | | | |
| Qwen2.5-14B-Instruct | 0.9527 | 0.6536 | 0.8524 | 0.8071 |
| PROMETHEUS 2 8x7B † | 0.9392 | 0.4496 | 0.8280 | 0.7976 |
| M-PROMETHEUS 14B * | 0.9172 | 0.5418 | 0.8343 | 0.8268 |
| **Backbone Model Ablations** | | | | |
| Mistral-7B-v0.2-Instruct | 0.8818 | 0.4373 | 0.6888 | 0.6957 |
| EuroLLM-9B-Instruct | 0.8953 | 0.4644 | 0.7668 | 0.8207 |
| Aya-Expanse-8B | 0.8581 | 0.4545 | 0.7678 | 0.8071 |
| Qwen2.5-7B-Instruct | 0.8936 | 0.5184 | 0.8479 | 0.8193 |
| **Training Data Ablations** | | | | |
| MT Eval Data | 0.9054 | 0.4939 | 0.8455 | 0.8159 |
| Translated Data | | | | |
|   3 Non-English Langs | 0.8986 | 0.5209 | 0.8542 | 0.8152 |
| Multilingual Data | | | | |
|   3 Non-English Langs | 0.8274 | 0.5663 | 0.8507 | 0.7953 |
|   5 Non-English Langs | 0.8615 | 0.5528 | 0.8266 | 0.8003 |

Table 38: Accuracy on M-RewardBench (Spanish) broken down by category.

| Model | M-RewardBench (Turkish) | | | |
| --- | --- | --- | --- | --- |
| | Chat | Chat Hard | Reasoning | Safety |
| **Proprietary Models** | | | | |
| GPT-4O | 0.9291 | 0.6118 | 0.8902 | 0.8410 |
| **Small (3B parameters)** | | | | |
| Qwen2.5-3B-Instruct | 0.7703 | 0.4287 | 0.6955 | 0.6501 |
| Glider 3B † | 0.6250 | 0.4435 | 0.8374 | 0.4851 |
| M-PROMETHEUS 3B * | 0.8902 | 0.4447 | 0.6668 | 0.7215 |
| **Medium (7B parameters)** | | | | |
| Qwen2.5-7B-Instruct | 0.9257 | 0.4914 | 0.8035 | 0.7582 |
| PROMETHEUS 2 7B † | 0.7787 | 0.4361 | 0.6941 | 0.6651 |
| Hercule 7B * | 0.8328 | 0.4287 | 0.6759 | 0.6121 |
| M-PROMETHEUS 7B * | 0.8480 | 0.4398 | 0.7972 | 0.7779 |
| **Large (14B+ parameters)** | | | | |
| Qwen2.5-14B-Instruct | 0.9392 | 0.5528 | 0.8227 | 0.7806 |
| PROMETHEUS 2 8x7B † | 0.8395 | 0.3968 | 0.8042 | 0.7418 |
| M-PROMETHEUS 14B * | 0.9139 | 0.4791 | 0.8189 | 0.8091 |
| **Backbone Model Ablations** | | | | |
| Mistral-7B-v0.2-Instruct | 0.7973 | 0.4201 | 0.6752 | 0.5455 |
| EuroLLM-9B-Instruct | 0.8936 | 0.4668 | 0.7479 | 0.7908 |
| Aya-Expanse-8B | 0.8851 | 0.4103 | 0.7752 | 0.7853 |
| Qwen2.5-7B-Instruct | 0.8682 | 0.4717 | 0.8308 | 0.7663 |
| **Training Data Ablations** | | | | |
| MT Eval Data | 0.8750 | 0.4472 | 0.8168 | 0.7731 |
| Translated Data | | | | |
|   3 Non-English Langs | 0.8986 | 0.4398 | 0.8178 | 0.7894 |
| Multilingual Data | | | | |
|   3 Non-English Langs | 0.8103 | 0.5251 | 0.8310 | 0.7584 |
|   5 Non-English Langs | 0.8446 | 0.5012 | 0.8066 | 0.7921 |

Table 39: Accuracy on M-RewardBench (Turkish) broken down by category.

| Model | M-RewardBench (Ukrainian) | | | |
|---|---|---|---|---|
| | Chat | Chat Hard | Reasoning | Safety |
| **Proprietary Models** | | | | |
| GPT-4o | 0.9561 | 0.6437 | 0.8913 | 0.8519 |
| **Small (3B parameters)** | | | | |
| Qwen2.5-3B-Instruct | 0.7804 | 0.4275 | 0.7000 | 0.7011 |
| Glider 3B † | 0.6014 | 0.4263 | 0.8517 | 0.6624 |
| M-PROMETHEUS 3B * | 0.8446 | 0.4300 | 0.6986 | 0.7473 |
| **Medium (7B parameters)** | | | | |
| Qwen2.5-7B-Instruct | 0.9189 | 0.5233 | 0.8325 | 0.7867 |
| PROMETHEUS 2 7B † | 0.7821 | 0.4619 | 0.7035 | 0.7018 |
| Hercule 7B * | 0.8226 | 0.4300 | 0.6857 | 0.6556 |
| M-PROMETHEUS 7B * | 0.8986 | 0.5049 | 0.8245 | 0.7683 |
| **Large (14B+ parameters)** | | | | |
| Qwen2.5-14B-Instruct | 0.9257 | 0.5676 | 0.8434 | 0.8125 |
| PROMETHEUS 2 8x7B † | 0.9274 | 0.4189 | 0.8196 | 0.7976 |
| M-PROMETHEUS 14B * | 0.9189 | 0.5381 | 0.8385 | 0.8084 |
| **Backbone Model Ablations** | | | | |
| Mistral-7B-v0.2-Instruct | 0.8480 | 0.4201 | 0.6962 | 0.6766 |
| EuroLLM-9B-Instruct | 0.8767 | 0.4201 | 0.7556 | 0.8166 |
| Aya-Expanse-8B | 0.8750 | 0.4079 | 0.7636 | 0.8003 |
| Qwen2.5-7B-Instruct | 0.8750 | 0.4668 | 0.8497 | 0.7826 |
| **Training Data Ablations** | | | | |
| MT Eval Data | 0.8784 | 0.4767 | 0.8371 | 0.7690 |
| Translated Data | | | | |
|   3 Non-English Langs | 0.8716 | 0.4791 | 0.8378 | 0.7935 |
| Multilingual Data | | | | |
|   3 Non-English Langs | 0.8005 | 0.5505 | 0.8594 | 0.7610 |
|   5 Non-English Langs | 0.8412 | 0.5356 | 0.8423 | 0.7758 |

Table 40: Accuracy on M-RewardBench (Ukrainian) broken down by category.

| | M-RewardBench (Vietnamese) | | | |
|---|---|---|---|---|
| **Model** | Chat | Chat Hard | Reasoning | Safety |
| **Proprietary Models** | | | | |
| GPT-4O | 0.9595 | 0.6732 | 0.8997 | 0.8601 |
| **Small (3B parameters)** | | | | |
| Qwen2.5-3B-Instruct | 0.8446 | 0.4287 | 0.6990 | 0.7126 |
| Glider 3B † | 0.5321 | 0.4840 | 0.7979 | 0.5870 |
| M-PROMETHEUS 3B * | 0.8632 | 0.4509 | 0.6836 | 0.7758 |
| **Medium (7B parameters)** | | | | |
| Qwen2.5-7B-Instruct | 0.9595 | 0.5430 | 0.8556 | 0.7812 |
| PROMETHEUS 2 7B † | 0.7669 | 0.4398 | 0.7049 | 0.6861 |
| Hercule 7B * | 0.7973 | 0.4337 | 0.6731 | 0.6556 |
| M-PROMETHEUS 7B * | 0.8953 | 0.5344 | 0.8360 | 0.7921 |
| **Large (14B+ parameters)** | | | | |
| Qwen2.5-14B-Instruct | 0.9493 | 0.6314 | 0.8503 | 0.8098 |
| PROMETHEUS 2 8x7B † | 0.8970 | 0.3919 | 0.8315 | 0.7534 |
| M-PROMETHEUS 14B * | 0.9020 | 0.5356 | 0.8455 | 0.8302 |
| **Backbone Model Ablations** | | | | |
| Mistral-7B-v0.2-Instruct | 0.7652 | 0.4373 | 0.6850 | 0.5251 |
| EuroLLM-9B-Instruct | 0.7078 | 0.4128 | 0.7367 | 0.7099 |
| Aya-Expanse-8B | 0.9054 | 0.3907 | 0.7724 | 0.8071 |
| Qwen2.5-7B-Instruct | 0.8767 | 0.4926 | 0.8497 | 0.8152 |
| **Training Data Ablations** | | | | |
| MT Eval Data | 0.8801 | 0.4840 | 0.8507 | 0.8234 |
| Translated Data | | | | |
|    3 Non-English Langs | 0.8919 | 0.5061 | 0.8343 | 0.8179 |
| Multilingual Data | | | | |
|    3 Non-English Langs | 0.7788 | 0.5935 | 0.8409 | 0.7814 |
|    5 Non-English Langs | 0.8277 | 0.5553 | 0.8255 | 0.7867 |

Table 41: Accuracy on M-RewardBench (Vietnamese) broken down by category.

| Model | M-RewardBench (Chinese (Simplified)) | | | |
| | Chat | Chat Hard | Reasoning | Safety |
|---|---|---|---|---|
| **Proprietary Models** | | | | |
| GPT-4o | 0.9561 | 0.6314 | 0.8899 | 0.8723 |
| **Small (3B parameters)** | | | | |
| Qwen2.5-3B-Instruct | 0.8277 | 0.4644 | 0.7070 | 0.6957 |
| Glider 3B † | 0.6216 | 0.4889 | 0.8524 | 0.7228 |
| M-PROMETHEUS 3B * | 0.8581 | 0.4681 | 0.6902 | 0.7649 |
| **Medium (7B parameters)** | | | | |
| Qwen2.5-7B-Instruct | 0.9426 | 0.5528 | 0.8364 | 0.7880 |
| PROMETHEUS 2 7B † | 0.8176 | 0.4545 | 0.7164 | 0.6943 |
| Hercule 7B * | 0.8091 | 0.4521 | 0.6748 | 0.6413 |
| M-PROMETHEUS 7B * | 0.8682 | 0.5676 | 0.8311 | 0.8003 |
| **Large (14B+ parameters)** | | | | |
| Qwen2.5-14B-Instruct | 0.9561 | 0.6339 | 0.8531 | 0.8315 |
| PROMETHEUS 2 8x7B † | 0.8919 | 0.4017 | 0.7860 | 0.7901 |
| M-PROMETHEUS 14B * | 0.8919 | 0.5811 | 0.8388 | 0.8193 |
| **Backbone Model Ablations** | | | | |
| Mistral-7B-v0.2-Instruct | 0.8615 | 0.4029 | 0.6969 | 0.7024 |
| EuroLLM-9B-Instruct | 0.8530 | 0.4717 | 0.7374 | 0.8193 |
| Aya-Expanse-8B | 0.8649 | 0.4251 | 0.7476 | 0.7880 |
| Qwen2.5-7B-Instruct | 0.8615 | 0.5405 | 0.8556 | 0.8139 |
| **Training Data Ablations** | | | | |
| MT Eval Data | 0.9054 | 0.5307 | 0.8476 | 0.8084 |
| Translated Data | | | | |
| 3 Non-English Langs | 0.8953 | 0.4865 | 0.8273 | 0.8139 |
| Multilingual Data | | | | |
| 3 Non-English Langs | 0.7778 | 0.5875 | 0.8254 | 0.7856 |
| 5 Non-English Langs | 0.8176 | 0.5651 | 0.8154 | 0.7935 |

Table 42: Accuracy on M-RewardBench (Chinese (Simplified)) broken down by category.

| Model | M-RewardBench (Chinese (Traditional)) | | | |
| | Chat | Chat Hard | Reasoning | Safety |
|---|---|---|---|---|
| **Proprietary Models** | | | | |
| GPT-4O | 0.9459 | 0.6511 | 0.8825 | 0.8451 |
| **Small (3B parameters)** | | | | |
| Qwen2.5-3B-Instruct | 0.8699 | 0.4300 | 0.7087 | 0.6902 |
| Glider 3B † | 0.6976 | 0.4828 | 0.7402 | 0.6698 |
| M-PROMETHEUS 3B * | 0.8378 | 0.4545 | 0.6811 | 0.7432 |
| **Medium (7B parameters)** | | | | |
| Qwen2.5-7B-Instruct | 0.9223 | 0.5455 | 0.8094 | 0.7908 |
| PROMETHEUS 2 7B † | 0.7703 | 0.4619 | 0.7245 | 0.7126 |
| Hercule 7B * | 0.8074 | 0.4656 | 0.6703 | 0.6522 |
| M-PROMETHEUS 7B * | 0.8581 | 0.5356 | 0.8346 | 0.8111 |
| **Large (14B+ parameters)** | | | | |
| Qwen2.5-14B-Instruct | 0.9493 | 0.6069 | 0.8322 | 0.8342 |
| PROMETHEUS 2 8x7B † | 0.9105 | 0.4005 | 0.7804 | 0.7996 |
| M-PROMETHEUS 14B * | 0.8750 | 0.5381 | 0.8105 | 0.8336 |
| **Backbone Model Ablations** | | | | |
| Mistral-7B-v0.2-Instruct | 0.7551 | 0.4226 | 0.6843 | 0.7167 |
| EuroLLM-9B-Instruct | 0.8412 | 0.4496 | 0.7360 | 0.8261 |
| Aya-Expanse-8B | 0.8108 | 0.4300 | 0.7448 | 0.8057 |
| Qwen2.5-7B-Instruct | 0.8446 | 0.5086 | 0.8458 | 0.8207 |
| **Training Data Ablations** | | | | |
| MT Eval Data | 0.8649 | 0.4988 | 0.8469 | 0.8234 |
| Translated Data | | | | |
| 3 Non-English Langs | 0.8750 | 0.4791 | 0.8462 | 0.8247 |
| Multilingual Data | | | | |
| 3 Non-English Langs | 0.7578 | 0.6036 | 0.8295 | 0.7925 |
| 5 Non-English Langs | 0.8108 | 0.5553 | 0.8350 | 0.8003 |

Table 43: Accuracy on M-RewardBench (Chinese (Traditional)) broken down by category.

## C.3  RewardBench

| | RewardBench | | | |
|---|---|---|---|---|
| Model | Chat | Chat Hard | Reasoning | Safety |
| **Proprietary Models** | | | | |
| GPT-4O | 0.9553 | 0.7259 | 0.8811 | 0.8760 |
| **Small (3B parameters)** | | | | |
| Qwen2.5-3B-Instruct | 0.8687 | 0.5186 | 0.7149 | 0.6737 |
| Glider 3B † | 0.8156 | 0.4287 | 0.7703 | 0.7161 |
| M-PROMETHEUS 3B * | 0.8785 | 0.4496 | 0.7851 | 0.7305 |
| **Medium (7B parameters)** | | | | |
| Qwen2.5-7B-Instruct | 0.9637 | 0.5713 | 0.7986 | 0.7957 |
| PROMETHEUS 2 7B † | 0.8550 | 0.4910 | 0.7710 | 0.7650 |
| Hercule 7B * | 0.7905 | 0.5526 | 0.6243 | 0.7469 |
| M-PROMETHEUS 7B * | 0.9078 | 0.5373 | 0.8419 | 0.8284 |
| **Large (14B+ parameters)** | | | | |
| Qwen2.5-14B-Instruct | 0.9525 | 0.6645 | 0.8365 | 0.8428 |
| PROMETHEUS 2 8x7B † | 0.9330 | 0.4671 | 0.8101 | 0.7522 |
| M-PROMETHEUS 14B * | 0.9358 | 0.5899 | 0.8514 | 0.8477 |
| **Backbone Model Ablations** | | | | |
| Mistral-7B-v0.2-Instruct | 0.8883 | 0.4474 | 0.7554 | 0.7422 |
| EuroLLM-9B-Instruct | 0.9008 | 0.4693 | 0.8419 | 0.7956 |
| Aya-Expanse-8B | 0.8813 | 0.4912 | 0.8135 | 0.8265 |
| Qwen2.5-7B-Instruct | 0.9050 | 0.5110 | 0.8351 | 0.8583 |
| **Training Data Ablations** | | | | |
| MT Eval Data | 0.9078 | 0.5351 | 0.8216 | 0.8476 |
| Translated Data | | | | |
| 3 Non-English Langs | 0.9050 | 0.5307 | 0.8284 | 0.8431 |
| Multilingual Data | | | | |
| 3 Non-English Langs | 0.8687 | 0.5504 | 0.8473 | 0.8756 |
| 5 Non-English Langs | 0.9106 | 0.5417 | 0.8514 | 0.8466 |

Table 44: Accuracy on RewardBench broken down by category.

## C.4 LitEval

| Model | LitEval | | | |
| | German→English | English→German | English→Chinese | German→Chinese |
|---|---|---|---|---|
| **Proprietary Models** | | | | |
| GPT-4O | 0.2680 | 0.4896 | 0.4189 | 0.4012 |
| **Small (3B parameters)** | | | | |
| Qwen2.5-3B-Instruct | 0.1195 | 0.2676 | 0.1208 | 0.1074 |
| Glider 3B † | 0.0992 | 0.2034 | 0.2285 | 0.1815 |
| M-PROMETHEUS 3B * | 0.2203 | 0.5062 | 0.4411 | 0.4624 |
| **Medium (7B parameters)** | | | | |
| Qwen2.5-7B-Instruct | 0.1258 | 0.3289 | 0.1799 | 0.0743 |
| PROMETHEUS 2 7B † | 0.1973 | 0.0854 | 0.0996 | 0.1184 |
| Hercule 7B * | 0.2623 | 0.3340 | 0.3872 | 0.4231 |
| M-PROMETHEUS 7B * | 0.2001 | 0.5321 | 0.4661 | 0.5427 |
| **Large (14B+ parameters)** | | | | |
| Qwen2.5-14B-Instruct | 0.2764 | 0.3901 | 0.2662 | 0.3102 |
| PROMETHEUS 2 8x7B † | 0.2479 | 0.3667 | 0.2561 | 0.4034 |
| M-PROMETHEUS 14B * | 0.2979 | 0.5702 | 0.4822 | 0.5658 |
| **Backbone Model Ablations** | | | | |
| Mistral-7B-v0.2-Instruct | 0.1135 | 0.0384 | 0.0139 | 0.1328 |
| EuroLLM-9B-Instruct | 0.1480 | 0.3767 | 0.1506 | 0.2987 |
| Aya-Expanse-8B | 0.0989 | 0.3293 | 0.2449 | 0.3446 |
| Qwen2.5-7B-Instruct | 0.2074 | 0.2850 | 0.3184 | 0.3240 |
| **Training Data Ablations** | | | | |
| MT Eval Data | 0.2541 | 0.3974 | 0.4786 | 0.5585 |
| Translated Data | | | | |
|   3 Non-English Langs | 0.1962 | 0.2266 | 0.1785 | 0.2872 |
| Multilingual Data | | | | |
|   3 Non-English Langs | 0.1245 | 0.3424 | 0.3779 | 0.4201 |
|   5 Non-English Langs | 0.1242 | 0.3856 | 0.4022 | 0.4369 |

Table 45: Kendall correlation on LitEval broken down by language pair.

## C.5 QAD

| Model | QAD | | |
| | French | Chinese | Hindi |
|---|---|---|---|
| **Small (3B parameters)** | | | |
| Qwen2.5-3B-Instruct | 50.86 | 47.99 | 64.01 |
| Glider 3B † | 56.02 | 46.12 | 69.49 |
| M-PROMETHEUS 3B * | 60.77 | 51.44 | 76.91 |
| **Medium (7B parameters)** | | | |
| Qwen2.5-7B-Instruct | 52.73 | 46.41 | 68.51 |
| PROMETHEUS 2 7B † | 57.85 | 51.58 | 78.21 |
| Hercule 7B * | 61.04 | 56.02 | 77.52 |
| M-PROMETHEUS 7B * | 65.58 | 55.30 | 78.21 |
| **Large (14B+ parameters)** | | | |
| Qwen2.5-14B-Instruct | 53.31 | 48.99 | 61.59 |
| PROMETHEUS 2 8x7B † | 61.59 | 51.87 | 74.91 |
| M-PROMETHEUS 14B * | 58.41 | 53.02 | 81.79 |
| **Backbone Model Ablations** | | | |
| Mistral-7B-v0.2-Instruct | 56.44 | 48.71 | 80.29 |
| EuroLLM-9B-Instruct | 59.94 | 51.01 | 78.50 |
| Aya-Expanse-8B | 52.88 | 48.99 | 79.74 |
| Qwen2.5-7B-Instruct | 64.14 | 57.15 | 80.20 |
| **Training Data Ablations** | | | |
| MT Eval Data | 57.85 | 49.28 | 71.99 |
| Translated Data | | | |
|    3 Non-English Langs | 63.61 | 55.16 | 80.65 |
| Multilingual Data | | | |
|    3 Non-English Langs | 66.10 | 57.71 | 82.30 |
|    5 Non-English Langs | 64.93 | 58.27 | 75.12 |

Table 46: Win rate on QAD broken down by language.

## D QAD Results with Gemma-2-2B-IT

In Table 47, we present QAD results when using judges to improve the outputs of Gemma-2-2B-IT, as opposed to Qwen.

|                          | QAD    |         |       |
|--------------------------|--------|---------|-------|
| Model                    | French | Chinese | Hindi |
| **Small (3B parameters)** |        |         |       |
| Qwen2.5-3B-Instruct      | 50.72  | 49.28   | 48.27 |
| Glider 3B †              | 52.73  | 47.27   | 53.16 |
| M-PROMETHEUS 3B *        | 58.97  | 53.02   | 53.02 |
| **Medium (7B parameters)** |      |         |       |
| Qwen2.5-7B-Instruct      | 53.88  | 48.56   | 50.72 |
| PROMETHEUS 2 7B †        | 59.25  | 56.16   | 58.55 |
| Hercule 7B *             | 57.29  | 54.16   | 61.18 |
| M-PROMETHEUS 7B *        | 59.25  | 47.99   | 59.39 |
| **Large (14B+ parameters)** |     |         |       |
| Qwen2.5-14B-Instruct     | 54.16  | 50.43   | 50.29 |
| PROMETHEUS 2 8x7B †      | 57.29  | 50.43   | 57.57 |
| M-PROMETHEUS 14B *       | 55.59  | 49.57   | 54.45 |
| **Backbone Model Ablations** |    |         |       |
| Mistral-7B-v0.2-Instruct | 58.55  | 52.01   | 54.45 |
| EuroLLM-9B-Instruct      | 51.44  | 50.86   | 52.88 |
| Aya-Expanse-8B           | 55.59  | 48.56   | 55.02 |
| Qwen2.5-7B-Instruct      | 56.44  | 50.29   | 57.57 |
| **Training Data Ablations** |     |         |       |
| MT Eval Data             | 54.16  | 47.70   | 54.59 |
| Translated Data          |        |         |       |
| 3 Non-English Langs      | 55.30  | 50.58   | 60.35 |
| Multilingual Data        |        |         |       |
| 3 Non-English Langs      | 60.49  | 55.87   | 57.29 |
| 5 Non-English Langs      | 60.35  | 56.02   | 61.45 |

Table 47: Win rate on QAD on Gemma-2-2B-IT broken down by language.

