# OpenReview forum: "M-Prometheus: A Suite of Open Multilingual LLM Judges"
_colmweb.org/COLM/2025/Conference — COLM 2025_

### Official Review · Reviewer_n3M7 · 2025-05-12

**Rating:** 6
**Confidence:** 4
**Ethics Flag:** 1

**Summary:**

The paper proposes multilingual LLM judges, which are trained on synthetic data generated in six languages. The authors show that their trained models perform better than other open-weights LLMs and LLM-judges on multilingual evaluation benchmarks, evaluation for literary machine translation, and on downstream performance when used for quality-aware decoding for improving generations of Qwen2.5-3B-Instruct. Through ablations, the authors show that the backbone model used for training the multilingual judge is an important consideration and training on translated data is not as effective as synthetically generated multilingual data. The paper provides a detailed list of experiments to test the effectiveness of their approach and the results to a large extent provide evidence in favor of that. I am reserved about the research contributions of the work and some claims that seem exaggerated.

**Reasons To Accept:**

- I think empirically, the paper to the most part is rigorous. The authors test the leading (multilingual as well as monolingual) meta-evaluation benchmarks and also include downstream applications of these models such as evaluation for Literary Machine Translation and Quality-Aware-Decoding
- The results also to the most part show the effectiveness of their method, specifically on the LitEval and QAD metrics. I am less convinced about the results on MM-Eval benchmark, which I discuss in the following section.

**Reasons To Reject:**

- In my opinion, the research contribution of the present work is a bit limited. From the technical aspect, the training method is an extension of prior works to a multilingual setting by generating synthetic data in non-English languages mainly through prompting. While I understand that not every paper needs to offer a technical contribution and strong empirical analysis can be at least as much important, I feel the latter to also not be a strong aspect of this work, which I detail in the points below.

- The claims about some of the results in the paper seem slightly exaggerated. On MM-Eval benchmark the authors state that their models are particularly strong on the Safety, Linguistics, and Language Hallucinations categories. However, on examining the results in Figure 3, for the Linguistics category their models perform either as good as the base model or worse when considering 7B and 14B parameter sizes. Only at 3B their models perform slightly better than the base model. Similarly, for the Language Hallucinations categor the gains are marginal as well. Only in the Safety Category the authors find very strong improvements with their method, where the base models perform particularly poorly. This also makes the overall improvements by their method on the MM-Eval benchmark seem overly inflated. I am not sure **if these results indicate the effectiveness of their method in general or specific to particular tasks?** Similarly, authors show strong performance on LitEval metric, but that again might be due to the presence of Machine Translation Eval data in their training data.

- The authors use the term *natively multilingual* at many places in the paper for the synthetic data generated in different languages, which I feel is a little misleading. I understand that authors use this term to distinguish from machine translated data, but the synthetic data is still machine generated and saying natively multilingual gives the impression of it being the data from the native speakers of these languages. I think *multilingual synthetic data* would be a more accurate term, as we do not know how closely LLM generated data resemble native speakers data, especially for low resource languages.

- The improvements on the QAD metric are impressive, but presently it has been evaluated only for a single model i.e. Qwen2.5-3B-Instruct. Hence, I am unsure about how much these results hold in general.

- In Section 5, the authors show interesting ablations on the factors that help in training better multilingual judges, but the analysis does appear preliminary in the present form and there is little discussion on why the observations might hold or their implications. E.g. The authors show that using Qwen 2.5 models as backbone offer significant improvements over the other backbones but do not engage with why that might be the case? It would have been interesting to see the performance of the other backbone models before they were trained on the synthetic data to begin with.

---

> ### Author Response · Authors · 2025-06-02
> **Response (2/2)**
>
> >The improvements on the QAD metric are impressive, but presently it has been evaluated only for a single model i.e. Qwen2.5-3B-Instruct. Hence, I am unsure about how much these results hold in general.
>
> This is a fair point. After receiving the reviews, we ran QAD on top of another model of a different family, Google’s Gemma-2-2b-it, with Qwen2.5-3B and 7B, and M-Prometheus 3B and 7B. The results are the following:
>
> ||French|Chinese|Hindi|
> |---|---|---|---|
> |Qwen2.5 3B|50.72%|49.28%|48.27%|
> |M-Prometheus 3B|58.97%|53.02%|59.39|
> |Qwen2.5 7B|53.88%|48.56%|50.72%|
> |M-Prometheus 7B|59.25%|47.99%|55.02%|
>
> As with Qwen2.5-3B, applying QAD with M-Prometheus yields significant win rate improvements over their respective backbone models—except in Chinese, where M-Prometheus 7B underperforms. Notably, M-Prometheus 3B outperforms its 7B counterpart in Chinese and Hindi.
>
> Due to the time-intensive nature of QAD, we have not run all baselines yet, but we plan to include the full set of experiments in the final version of the paper.
>
> >In Section 5, the authors show interesting ablations on the factors that help in training better multilingual judges, but the analysis does appear preliminary in the present form and there is little discussion on why the observations might hold or their implications. E.g. The authors show that using Qwen 2.5 models as backbone offer significant improvements over the other backbones but do not engage with why that might be the case? It would have been interesting to see the performance of the other backbone models before they were trained on the synthetic data to begin with.
>
> In current literature there is little understanding on how to build strong multilingual judges, as we discuss in the introduction and related work sections. Thus, with these ablations, we set out to understand what factors drove the performance of our system, with a focus on backbone model and training data design (the two principal components in model adaptation). We found that the choice of backbone model and the usage of MT-specific data and multilingual data—as opposed to translated data—were the strongest drivers of performance. The latter finding is particularly interesting, since it contradicts existing literature. We believe Section 5 provides important insights on open questions regarding multilingual LLM judge training, representing relevant scientific contributions. We do not consider this analysis to be preliminary.
>
> That said, studying the factors behind why certain backbone models perform better than others is an interesting derivative question. It is not easy to answer with much certainty, because we are not fully aware of how these models were trained (e.g., the training data is often not released and not many details are given about the training procedure). In the case of Qwen2.5, Aya, EuroLLM, and Mistral, we can only say for sure that Qwen2.5 was pretrained on more tokens. In any case, we agree with your suggestion to test the models before finetuning, as it might help explain differences in downstream performance. Here are the results:
>
> ||MM-Eval|M-RewardBench|RewardBench|LitEval|QAD|
> |---|---|---|---|---|---|
> |Mistral-7B-v0.2|0.5031|0.5932|0.6481|0.0958|53.56|
> |EuroLLM-9B|0.5834|0.6288|0.6890|0.0319|55.15|
> |Aya-Expanse-8B|0.5143|0.6332|0.6579|0.0008|52.05|
> |Qwen-2.5-7B|0.6608|0.7801|0.7823|0.1772|55.88|
>
> Qwen dominates the other models across all benchmarks considered, which partly explains its superior performance after finetuning. QAD is an interesting exception, where Qwen outperforms EuroLLM before finetuning, but not after finetuning
>
> We will include these results in the final version of the paper. Thank you for the suggestion.
>
> We believe to have addressed all your main questions, please let us know if any further clarification is necessary. In case our answer alleviates your concerns, we kindly ask you to consider updating your recommendation score.

---

> > ### Comment · Reviewer_n3M7 · 2025-06-03
> > **Thanks for your response**
> >
> > Thanks for the detailed response and running new experiments in a short period of time. I have adjusted my score accordingly.

---

> > ### Author Response · Authors · 2025-06-03
> > **Thank you**
> >
> > We are glad you found our response useful. Thank you again for your comments, and for adjusting your assessment.

---

> ### Author Response · Authors · 2025-06-02
> **Response (1/2)**
>
> Thank you for your review and your detailed comments. We will address your questions below.
>
> >The claims about some of the results in the paper seem slightly exaggerated. On MM-Eval benchmark the authors state that their models are particularly strong on the Safety, Linguistics, and Language Hallucinations categories. However, on examining the results in Figure 3, for the Linguistics category their models perform either as good as the base model or worse when considering 7B and 14B parameter sizes. Only at 3B their models perform slightly better than the base model. Similarly, for the Language Hallucinations category the gains are marginal as well. Only in the Safety Category the authors find very strong improvements with their method, where the base models perform particularly poorly. This also makes the overall improvements by their method on the MM-Eval benchmark seem overly inflated. I am not sure if these results indicate the effectiveness of their method in general or specific to particular tasks? Similarly, authors show strong performance on LitEval metric, but that again might be due to the presence of Machine Translation Eval data in their training data.
>
>
> We appreciate your observation and agree—especially when comparing the 14B models—that M-Prometheus’s performance on the Linguistics category is not as strong as in Safety and Language Hallucinations. This was an oversight on our part, and we will revise the text accordingly in the final version.
>
> That said, the gains in Language Hallucinations are notable, particularly for larger models: M-Prometheus outperforms Qwen by 0.3, 7, and 5 accuracy points (out of 100) at the 3B, 7B, and 14B scales, respectively. We recognize this is not clearly visible in the current plot due to scale and size, so we will include a table to clarify these results in the final version. We note that overall, and considering M-RewardBench as well, the gains from finetuning appear more consistent in smaller models—a potentially interesting finding that we highlight in Lines 203 and 216.
>
> Our training data was not tailored to any specific general capabilities benchmark category, so we have no strong reason to believe the model performs well only on certain tasks. Notably, M-Prometheus performs strongly on QAD, which is derived from M-ArenaHard—a benchmark designed to assess general model capabilities rather than task-specific skills. Success in this benchmark would imply a certain level of general competence. Additionally, although we include MT evaluation data in training, strong performance on LitEval suggests some degree of cross-domain generalization. LitEval focuses on literary machine translation, a domain where even specialized metrics struggle [1]. M-Prometheus surpasses xCOMET, a state-of-the-art MT metric reported in [1].
>
> We took care to build an evaluation suite that tests both task-specific and general abilities. Taken together, the results indicate that M-Prometheus generalizes well across diverse evaluation challenges and does not merely excel at tasks it was trained on. We kindly ask you to revise your assessment of our research, technical, and empirical contributions with this in mind.
>
> [1] Zhang, R., Zhao, W., & Eger, S. (2024). How good are llms for literary translation, really? literary translation evaluation with humans and llms. arXiv preprint arXiv:2410.18697.
>
> >The authors use the term natively multilingual at many places in the paper for the synthetic data generated in different languages, which I feel is a little misleading. I understand that authors use this term to distinguish from machine translated data, but the synthetic data is still machine generated and saying natively multilingual gives the impression of it being the data from the native speakers of these languages. I think multilingual synthetic data would be a more accurate term, as we do not know how closely LLM generated data resemble native speakers data, especially for low resource languages.
>
> Thank you for raising this. Other reviewers share this concern, and we have decided to accept your suggestion and change the naming to “multilingual synthetic data”.

---

### Official Review · Reviewer_oHAe · 2025-05-12

**Rating:** 7
**Confidence:** 4
**Ethics Flag:** 1

**Summary:**

The paper describes a set of language models trained to perform automatic direct assessment and ranking of texts in different languages.
Two types of evaluation are performed: general capability (on 20 languages) and machine translation (on four language pairs).
In addition, applying automatic scores to improve the model outputs has been investigated as well.

**Questions To Authors:**

line 1: evaluating long-form text -- generated also by LLMs, right?

line 15: across all 3 tested languages -- which 3 languages? Source, target?
before that, 4 language pairs were mentioned, and earlier a span of 20 languages was mentioned
therefore, the distribution of languages is unclear

line 40: Those works => The publications  or The work

line 48: What does Prometeus 2 do? What is the advantage/novelty of M-Prometeus?

line 53:  evaluation of literary machine translation evaluation


line 61: across languages: Which languages? How many languages?

Section 2.1
All training data is generated by LLMs, there is no human-generated training data?

"natively multilingual data" -- maybe some texts are not written by native speakers
"original language" would be a better term (to indicate that those are written directly in the given language, not translated from another one)

Related work should be at the beginning, not at the end

**Reasons To Accept:**

Multilingual abilities of language models are still under-investigated, so this is an important topic.

Using scores to improve outputs is also an important application which has not been explored much.

The model and the data will be publicly available.

**Reasons To Reject:**

It seems that almost everything (except LitEval-corpus) is generated by language models without any interaction with human evaluators (all training and test texts and corresponding evaluation scores) .

---

> ### Author Response · Authors · 2025-06-02
> **Response (2/2)**
>
> >Section 2.1 All training data is generated by LLMs, there is no human-generated training data?
> "natively multilingual data" -- maybe some texts are not written by native speakers "original language" would be a better term (to indicate that those are written directly in the given language, not translated from another one)
>
> This has been pointed out by other reviewers, and we agree it can be misleading. We will change the terminology to “synthetic multilingual data”.
>
> >Related work should be at the beginning, not at the end
>
> We will move the section to the beginning in the final version.

---

> > ### Comment · Reviewer_oHAe · 2025-06-06
> >
> > Thank you for the response and clarification.
> > Also thanks for running additional experiments according to the comments of other reviewers.
> > I will adjust my score according to those revisions.

---

> > > ### Author Response · Authors · 2025-06-06
> > > **Response**
> > >
> > > We are glad you found our responses and additional experiments useful. Thank you for your comments, and for adjusting your assessment.

---

> ### Author Response · Authors · 2025-06-02
> **Response (1/2)**
>
> Thank you for your review. We are glad you found our work timely and our contributions useful. We address your comments and questions below.
> >It seems that almost everything (except LitEval-corpus) is generated by language models without any interaction with human evaluators (all training and test texts and corresponding evaluation scores) .
>
> All training data was indeed synthetically generated. Our approach was heavily inspired by Prometheus [1], which was also trained on fully synthetic data and exhibited state-of-the-art results. Indeed, the usage of synthetic data has become increasingly common across applications, from training to evaluation, to great success [2,3,4]. However, as we write in our Ethics Statement, our approach is not meant to substitute, but rather complement, human evaluation, and users should always review outputs, given the potential issues with automatic evaluation. That said, it should be noted that human-annotated data is not free from issues and bias: for example, there is a large body of work on the biases of state-of-the-art machine translation evaluation metrics [5,6,7], which are trained on large amounts of human-annotated data.
>
> For meta-evaluation, however, we were careful to ensure that all benchmarks used were constructed with some degree of human intervention. 1) The Chat and Linguistics categories of MM-Eval were entirely sourced from human data. 2) M-RewardBench instances were filtered by human translators who were hired to find translations with errors or English-specific concepts that do not exist in other languages. 3) LitEval is composed of human annotatations. 4) M-ArenaHard contains the translations of 500 queries written by humans sourced from the Chatbot Arena.
>
> >line 1: evaluating long-form text -- generated also by LLMs, right?
>
> Not necessarily; the long-form text can be written by humans as well. Consider the examples of our test data we mentioned when addressing your previous comment.
>
> >line 15: across all 3 tested languages -- which 3 languages? Source, target? before that, 4 language pairs were mentioned, and earlier a span of 20 languages was mentioned therefore, the distribution of languages is unclear
>
> The 3 languages are detailed in line 175 (French, Chinese, and Hindi), the 4 language pairs are in footnote 3, and the 20 languages are in footnote 2. We will make sure to include these details earlier and more prominently in the final version of the paper.
>
> >Those works => The publications or The work
>
> Thank you, we will switch to “The publications”, since we are referring to multiple works.
>
> >What does Prometeus 2 do? What is the advantage/novelty of M-Prometeus?
>
> The innovations are the strongly enhanced multilingual capabilities (as shown in the results section), the enhanced capability to perform reference-free evaluation, and the capacity to perform machine translation evaluation. The first two innovations are shown in Figure 1. We will make them clearer in the final version.
>
> >line 53: evaluation of literary machine translation evaluation
>
> Thank you, we will fix this in the final version.
>
> >line 61: across languages: Which languages? How many languages?
>
> It’s French, Chinese, and Hindi (L.175). We realize this is not clear, and we will fix this in the final version. Thank you.
>
> [1] Kim, S., Shin, J., Cho, Y., Jang, J., Longpre, S., Lee, H., ... & Seo, M. (2023, October). Prometheus: Inducing fine-grained evaluation capability in language models. In The Twelfth International Conference on Learning Representations.
>
> [2] Wang, Y., Kordi, Y., Mishra, S., Liu, A., Smith, N. A., Khashabi, D., & Hajishirzi, H. (2022). Self-instruct: Aligning language models with self-generated instructions. arXiv preprint arXiv:2212.10560.
>
> [3] Liu, R., Wei, J., Liu, F., Si, C., Zhang, Y., Rao, J., ... & Dai, A. M. (2024). Best practices and lessons learned on synthetic data. arXiv preprint arXiv:2404.07503.
>
> [4] Sprague, Z., Ye, X., Bostrom, K., Chaudhuri, S., & Durrett, G. (2023). Musr: Testing the limits of chain-of-thought with multistep soft reasoning. arXiv preprint arXiv:2310.16049.
>
> [5] Stanovsky, G., Smith, N. A., & Zettlemoyer, L. (2019, July). Evaluating Gender Bias in Machine Translation. In Proceedings of the 57th Annual Meeting of the Association for Computational Linguistics (pp. 1679-1684).
>
> [6] Zouhar, V., Chen, P., Lam, T. K., Moghe, N., & Haddow, B. (2024, November). Pitfalls and Outlooks in Using COMET. In Proceedings of the Ninth Conference on Machine Translation (pp. 1272-1288).
>
> [7] Pombal, J., Guerreiro, N. M., Rei, R., & Martins, A. F. (2025). Adding chocolate to mint: Mitigating metric interference in machine translation. arXiv preprint arXiv:2503.08327.

---

### Official Review · Reviewer_P6dq · 2025-05-12

**Rating:** 8
**Confidence:** 3
**Ethics Flag:** 1

**Summary:**

The authors describe a suite of LLMs (3B, 7B, 14B) based on Qwen2.5-Instruct and fine-tuned as multilingual quality judges.
Different from previous work they evaluate on native non-English data and literary MT. In addition to describing the training recipe and overall quality experiments they provide ablation studies on the main drivers for multilingual performance. The authors also promise to release all models and code.

**Questions To Authors:**

* Do you have an explanation for the underperformance on M-RewardBench?
* Do you have an explanation for the underperformance in the Chat and Reasoning categories?

**Reasons To Accept:**

The released models are a clear benefit to the community. Some of the findings contradict previous results, most notably the non-effectiveness of translated training data for capabilities as a multilingual judge - it makes sense since previous work only evaluated on translated benchmarks. The ablation studies provide some relevant insights.

**Reasons To Reject:**

It's unsatisfying and unclear why the 7B and 14B M-Prometheus models underperform their backbone Qwen2.5-Instruct on M-RewardBench, and on English RewardBench by such a large margin.

---

> ### Author Response · Authors · 2025-06-02
> **Response**
>
> Thank you for your review and for recognizing the potential impact of M-Prometheus and our findings. We address your comments and questions below.
>
> >It's unsatisfying and unclear why the 7B and 14B M-Prometheus models underperform their backbone Qwen2.5-Instruct on M-RewardBench
>
> The gap in performance on M-RewardBench between the Qwen and M-Prometheus models at 7B and 14B parameters is not large: 0.005 and 0.013 accuracy points, respectively. Yet, we agree it is worth discussing. As we wrote in our response to reviewer Aagm, M-RewardBench—a translated benchmark—contains artifacts that reduce fluency (i.e., translationese [3]), and fails to capture native multilingual linguistic patterns [4]. This can mislead evaluator LLMs and inadvertently benefit English-centric models that were not finetuned on multilingual data (which is the case of the Qwen Instruct models) [1,2]. As mentioned in the paper (L.143), these limitations are the primary reason why we include another benchmark, MM-Eval, and also why we evaluate our models on other tasks, like literary MT evaluation and QAD. When considering all evaluation dimensions, M-Prometheus more consistently outperforms baselines.
>
> >Do you have an explanation for the underperformance in the Chat and Reasoning categories?
>
> Figure 3 shows that M-Prometheus and Qwen models perform similarly in the Reasoning category of M-RewardBench (the gap is slightly larger in MM-Eval but remains small), so we focus on the Chat category. The disparity in Chat performance is harder to explain. One possibility is that Qwen Instruct models are specifically designed for strong assistant capabilities, which may enhance their ability to evaluate chat queries. In contrast, while our fine-tuning improves performance on other dimensions, it may inadvertently weaken this capability.
>
> >It's unsatisfying and unclear why the 7B and 14B M-Prometheus models underperform their backbone Qwen2.5-Instruct on [...] English RewardBench by such a large margin.
>
> Models have limited capacity, so to achieve better performance on non-English languages, they often sacrifice performance on English (often referred to as the curse of multilinguality). This is a common finding in the literature of multilingual adaptation of LLMs [5,6]. That said, M-Prometheus 3B outperforming Qwen 3B even in English supports the strength of our results and the soundness of our approach.
>
> Thank you for raising these points. We will include more discussion on them in the final version of the paper.
>
> [1] Chen, P., Yu, S., Guo, Z., & Haddow, B. (2024). Is it good data for multilingual instruction tuning or just bad multilingual evaluation for large language models?. arXiv preprint arXiv:2406.12822.
>
> [2] Plaza, I., Melero, N., Pozo, C.D., Conde, J., Reviriego, P., Mayor-Rocher, M., & Grandury, M. (2024). Spanish and LLM Benchmarks: is MMLU Lost in Translation? ArXiv, abs/2406.17789.
>
> [3] Parker Riley, Isaac Caswell, Markus Freitag, and David Grangier. 2020. Translationese as a Language in “Multilingual” NMT. In Proceedings of the 58th Annual Meeting of the Association for Computational Linguistics, pages 7737–7746, Online. Association for Computational Linguistics.
>
> [4] Son, G., Yoon, D., Suk, J., Aula-Blasco, J., Aslan, M., Kim, V. T., ... & Kim, S. (2024). MM-Eval: A Multilingual Meta-Evaluation Benchmark for LLM-as-a-Judge and Reward Models. arXiv preprint arXiv:2410.17578.
>
> [5] Conneau, A., Khandelwal, K., Goyal, N., Chaudhary, V., Wenzek, G., Guzmán, F., ... & Stoyanov, V. (2019). Unsupervised cross-lingual representation learning at scale. arXiv preprint arXiv:1911.02116.
>
> [6] Xu, Y., Hu, L., Zhao, J., Qiu, Z., Xu, K., Ye, Y., & Gu, H. (2025). A survey on multilingual large language models: Corpora, alignment, and bias. Frontiers of Computer Science, 19(11), 1911362.

---

> > ### Comment · Reviewer_P6dq · 2025-06-06
> >
> > Thank you for your responses!
> > It seems to me the paper would still benefit from some analysis on the performance drop on the Chat category, and ideally showing that it can be successfully mitigated.
> >
> > While I agree with Reviewer n3M7 that the scientific contributions are incremental, I believe this paper and the models will be of great interest to the community and therefore reaffirm my recommendation for acceptance of the paper.

---

> > > ### Author Response · Authors · 2025-06-06
> > > **Response**
> > >
> > > Thank you for your reaffirmation. We agree that it would be interesting to further analyse some aspects of our work, including the category-wise performance of the various models we report.
> > >
> > > We are glad you found our contributions to be of great interest to the community.

---

### Official Review · Reviewer_Aagm · 2025-05-12

**Rating:** 7
**Confidence:** 3
**Ethics Flag:** 1

**Summary:**

This paper presents a data creation approach for finetuning a multilingual LLM for the LLM-as-a-judge use case, specifically training it for direct assessment and pairwise comparison. The data synthesization is based on previous work, the English-only Prometheus-2 collection. The authors take the English score rubrics, and let a commercial LLM generate instructions of variying quality in multiple non-English languages. From this data pairwise preference data can be extracted as well. With this data, a open-weights LLM (Qwen2.5 in varying sizes) is fine-tuned. The resulting models are then compared on multiple benchmarks, including non-English centric reward benchmarks. The results show that the finetuned models do well on pairwise preferences in a multilingual setting, but fail on a translated variant of the popular RewardBench. On the other hand, the model does well on assessing literary translation.

**Questions To Authors:**

- Line 87: I wouldn't consider LLM-generated data as "native".
- Line 95: Can you give an example in the running text?
- Line 108: Can you describe how you deal with the fact that LLMs are still sensitive to the order in which the preference pairs are presented?
- Table 1: Can you give more insight why the finetuned models would do worse on translated data?
- Table 1: Can you also give more insight why the finetuned models would do so much better on the QAD task?

**Reasons To Accept:**

- Improvement of open-weight LLM-as-a-judge models for the multilingual case.

**Reasons To Reject:**

- Unsure why the model would perform worse on translated data, even worse than the base model.

---

> ### Author Response · Authors · 2025-06-02
> **Response**
>
> Thank you for your review. We address your comments and questions below.
> > Unsure why the model would perform worse on translated data, even worse than the base model.
> [...]
> > Can you give more insight why the finetuned models would do worse on translated data?
>
> We assume you are referring to the fact that the Qwen-7B and 14B Instruct models slightly outperform the M-Prometheus models of the same size on M-RewardBench, a translated version of RewardBench (note that M-Prometheus 3B outperforms Qwen Instruct 3B on this benchmark). Models trained on multilingual data often underperform on translated benchmarks [1,2] due to artifacts that reduce fluency (i.e., _translationese_ [3]) and the absence of native multilingual linguistic patterns, which can mislead evaluator LLMs [4]. These issues reflect limitations of the benchmark rather than the models, which is why we also include MM-Eval [4] (see L.143), a benchmark with natively multilingual data whose scores correlate better with multilingual reward model scores than those of M-RewardBench. We will include these insights in the final version of the paper.
>
> > Line 87: I wouldn't consider LLM-generated data as "native".
>
> This is a fair point, and it has also been mentioned by other reviewers. We will change “native” to “synthetic”, as suggested by reviewer n3M7.
>
> > Line 95: Can you give an example in the running text?
>
> Yes, this will be useful for helping readers understand our training data. We will include this in the camera-ready version.
>
> > Can you describe how you deal with the fact that LLMs are still sensitive to the order in which the preference pairs are presented?
>
> We randomize the order in which the preferred and dispreferred answer appear, such that the preferred answer appears first 50% of the time and second on the other 50%. Thank you for pointing this out, we will include this in the final version.
>
> > Can you also give more insight why the finetuned models would do so much better on the QAD task?
>
> We attribute this primarily to two factors. First, our QAD setup uses significantly less translated data compared to other benchmarks (e.g., M-RewardBench). While M-ArenaHard prompts are translated, the candidate answers from Qwen-3B-Instruct are multilingual, and the judge LLM must evaluate them accordingly. We posit that fine-tuning on multilingual data enhances a model’s ability to assess multilingual outputs. Second, fine-tuned judges are specifically trained to assign scores from 1 to 5, which are necessary for QAD evaluation. In contrast, non-fine-tuned models lack this calibration and are likely to produce less meaningful scores, reducing their effectiveness in QAD.
>
> One of the works we cite when describing our QAD setup [5] (see L.173)—which introduces QAD with an LLM judge—reports similar findings: fine-tuned judges often outperform general-purpose instruct models as utility metrics for QAD. We will make sure to include further discussion on this result in the final version of the paper.
>
> [1] Chen, P., Yu, S., Guo, Z., & Haddow, B. (2024). Is it good data for multilingual instruction tuning or just bad multilingual evaluation for large language models?. arXiv preprint arXiv:2406.12822.
>
> [2] Plaza, I., Melero, N., Pozo, C.D., Conde, J., Reviriego, P., Mayor-Rocher, M., & Grandury, M. (2024). Spanish and LLM Benchmarks: is MMLU Lost in Translation? ArXiv, abs/2406.17789.
>
> [3] Parker Riley, Isaac Caswell, Markus Freitag, and David Grangier. 2020. Translationese as a Language in “Multilingual” NMT. In Proceedings of the 58th Annual Meeting of the Association for Computational Linguistics, pages 7737–7746, Online. Association for Computational Linguistics.
>
> [4] Son, G., Yoon, D., Suk, J., Aula-Blasco, J., Aslan, M., Kim, V. T., ... & Kim, S. (2024). MM-Eval: A Multilingual Meta-Evaluation Benchmark for LLM-as-a-Judge and Reward Models. arXiv preprint arXiv:2410.17578.
>
> [5] Wu, I., Fernandes, P., Bertsch, A., Kim, S., Pakazad, S., & Neubig, G. (2024). Better instruction-following through minimum bayes risk. arXiv preprint arXiv:2410.02902.

---

> > ### Comment · Reviewer_Aagm · 2025-06-08
> >
> > Thanks for your reply, I raised my rating to 7.

---

> > > ### Author Response · Authors · 2025-06-08
> > > **Response**
> > >
> > > Thank you again for your comments, and for updating your rating.

---

### Decision · Program_Chairs · 2025-07-08

**Decision:**

Accept

**Comment:**

This paper presents M-Prometheus, multilingual LLM judges fine-tuned from Qwen2.5 models (3B, 7B, 14B) using synthetic multilingual data. The work extends Prometheus-2 to enable direct assessment and pairwise comparison across multiple languages.

## Contributions & Strengths
All four reviewers unanimously praised the community benefit of releasing open-weight multilingual evaluation tools, addressing a significant gap in available resources. Key finding that synthetic multilingual data outperforms translated data challenges existing assumptions and provides valuable guidance for future multilingual judge development.

**Comprehensive Evaluation**: Rigorous assessment across general capabilities (MM-Eval, M-RewardBench), specialized tasks (literary MT evaluation), and downstream applications (Quality-Aware Decoding).

**Experimental Rigor**: Thorough ablation studies identifying backbone model choice and training data composition as critical factors for multilingual judge performance.

## Limitations and Concerns Raised
**Performance Inconsistencies**: Counterintuitive underperformance on M-RewardBench compared to base models, particularly for larger variants, raises questions about overfitting to synthetic data patterns.

**Limited Technical Novelty**: The contribution is primarily an incremental extension of existing methods, lacking fundamental innovations.

**Evaluation Methodology**: Heavy reliance on synthetic data in training and evaluation creates potential echo chamber effects, despite inclusion of human-annotated benchmarks.

**Scope Limitations**: Detailed analysis focuses on only three languages, and single model family QAD evaluation. Authors addressed this with additional experiments.

## Recommendation
**Accept**. While technically incremental, M-Prometheus makes a solid empirical contribution with clear community value. The systematic extension to multilingual settings, comprehensive evaluation framework, and practical utility justify acceptance. The work successfully addresses an important problem area and provides both tools and insights for future multilingual evaluation research.  Finally, the authors should update the paper in light of discussions including clarification made in the review.